# A Unifying View of Linear Function Approximation in Off-Policy Reinforcement Learning through Matrix Splitting and Preconditioning

**Zechen Wu**
Department of Computer Science
Duke University
zechen.wu@duke.edu

**Amy Greenwald**
Department of Computer Science
Brown University
amy_greenwald@brown.edu

**Ronald Parr**
Department of Computer Science
Duke University
parr@cs.duke.edu

## Abstract

In off-policy policy evaluation (OPE) tasks within reinforcement learning, Temporal Difference Learning(TD) and Fitted Q-Iteration (FQI) have traditionally been viewed as differing in the number of updates toward the target value function: TD makes one update, FQI makes an infinite number, and Partial Fitted Q-Iteration (PFQI) performs a finite number. We show that this view is not accurate, and provide a new mathematical perspective under linear value function approximation that unifies these methods as a single iterative method solving the same linear system, but using different matrix splitting schemes and preconditioners. We show that increasing the number of updates under the same target value function, i.e., the target network technique, is a transition from using a constant preconditioner to using a data-feature adaptive preconditioner. This elucidates, for the first time, why TD convergence does not necessarily imply FQI convergence, and establishes tight convergence connections among TD, PFQI, and FQI. Our framework enables sharper theoretical results than previous work and characterization of the convergence conditions for each algorithm, without relying on assumptions about the features (e.g., linear independence). We also provide an encoder-decoder perspective to better understand the convergence conditions of TD, and prove, for the first time, that when a large learning rate doesn't work, trying a smaller one may help. Our framework also leads to the discovery of new crucial conditions on features for convergence, and shows how common assumptions about features influence convergence, e.g., the assumption of linearly independent features can be dropped without compromising the convergence guarantees of stochastic TD in the on-policy setting. This paper is also the first to introduce matrix splitting into the convergence analysis of these algorithms.

## 1 Introduction

In off-policy policy evaluation (OPE) tasks within reinforcement learning, the Temporal Difference (TD) algorithm [36, 38] can be prone to divergence [2], while Fitted Q-Iteration (FQI) [15, 31, 24] is reputed to be more stable [43]. Traditionally, TD and FQI are viewed as differing in the number of updates toward a target value function: TD makes one update, FQI makes an infinite number,

and Partial Fitted Q-Iteration (PFQI) performs a finite number, similar to target networks in Deep Q-Networks (DQN) [27]. Fellows et al. [16] showed that under certain conditions that make FQI converge, PFQI can be stabilized by increasing the number of updates towards the target. The traditional perspective fails to fully capture the convergence connections between these algorithms and may lead to incorrect conclusions. For example, one might erroneously conclude that TD convergence necessarily implies FQI convergence.

This paper focuses on policy evaluation, rather than control, while using linear value function approximation without assuming on-policy sampling. We provide a unifying perspective on linear function approximation, revealing the fundamental convergence conditions of TD, FQI and PFQI, and comprehensively addressing the relationships between them. Our main technical contribution begins in Section 3, where we describe these algorithms as the same iterative method for solving the same *target linear system*, LSTD [10, 9, 28]. The key difference between these methods is their preconditioners, with PFQI using a preconditioner that transitions between that of TD and FQI. However, we also show in Section 8 that the convergence of one method does not necessarily imply convergence of the other. Additionally, we show that the convergence of these algorithms depends solely on two factors: the consistency of the target linear system and how the target linear system is split to formulate the preconditioner and the iterative components.

In Section 4, we analyze the target linear system itself. We examine consistency (existence of solution) and nonsingularity (uniqueness of solution), providing necessary and sufficient conditions for both. We introduce a new condition, *rank invariance*, which is necessary and sufficient to guarantee consistency of the target linear system *regardless of the reward function*. We demonstrate that this condition is quite mild and is naturally satisfied in most cases. Rank invariance, together with linearly independent features, form the necessary and sufficient conditions for the target linear system to have a unique solution. We also demonstrate that when the true Q-functions can be represented by the linear function approximator, any solution of the target linear system corresponds to parameters that realize the Q-function if and only if rank invariance holds.

Sections 5 to 7, study the convergence of FQI, TD, and PFQI, providing necessary and sufficient conditions for convergence of each, with interpretations of these conditions and the components of the fixed points to which they converge. We also consider the impact of various common assumptions about the feature space on convergence. For FQI, when rank invariance holds, the splitting of the target linear system into its iterative components and a preconditioner is a *proper splitting* [4]. This yields relaxed convergence conditions and guarantees a unique fixed point, providing a theoretical explanation for why FQI exhibits greater robustness in convergence in practice. While it is known that on-policy stochastic TD converges assuming a decaying learning rate and linearly independent features [40], we prove that the assumption of linearly independent features can be dropped. For PFQI, we prove that when the features are not linearly independent, increasing the number of updates toward the same target without reducing to a smaller learning rate can cause divergence. In methods that infrequently update the target value function (e.g., DQN), increasing the number of updates toward each target value function can be destabilizing, particularly when the feature representation is poor.

Section 8, uses our results for the convergence of PFQI, TD, and FQI, along with the close connection between their preconditioners, to reveal PFQI's convergence relationship with TD and FQI, elucidating why the convergence of TD and FQI do not necessarily imply convergence of each other.

**Related work** Bertsekas and Tsitsiklis [6] provided early results on convergence and instability of TD. For linearly independent features, Schoknecht [34] provides sufficient conditions for TD convergence. Fellows et al. [16] propose a sufficient condition for TD convergence with general function approximation. Lee and Kim [25] studies finite-sample behavior of TD from a stochastic linear systems perspective, while more recent convergence results in OPE scenarios are documented by Dann et al. [13]. Tsitsiklis and Van Roy [40], and Borkar and Meyn [8] present an ODE-based view connecting expected TD and stochastic TD, allowing application of results from Harold et al. [19], Borkar and Borkar [7], Benveniste et al. [3]. These results establish almost sure convergence of stochastic TD to a fixed-point set, aligning with previous expected TD results [13].

Voloshin et al. [43] empirically evaluates performance of FQI on various OPE tasks, and Perdomo et al. [30] provides finite-sample analyses of FQI and LSTD with linear approximation under linear realizability assumptions. PFQI can be interpreted as adapting the target network structure from

Mnih et al. [27] to the OPE setting. Fellows et al. [16] and Zhang et al. [45] show that under certain conditions ensuring FQI convergence, increasing the number of updates toward the same target value function can also stabilize PFQI. Che et al. [11] shows that under linear function approximation, and numerous assumptions on features, transition dynamics, and sample distributions, increasing updates toward the same target value stabilizes PFQI providing high-probability bounds on estimation error.

The unifying view provided in this paper provides a simpler and clearer path to definitively answering many longstanding questions, while also allowing us to clarify and refine some observations made in previous work. Corrections to previous results in the literature are discussed in detail in Appendix I.

## 2 Preliminaries

The Appendix A.1 provides a review of all linear algebra concepts and notation used herein.

An MDP is a tuple, $(\mathcal{S}, \mathcal{A}, P, R, \gamma)$, where $\mathcal{S}$ is a finite state space, $\mathcal{A}$ is a finite action space, $P : \mathcal{S} \times \mathcal{A} \to \Delta(\mathcal{S})$ is a Markovian transition model, $R : \mathcal{S} \times \mathcal{A} \to \mathbb{R}$ is a reward function, and $0 < \gamma \leq 1$ is a discount factor. We focus on the common $0 < \gamma < 1$ case. A *Q-function*, $Q_\pi : \mathcal{S} \times \mathcal{A} \to \mathbb{R}$, for policy, $\pi : \mathcal{S} \to \Delta(\mathcal{A})$, represents the expected, discounted cumulative rewards starting from $(s, a)$. In vector form, $Q_\pi \in \mathbb{R}^h$, $h = |\mathcal{S} \times \mathcal{A}|$, with: $Q_\pi = R + \gamma \mathbf{P}_\pi Q_\pi = (I - \gamma \mathbf{P}_\pi)^{-1} R$, where $\mathbf{P}_\pi \in \mathbb{R}^{h \times h}$ is the row-stochastic transition matrix induced by $\pi$, $\mathbf{P}_\pi((s, a), (s', a')) = P(s' \mid s, a)\pi(a' \mid s')$, and $R \in \mathbb{R}^h$ is vector form reward function. *Policy evaluation* finds $Q_\pi(s, a)$ for each $(s, a)$. In *on-policy* evaluation, data are sampled following $\pi$. In *off-policy* policy evaluation, they are sampled with distribution $\mu(s, a)$, which can be uniform, user-provided, or implicit in a sampling distribution. In the on-policy setting, any state-actions visited with nonzero probability can be removed from the problem, as it would be impossible to estimate their values under $\pi$ since they can be never visited. We assume $\mu(s, a) > 0$ for every state-action pair that would be visited under $\pi$, i.e., the *assumption of coverage* [39]. We represent $\mu$ as a vector, $\mu \in \mathbb{R}^h$, and $\mathbf{D} = \text{diag}(\mu)$. In an *on-policy setting*, $\mu$ is the stationary distribution: $\mu \mathbf{P}_\pi = \mu$.

In contrast with the tabular setting [14, 21], large state and action spaces require function approximation to represent the Q-function. The linear case is extensively studied because it is amenable to analysis, computationally tractable, and a step towards understanding more complex methods such as neural networks, which typically have linear final layers. State-action pairs are featurized with $d$ features $\phi_1 \dots \phi_d$, and corresponding $d$-dimensional feature vector, $\phi(s, a) \to \mathbb{R}^d$. In matrix form, $\Phi[i, j] = \phi_j((s, a)_i)$, for $(s, a)_i \in |\mathcal{S} \times \mathcal{A}|$. The goal of linear function approximation is to find $\theta$ such that $\Phi\theta = Q_\theta \approx Q$. We focus on a family of common algorithms interpreted as solving for a $\theta$ which satisfies a linear fixed point equation known as LSTD [10, 9, 28]. These algorithms share state-action covariance $(\Sigma_{cov})$ and cross-covariance $(\Sigma_{cr})$ matrices, and mean feature-reward vector[1]:

$$\Sigma_{\text{cov}} := \Phi^\top \mathbf{D} \Phi, \quad \Sigma_{\text{cr}} := \Phi^\top \mathbf{D} \mathbf{P}_\pi \Phi, \quad \theta_{\phi,r} := \Phi^\top \mathbf{D} R \tag{1}$$

### 2.1 Introduction to algorithms

The algorithms described below are presented in their *expected* form, in which the true transition matrices and complete feature vectors are employed. Appendix K provides additional details on the batch setting, in which these quantities are estimated from batches of data.

**FQI** The Fitted Q-iteration [15, 31, 24](FQI) update takes the following form under linear function approximation (detailed introduction in Appendix A.3.1):

$$\theta_{k+1} = \gamma \Sigma_{cov}^\dagger \Sigma_{cr} \theta_k + \Sigma_{cov}^\dagger \theta_{\phi,r} \tag{2}$$

**Stochastic TD and batch TD** Stochastic Temporal Difference Learning (TD) [36, 38] is an iterative stochastic approximation method that does one update per $(s, a, r, s')$ sample. With linear function approximation and learning rate $\alpha \in \mathbb{R}^+$ the update equation is Equation (3). Batch TD (update below) uses the entire dataset instead of samples to update (detailed in Appendix K). A full mathematical derivation of both forms is provided in Appendix A.3.2.

$$\theta_{k+1} = \theta_k - \alpha \left[ \phi(s, a) \left( \phi(s, a)^\top \theta_k - \gamma \phi(s', a')^\top \theta_k - r(s, a) \right) \right] \tag{3}$$

---

[1]For more detailed definition of notation in Section 2, please see Appendix A.2

**Expected TD** This paper largely focuses on expected TD, which can be understood as modeling the expected behavior of a TD-style update applied to the entire state space simultaneously. This abstracts away sample complexity considerations, and focuses attention on mathematical and algorithmic properties rather than statistical ones, but the results in this paper can be easily adapted for stochastic TD and Batch TD (explained in the Appendix E.14). The expected TD update equation with linear function approximation is Equation (4) (a detailed derivation is provided in Appendix A.3.2):

$$\theta_{k+1} = (I - \alpha\Sigma_{cov})\theta_k + \alpha(\gamma\Sigma_{cr}\theta_k + \theta_{\phi,r}) \tag{4}$$

**Partially fitted Q-iteration (PFQI)** PFQI differs from FQI and TD by employing two sets of parameters: target parameters $\theta_k$ and learning parameters $\theta_{k,t}$ [16]. The target parameters $\theta_k$ parameterize the TD target $[\gamma Q_{\theta_k}(s', a') - r(s, a)]$, while the learning parameters $\theta_{k,t}$ parameterize the learning Q-function $Q_{\theta_{k,t}}$. While $\theta_{k,t}$ is updated at every timestep, $\theta_k$ is updated only every $t$ timesteps. In this context, $Q_{\theta_k}$ in the TD target is referred to as the *target value function*, and its value $Q_{\theta_k}(s, a)$ is called the *target value*. After $t$ timesteps, we update the target parameters: $\theta_k = \theta_{k,t}$. DQN [27] popularized this approach, using neural networks as function approximators. The net for the TD target is known as the *Target Network*. When using a linear function approximator, the update equation at each timestep becomes: $\theta_{k,t+1} = (I - \alpha\Sigma_{cov})\theta_{k,t} + \alpha(\gamma\Sigma_{cr}\theta_k + \theta_{\phi,r})$. Modeling the update to $\theta_{k+1}$ as a function of $\theta_k$ (a complete mathematical derivation is provided in Appendix A.3.3):

$$\theta_{k+1} = \left( \alpha\sum_{i=0}^{t-1}(1 - \alpha\Sigma_{cov})^i \gamma\Sigma_{cr} + (I - \alpha\Sigma_{cov})^t \right)\theta_k + \alpha\sum_{i=0}^{t-1}(1 - \alpha\Sigma_{cov})^i \cdot \theta_{\phi,r}.$$

## 3  Unified view: preconditioned iterative method for solving linear system

The typical, vanilla, iterative method for solving a linear system $Ax = b$, where $A \in \mathbb{R}^{n \times n}$ and $b \in \mathbb{R}^n$, is:

$$x_{k+1} = (I - A)\,x_k + b \tag{5}$$

Convergence depends on consistency of the linear system and the properties of $I - A$. Preconditioning via a matrix $M$ can improve convergence [33]. $MAx = Mb$ is called a preconditioned linear system, where nonsingular matrix $M$ is called a **preconditioner**. Its solution is the same as the original linear system. The iterative method to solve this preconditioned system is:

$$x_{k+1} = \underbrace{(I - MA)}_{H}\,x_k + \underbrace{Mb}_{c} \tag{6}$$

Now, convergence depends on the properties of $H$. The choice of preconditioner adjusts the convergence properties of the iterative method without changing the solution.

**Unified view** The three algorithms—TD, FQI, and Partial FQI—are the same iterative method for solving the same linear system / fixed point equation (Equation (7)) but using different preconditioner $M$. We refer to this linear system as the **target linear system**:

$$\underbrace{(\Sigma_{cov} - \gamma\Sigma_{cr})}_{A}\underbrace{\theta}_{x} = \underbrace{\theta_{\phi,r}}_{b}. \tag{7}$$

TD uses a positive constant preconditioner: $M_{\text{TD}} = \alpha I$. FQI uses the inverse of the feature covariance matrix[2]: $M_{\text{FQI}} = \Sigma_{cov}^{-1}$ as a preconditioner. PFQI uses $M_{\text{PFQI}} = \alpha\sum_{i=0}^{t-1}(I - \alpha\Sigma_{cov})^i$ as a preconditioner.[3] Equation (8) provides an example of such a formulation for TD. For detailed calculations and expressions for each algorithm, please refer to Appendix B.1. When the target linear system is consistent, the matrix inversion method used to solve it is exactly LSTD[10, 9, 28]. Therefore, we denote the $A$ matrix and vector $b$ of the target linear system as $A_{\text{LSTD}} = (\Sigma_{cov} - \gamma\Sigma_{cr})$ and $b_{\text{LSTD}} = \theta_{\phi,r}$, and

---

[2]Here, we assume the invertibility of $\Sigma_{cov}$; later, we provide an analysis of FQI without this assumption.

[3]Here, we assume that $\left(\sum_{i=0}^{t-1}(I - \alpha\Sigma_{cov})^i\right)$ is nonsingular for clarity of presentation. However, this assumption does not affect generality. Because $\Sigma_{cov}$ is symmetric positive semidefinite and we can always easily find a scalar $\alpha$ such that $(\alpha\Sigma_{cov})$ has no eigenvalues equal to 1 or 2, under these conditions, Lemma B.2 guarantees that $\left(\sum_{i=0}^{t-1}(I - \alpha\Sigma_{cov})^i\right)$ is nonsingular for any eligible $t$.

$\Theta_{\text{LSTD}}$ as set of solutions of the target linear system, $\Theta_{\text{LSTD}} = \{\theta \in \mathbb{R}^d \mid (\Sigma_{cov} - \gamma\Sigma_{cr})\,\theta = \theta_{\phi,r}\}$. The $H$ matrix, defined as $I - MA$, for TD, FQI and PFQI is $H_{\text{TD}}$, $H_{\text{FQI}}$ and $H_{\text{PFQI}}$, respectively.

$$\underbrace{\theta_{k+1}}_{x_{k+1}} = \underbrace{\left[ I - \underbrace{\alpha I}_{M}\underbrace{(\Sigma_{cov} - \gamma\Sigma_{cr})}_{A} \right]}_{H} \underbrace{\theta_k}_{x_k} + \underbrace{\underbrace{\alpha I}_{M}\underbrace{\theta_{\phi,r}}_{b}}_{c} \tag{8}$$

**Preconditioner transformation**   From above, we can see that TD, FQI, and PFQI differ only in their choice of preconditioners, while other components in their update equations remain the same—they all use $A_{\text{LSTD}}$ as their $A$ matrix and $b_{\text{LSTD}}$ as their $b$ matrix. Looking at the preconditioner matrix ($M$) of each algorithm, it is evident that these preconditioners are strongly interconnected, as demonstrated in Equation (9). When $t = 1$, the preconditioner of TD equals that of PFQI. However, as $t$ increases, the preconditioner of PFQI converges to the preconditioner of FQI. We can clearly see that increasing the number of updates toward the target value function (denoted by $t$)—a technique known as target network [27]—essentially transforms the algorithm from using a constant preconditioner to using the inverse of the covariance matrix as preconditioner, in the context of linear function approximation.

$$\underbrace{\alpha I}_{\text{TD}} \underset{t=1}{\rightleftharpoons} \underbrace{\alpha \sum_{i=0}^{t-1} (I - \alpha\Sigma_{cov})^i}_{\text{PFQI}} \xrightarrow{t\to\infty} \underbrace{\Sigma_{cov}^{-1}}_{\text{FQI}} \tag{9}$$

**FQI without assuming invertible covariance matrix**   Our unified view of FQI uses $M_{\text{FQI}} = \Sigma_{cov}^{-1}$ as the preconditioner to solve the target linear system, but this requires full column rank $\Phi$. When $\Phi$ is not full column rank, we revert to the original form of FQI in (2), which we refer to as the **FQI linear system** (Equation (10)), with solution set $\Theta_{\text{FQI}}$.

$$\underbrace{\left(I - \gamma\Sigma_{cov}^{\dagger}\Sigma_{cr}\right)}_{A_{\text{FQI}}} \underbrace{\theta}_{x} = \underbrace{\Sigma_{cov}^{\dagger}\theta_{\phi,r}}_{b_{\text{FQI}}}. \tag{10}$$

This also implies $H_{\text{FQI}} = I - A_{\text{FQI}}$. See Appendix B.2 for more details, where we also prove Proposition 3.1, showing the relationship between the FQI linear system and the target linear system.

**Proposition 3.1.** *(1)* $\Theta_{LSTD} \supseteq \Theta_{FQI}$. *(2)* $\Theta_{LSTD} = \Theta_{FQI}$ *if and only if* $\text{Rank}\left(\Sigma_{cov} - \gamma\Sigma_{cr}\right) = \text{Rank}\left(I - \gamma\Sigma_{cov}^{\dagger}\Sigma_{cr}\right)$. *(3) If* $\Phi$ *is full column rank,* $\Theta_{LSTD} = \Theta_{FQI}$.

## 4   Singularity and consistency of the target linear system (LSTD system)

**Consistency of the target linear system**   A linear system $Ax = b$ has a solution if and only if $b \in \text{col}(A)$, so the target linear system is consistent if and only if $b_{\text{LSTD}} \in \text{Col}\left(A_{\text{LSTD}}\right)$. Proposition 4.2 provides the necessary and sufficient conditions on consistency for any $R$, i.e., universal consistency. We call this "Rank Invariance" (Condition 4.1). It can be easily achieved and should widely exist, as by Lemma C.2 it holds if and only if $\gamma\Sigma_{cov}^{\dagger}\Sigma_{cr}$ has no eigenvalue equal to 1 (detailed explanation in Appendix C.2). There are many other conditions equivalent to rank invariance as well (see Lemma C.1). Rank invariance and linearly independent features (Condition 4.3) are distinct conditions: One does not necessarily imply the other (explanation in Appendix C.1). Therefore, the existence of a solution to the target linear system cannot be guaranteed solely from the assumption of linearly independent features.

**Condition 4.1** (Rank Invariance). $\text{Rank}\left(\Phi\right) = \text{Rank}\left(\Phi^{\top}\mathbf{D}(I - \gamma\mathbf{P}_{\pi})\Phi\right)$

**Proposition 4.2** (Universal Consistency). *The target linear system:* $\left(\Phi^{\top}\mathbf{D}(I - \gamma\mathbf{P}_{\pi})\Phi\right)\theta = \Phi^{\top}DR$ *is consistent for any* $R \in \mathbb{R}^h$ *if and only if rank invariance holds.*

**Nonsingularity of the target linear system**   Below, we identify rank invariance and linearly independent features (Condition 4.3) as necessary and sufficient conditions for nonsingularity of the target linear system. While rank invariance is not difficult to satisfy if linearly independent features (Condition 4.3) holds, it is nevertheless a necessary condition that was overlooked by previous papers, e.g., Ghosh and Bellemare [17], which mistakenly claimed that linearly independent features (Condition 4.3) alone is sufficient to ensure the uniqueness of the TD fixed point in the off-policy setting, assuming the fixed point exists.

**Condition 4.3** (Linearly Independent Features). $\Phi$ is full column rank (linearly independent columns).

**Condition 4.4** (Nonsingularity Condition). $A_{\text{LSTD}} = \left(\Phi^\top \mathbf{D}(I - \gamma \mathbf{P}_\pi)\Phi\right)$ is nonsingular.

**Proposition 4.5.** $(\Sigma_{cov} - \gamma \Sigma_{cr})$ *is a nonsingular matrix (i.e., Condition 4.4 holds) if and only if $\Phi$ is full column rank (i.e., Condition 4.3 holds) and rank invariance (Condition 4.1) holds.*

**Nonsingularity of the FQI linear system**   Unlike the target linear system, which requires both linearly independent features (Condition 4.3) and rank invariance (Condition 4.1) to ensure the uniqueness of its solution, in Proposition 4.6, we prove that the FQI linear system requires only rank invariance—both as a necessary and sufficient conditions. This highlights the fundamental role of rank invariance and, more importantly, shows that the FQI linear system forms a more robust linear system whose nonsingularity is not restricted by independence assumptions but rather relies on a broadly satisfied condition.

**Proposition 4.6.** $A_{FQI}$ *is nonsingular if and only if rank invariance (Condition 4.1) holds.*

**Over-parameterization**   The consistency and nonsingularity of the target linear system in the over-parameterized setting are analyzed in detail, with results provided in Appendix J.1.

## 4.1   On-policy setting

Proposition 4.7 shows that in the on-policy setting, rank invariance holds, implying that the target linear system is universally consistent, and thus fixed points for TD, FQI, and PFQI necessarily exist. Moreover, when linearly independent features (Condition 4.3) also holds, Proposition 4.5 implies that the target linear system is nonsingular, aligning with Tsitsiklis and Van Roy [40], which proved that in the on-policy setting with linearly independent features, TD has exactly one fixed point.

**Proposition 4.7.** *In the on-policy setting, rank invariance (Condition 4.1) holds.*

## 4.2   Fixed point and linear realizability

**Assumption 4.8** (Linear Realizability). $Q_\pi$ is linearly realizable in a known feature map $\phi : \mathcal{S} \times \mathcal{A} \to \mathbb{R}^d$ if there exists a vector $\theta^\pi \in \mathbb{R}^d$ such that for all $(s,a) \in \mathcal{S} \times \mathcal{A}, Q_\pi(s,a) = \phi(s,a)^\top \theta^\pi$, i.e., $Q_\pi = \Phi \theta^\pi$.

Proposition 4.9 demonstrates three points: 1) the target linear system may remain consistent even when the true Q-function is not realizable ($\Theta_\pi = \emptyset$); 2) if the true Q-function is realizable, the target linear system is necessarily consistent, and every **perfect parameter** (any vector in $\Theta_\pi$) is guaranteed to be included in the solution set of target linear system; 3) when linear realizability holds, rank invariance is both necessary and sufficient to ensure that every solution of target linear system is a perfect parameter, further implying that rank invariance is necessary and sufficient condition to ensure that any fixed points of the iterative algorithm solving target linear system are perfect parameters.

**Proposition 4.9.** *When linear realizability holds (Assumption 4.8), $\Theta_{LSTD} \supseteq \Theta_\pi$ always holds, and $\Theta_{LSTD} = \Theta_\pi$ holds if and only if rank invariance (Condition 4.1) holds.*

# 5   The convergence of FQI

Theorem 5.1, establishes necessary and sufficient conditions for FQI convergence: 1) the linear system must be consistent; 2) $H_{\text{FQI}}$ must be semiconvergent. The fixed point it converges to consists of two components: $\left(I - A_{\text{FQI}} \left(A_{\text{FQI}}\right)^{\text{D}}\right) \theta_0$, a vector from $\text{Ker}\left(A_{\text{FQI}}\right)$ associated with initial point, and $\left(A_{\text{FQI}}\right)^{\text{D}} b_{\text{FQI}}$, the Drazin (group) inverse solution of FQI linear system[4]. A detailed interpretation of the convergence conditions and fixed points is in Appendix D.1.Theorem 5.1, establishes necessary and sufficient conditions for FQI convergence: 1) the linear system must be consistent; 2) $H_{\text{FQI}}$ must be semiconvergent. The fixed point it converges to consists of two components: $\left(I - A_{\text{FQI}} \left(A_{\text{FQI}}\right)^{\text{D}}\right) \theta_0$, a vector from $\text{Ker}\left(A_{\text{FQI}}\right)$ associated with initial point, and

---

[4]The Drazin inverse solution $(A_{\text{FQI}})^{\text{D}} b_{\text{FQI}}$ equals the group inverse solution $(A_{\text{FQI}})^{\#} b_{\text{FQI}}$ (Appendix D.1).

$(A_{\text{FQI}})^{\text{D}} b_{\text{FQI}}$, the Drazin (group) inverse solution of FQI linear system[5]. A detailed interpretation of the convergence conditions and fixed points is in Appendix D.1.

**Theorem 5.1.** *FQI converges for any initial point $\theta_0$ if and only if $(b_{FQI}) \in \text{Col}(A_{FQI})$ and $(H_{FQI} = I - A_{FQI})$ is semiconvergent. It converges to*

$$\left[ (A_{FQI})^{\text{D}} b_{FQI} + \left( I - A_{FQI} (A_{FQI})^{\text{D}} \right) \theta_0 \right] \in \Theta_{LSTD}.$$

## 5.1 Rank invariance

**Proper splitting**   When rank invariance holds, FQI is an iterative method that employs a proper splitting scheme[4][6] to construct its iterative components and a preconditioner for solving the target linear system (Lemma 5.2), which yields significant advantages. For example, the FQI linear system $(A_{\text{FQI}})$ becomes nonsingular (Proposition 4.6), ensuring existence and uniqueness of the solution. This also ensures that 1 is not an eigenvalue of $\gamma \Sigma_{cov}^{\dagger} \Sigma_{cr}$, a common cause of FQI divergence.

**Lemma 5.2.** *If rank invariance (Condition 4.1) holds, $\Sigma_{cov}$ and $\Sigma_{cr}$ are a proper splitting of $(\Sigma_{cov} - \gamma \Sigma_{cr})$.*

Corollary 5.3, addresses the impact of rank invariance on FQI convergence. The nonsingularity of FQI linear system is guaranteed, the set of fixed points is just a single point, and the requirement on $H_{\text{FQI}}(= \gamma \Sigma_{cov}^{\dagger} \Sigma_{cr})$ being semiconvergent is relaxed to $\rho\left(\gamma \Sigma_{cov}^{\dagger} \Sigma_{cr}\right) < 1$. Thus, rank invariance can help the convergence of FQI. Although it doesn't transform the FQI linear system exactly to the target linear system, the solution of the FQI linear system is also solution of the target linear system.

**Corollary 5.3.** *If rank invariance (Condition 4.1) holds, FQI converges for any initial point $\theta_0$ if and only if $\rho\left(\gamma \Sigma_{cov}^{\dagger} \Sigma_{cr}\right) < 1$. It converges to $\left[(I - \gamma \Sigma_{cov}^{\dagger} \Sigma_{cr}) \Sigma_{cov}^{\dagger} \theta_{\phi,r}\right] \in \Theta_{LSTD}$.*

**Linearly independent features and nonsingular FQI linear system**   When $\Phi$ is full column rank, the FQI linear system becomes exactly equivalent to the target linear system (Section 3). Thus, the consistency condition changes to $b_{\text{LSTD}} \in \text{Col}(A_{\text{LSTD}})$, and $\Sigma_{cov}$ becomes invertible. FQI is then an iterative method using $\Sigma_{cov}^{-1}$ as a preconditioner to solve the target linear system, with $M_{\text{FQI}} = \Sigma_{cov}^{-1}$ and $H_{\text{FQI}} = I - M_{\text{FQI}} A_{\text{LSTD}}$. Beyond this, the convergence conditions for FQI remain largely unchanged compared to Theorem 5.1, which lacks the linearly independent features assumption. We conclude that the linearly independent features assumption does not play a crucial role in FQI's convergence but instead determines the specific linear system that FQI is iteratively solving[7]. The nonsingularity of $A_{\text{FQI}}$ is an ideal setting for FQI, guaranteeing the existence and uniqueness of its fixed point, and reducing its necessary and sufficient conditions for convergence to $\rho(H_{\text{FQI}}) < 1$ (Corollary 5.3). The nonsingularity does not depend on linearly independent features but only on rank invariance (Proposition 4.6), which commonly holds in practice, making FQI inherently more robust in convergence. This observation partially explains why FQI is often empirically found to be more stable than TD, whose uniqueness of the fixed point relies on linearly independent features (Condition 4.3).

Previously, Asadi et al. [1], Xiao et al. [44] provided necessary and sufficient conditions for FQI convergence under the linearly independent features assumption and over-parameterized setting, respectively, however, as we detail in Appendix I they are only sufficient conditions.

The over-parameterized setting is discussed in Appendix J.2. Also, the results in this section can be easily adapted to the batch setting, as explained in Appendix K.1.

## 6   The convergence of TD

Theorem 6.1, establishes necessary and sufficient conditions for TD convergence: 1) the linear system must be consistent; 2) $H_{\text{TD}}$ must be semiconvergent. The fixed point it converges to is composed of $\left(I - (A_{\text{LSTD}})(A_{\text{LSTD}})^{\text{D}}\right) \theta_0$, a vector from $\text{Ker}(A_{\text{LSTD}})$ associated with initial point,

---

[5]The Drazin inverse solution $(A_{\text{FQI}})^{\text{D}} b_{\text{FQI}}$ equals the group inverse solution $(A_{\text{FQI}})^{\#} b_{\text{FQI}}$ (Appendix D.1).
[6]$\Sigma_{cov}$ and $\Sigma_{cr}$ form a proper splitting of $(\Sigma_{cov} - \gamma \Sigma_{cr})$
[7]For a detailed conclusion and calculation refer to Appendix D.3

and $(A_{LSTD})^D b_{LSTD}$, the Drazin (group) inverse solution[8] of the target linear system. For a detailed interpretation of the convergence conditions and fixed point, see Appendix E.1.

**Theorem 6.1.** *TD converges for any initial point $\theta_0$ if and only if $b_{LSTD} \in \mathrm{Col}\,(A_{LSTD})$, and $H_{TD}$ is semiconvergent. It converges to $\left[ (A_{LSTD})^D b_{LSTD} + \left( I - (A_{LSTD})(A_{LSTD})^D \right) \theta_0 \right] \in \Theta_{LSTD}$.*

The convergence condition involves the learning rate $\alpha$. We define TD as **stable** when there exists a learning rate that makes TD converge from any initial point $\theta_0$. For the formal definition, refer to Definition E.1. Corollary 6.2, provides necessary and sufficient conditions for the existence of a learning rate that ensures TD convergence. When such a rate exists, Corollary 6.3 identifies all possible values, showing that they form an interval $(0, \epsilon)$, rather than isolated points, aligning with widely held intuitions: When a large learning rate doesn't work, a smaller one may help. It presents so far the sharpest characterization on convergence of TD. The condition "$b_{LSTD} \in \mathrm{Col}\,(A_{LSTD})$ is strictly positive stable" was previously shown to guarantee TD convergence under the assumption of Condition 4.3 [34].

**Corollary 6.2.** *TD is stable if and only if the following 3 conditions hold: (1) Consistency condition: $b_{LSTD} \in \mathrm{Col}\,(A_{LSTD})$ (2) Positive semi-stability condition: $A_{LSTD}$ is positive semi-stable (3)Index condition: $\mathbf{Index}\,(A_{LSTD}) \leq 1$. Additionally, if $A_{LSTD}$ is an M-matrix, the positive semi-stable condition can be relaxed to: $A_{LSTD}$ is nonnegative stable.*

**Corollary 6.3.** *When TD is stable, TD converges if and only if learning rate $\alpha \in (0, \epsilon)$, where*

$$\epsilon = \min_{\lambda \in \sigma(\Sigma_{cov} - \gamma \Sigma_{cr}) \setminus \{0\}} \left( \frac{2 \cdot \Re(\lambda)}{|\lambda|^2} \right).$$

This highlights a fundamental contrast between TD and FQI. Since TD's preconditioner is only a constant, its convergence depends on $A_{LSTD} = \left[ \Phi^\top \mathbf{D}(I - \gamma \mathbf{P}_\pi) \Phi \right]$, an intrinsic property of the target linear system. In contrast, FQI employs a data–feature adaptive preconditioner that alters its convergence characteristics. Moreover, in Appendix E.5, we describe the target linear system as an **encoder-decoder process**, showing that TD convergence requires preservation of the positive semi-stability of the system's dynamics: $\mathbf{D}(I - \gamma \mathbf{P}_\pi)$, which is an M-matrix (Proposition E.9). This explains why TD can diverge [2], even when each state-action pair is represented by linearly independent feature vectors (over-parameterization), and proves that TD convergence is guaranteed when these feature vectors are orthogonal[9] (Proposition E.10).

**Linarly independent features, rank invariance, and nonsingularity**    There may be an expectation that TD is more stable if $\Phi$ is full column rank, but this does not guarantee any of the conditions of Corollary 6.2. Ghosh and Bellemare [17] claimed that under the assumption of Condition 4.3, the necessary and sufficient condition for TD convergence is $A_{LSTD}$ being positive stable, but as we detail in Appendix I, it is only a sufficient condition. Rank invariance ensures only the consistency of the target linear system but does not relax other stability conditions. When the target linear system is nonsingular, the solution of the target linear system (the fixed point of TD) must exist and be unique. The necessary and sufficient condition for TD to be stable reduces to the condition that $A_{LSTD}$ is positive stable. More details about these results are presented in Appendix E.8.

**Over-parameterization**    We also provide convergence results (e.g, necessary and sufficient conditions) in the **over-parameterized setting** in Appendix E.6, and also correct the over-parameterized TD convergence conditions provided in previous literature[44, 11].

**On-policy TD without linearly independent features**    In the on-policy setting, it is well-known that if $\Phi$ has full column rank, then $\left[ \Phi^\top \mathbf{D}(I - \gamma \mathbf{P}_\pi) \Phi \right]$ is positive definite. This property serves as the central piece supporting the proof of TD's convergence [40]. It aligns with and is well-reflected in our off-policy findings in Corollary 6.2, as further explained in Appendix E.13.1). However, when $\Phi$ does not have full column rank, $\left[ \Phi^\top \mathbf{D}(I - \gamma \mathbf{P}_\pi) \Phi \right]$ becomes positive semidefinite [39], a property that no longer guarantees TD stability. We demonstrate that even without assuming $\Phi$ is full rank, $\left[ \Phi^\top \mathbf{D}(I - \gamma \mathbf{P}_\pi) \Phi \right]$ is an RPN matrix (Proposition E.21) and prove that TD is stable without requiring $\Phi$ to have full column rank (Theorem 6.4), relaxing previous the full column rank requirements [40].

---

[8] $(A_{LSTD})^D b_{LSTD} = (A_{LSTD})^\# b_{LSTD}$, which is proved in Appendix E.1

[9] Here, "orthogonal" does not imply "orthonormal," which imposes an additional norm constraint.

**Theorem 6.4.** *In the on-policy setting ($\mu \mathbf{P}_\pi = \mu$), when $\Phi$ is not full column rank, TD is stable.*

**Stochastic TD and Batch TD** It is known that if expected TD converges to a fixed point, then stochastic TD, with decaying step sizes (as per the Robbins-Monro condition [32, 40] or stricter step size conditions), will also converge to a bounded region within the solution set of the fixed point [3, 19, 13, 40]. Therefore, the necessary and sufficient conditions for the convergence of expected TD can be easily extended to stochastic TD, forming necessary and sufficient condition for convergence of stochastic TD to a bounded region of the fixed point's solution set. For example, stochastic TD with decaying step sizes, under the same on-policy setting but without assuming linearly independent features, converges to a bounded region of the fixed point's solution set, a relaxation of conditions in Tsitsiklis and Van Roy [40] that, to our knowledge, has not been previously established. Additionally, for batch TD, By replacing expected symbol with their empirical counterpart (e.g, $\Sigma_{cov} \to \widehat{\Sigma}_{cov}$)[10]. We can convert the convergence results of expected TD to Batch TD.

# 7 The convergence of PFQI

In Theorem 7.1, the necessary and sufficient condition for PFQI convergence is established, comprising two primary conditions: 1) consistency of the target linear system and 2) the semiconvergence of $H_{\text{PFQI}} = I - M_{\text{PFQI}} A_{\text{LSTD}}$. The fixed point is sum of two components: $(M_{\text{PFQI}} A_{\text{LSTD}})^{\text{D}} M_{\text{PFQI}} b_{\text{LSTD}}$, and $\left( I - (M_{\text{PFQI}} A_{\text{LSTD}})(M_{\text{PFQI}} A_{\text{LSTD}})^{\text{D}} \right) \theta_0$, a vector from $\text{Ker}\,(A_{\text{LSTD}})$ associating with the initial point. For a detailed interpretation of the convergence conditions and fixed point, see Appendix F.1. Also, the results in this section can be easily adapted to the batch setting, as explained in Appendix K.3.

**Theorem 7.1.** *PFQI converges for any initial point $\theta_0$ if and only if $b_{LSTD} \in \text{Col}\,(A_{LSTD})$ and $H_{PFQI}$ is semiconvergent. It converges to the following point in $\Theta_{LSTD}$:*

$$(M_{PFQI} A_{LSTD})^{\text{D}} M_{PFQI} b_{LSTD} + \left( I - (M_{PFQI} A_{LSTD})(M_{PFQI} A_{LSTD})^{\text{D}} \right) \theta_0. \tag{11}$$

**Linearly independent features** As we show in Proposition F.2, linearly independent features (Condition 4.3) does not directly relax the convergence conditions above[11]. However, linearly independent features can be indirectly helpful through PFQI's preconditioner, $M_{\text{PFQI}} = \alpha \sum_{i=0}^{t-1} (I - \alpha \Sigma_{cov})^i$. Without it, $H_{\text{PFQI}} = I - M_{\text{PFQI}} A_{\text{LSTD}}$ may diverge (explanation in Appendix F.3), except in some specific cases, like an over-parameterized representation, which we show in Appendix J.3, where the divergent components can be canceled out. Thus, when the features are not linearly independent, taking a large or increasing number of updates under each target value function will most likely not only fail to stabilize the convergence of PFQI, but can make it *more* divergent. This provides a more nuanced understanding of the impact of slowly updated target networks, as commonly used in deep RL. While typically viewed as stabilizing the learning process, they can have the opposite effect if the provided or learned feature representation is not good.

**Rank invariance and nonsingularity** Under rank invariance (Condition 4.1), the consistency condition for the convergence of PFQI can be completely dropped. However, unlike FQI, the other conditions cannot be relaxed. Moreover, for the convergence of PFQI under nonsingularity (Condition 4.4), the fixed point is unique. In this case, $H_{\text{PFQI}}$ must be strictly convergent ($\rho\,(H_{\text{PFQI}}) < 1$) rather than merely semiconvergent. More detailed results are included in Appendix F.4.

**Over-parameterization** The necessary and sufficient conditions for the convergence of PFQI in the over-parameterized setting are provided in Appendix J.3, and the influence of $t$ on its convergence in this setting is also discussed.

# 8 PFQI as transition between TD and FQI

PFQI is often intuitively understood as a step from TD towards FQI, an intuition which suggests that stability might increase as the number of steps $t$ for which the target is held constant increases from 1

---

[10]Detailed definition on each symbol's empirical version, please see Appendix K.

[11]See Appendix F.3 for more details on convergence conditions of FQI with linearly independent features.

(TD) towards infinity (FQI). This intuition is partly supported by Chen et al. [12], which shows a stabilizing effect for target networks under some strong assumptions. This section provides the first general results on the convergence relationships between PFQI and its limiting cases of TD and FQI in the linear value function approximation setting. These results show that the intuitive understanding of these algorithms is mostly correct, but more subtle than it might initially seem, ultimately leading to surprising cases where TD converges but FQI does not, and vice versa.

We begin by considering what TD implies about PFQI. Our result shows a relationship beween $\alpha$ and $t$ rather than an unconditional implication:

**Theorem 8.1.** *(when TD stability → PFQI convergence) If TD is stable, then for any finite $t \in \mathbb{N}$ there exists $\epsilon_t \in \mathbb{R}^+$ that for any $\alpha \in (0, \epsilon_t)$, PFQI converges.*

This relationship only holds when $t$ is finite. If $t \to \infty$, $\epsilon \to 0$ is possible. Next, we consider what PFQI tells us about FQI. As with TD and FQI, the implication is not unconditional:

**Proposition 8.2.** *(when PFQI convergence → FQI convergence) For a full column rank matrix $\Phi$ (satisfying Condition 4.3) and any learning rate $\alpha \in \left(0, \frac{2}{\lambda_{max}(\Sigma_{cov})}\right)$, if there exists an integer $T \in \mathbb{Z}^+$ such that PFQI converges for all $t \geq T$ from any initial point $\theta_0$, then FQI converges from any initial point $\theta_0$.*

One might wonder whether the convergence of FQI implies the convergence of PFQI when the features are linearly independent. This is not sufficient, but under the stronger assumption of a nonsingular target system the relationship does indeed become bidirectional.

**Theorem 8.3. (nonsingular target system: PFQI convergence ↔ FQI convergence)** *When the target linear system is nonsingular, the following statements are equivalent 1) FQI converges from any initial point $\theta_0$. 2) For any learning rate $\alpha \in \left(0, \frac{2}{\lambda_{max}(\Sigma_{cov})}\right)$, there exists an integer $T \in \mathbb{Z}^+$ such that for all $t \geq T$, PFQI converges from any initial point $\theta_0$*

**Surprising counterexamples** Does TD stability imply FQI stability with linearly independent features? Proposition 8.2 and Theorem 8.3 reveal that the convergence of PFQI for any sufficiently large $t$ implies convergence of FQI, which necessarily includes the case as $t \to \infty$. However, the stability of TD does not necessarily guarantee the convergence of PFQI when $t \to \infty$. As $t$ becomes larger, $\epsilon_t$ usually becomes smaller, shrinking the interval $(0, \epsilon_t)$, from which $\alpha$ is safely chosen. As $t \to \infty$, $\epsilon_t$ could approach zero, causing this interval to vanish. Appendix G.3 presents examples with linearly independent features where TD is stable while FQI does not converge, and vice versa. We further analyze and establish conditions under which the convergence of TD and FQI imply each other in Appendix H.

# 9    Discussion

We presented a novel perspective that unifies TD, FQI, and PFQI via matrix splitting and preconditioning, in the context of linear function approximation for OPE. This approach offers key benefits: simplifying convergence analysis, enabling sharper theoretical results, and uncovering crucial conditions and fundamental connections governing each algorithm's convergence. This framework could also give insight into policy optimization. This perspective could be expanded to include other TD variants [36, 39, 37, 38], and possibly nonlinear function approximation. Our results could also potentially inform design of new algorithms with improved convergence properties.

## Acknowledgment

Zechen Wu and Ronald Parr were partially supported by ARO Grant #W911NF2210251.

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

# A   Preliminaries

## A.1   Linear and matrix algebra

Given an $n \times m$ real matrix $A$, let $\mathrm{Col}\,(A)$ and $\mathrm{Row}\,(A)$ denote its column and row spaces, respectively. The null space of $A$, denoted $\mathrm{Ker}\,(A)$, is defined as $\{x \in \mathbb{C}^n \mid Ax = 0\}$. The complementary subspace to $\mathrm{Ker}\,(A)$, denoted $\overline{\mathrm{Ker}}\,(A)$, includes all vectors in $\mathbb{R}^m$ that are not in $\mathrm{Ker}\,(A)$, formally expressed as $\overline{\mathrm{Ker}}\,(A) = \{v \in \mathbb{R}^m \mid v \notin \mathrm{Ker}\,(A)\}$. Any vector $v \in \mathbb{R}^m$ must lie in one of these two subspaces: either $\mathrm{Ker}\,(A)$ or $\overline{\mathrm{Ker}}\,(A)$, but not both. $A \geq 0$ and $A \gg 0$ means matrix $A$ is element-wise nonnegative and positive, respectively. $A$ is **monotone** when $Ax \geq 0$ implies $x \geq 0$, for all $x \in \mathbb{R}^n$. $A^{\mathrm{H}}$ and $v^{\mathrm{H}}$ are the conjugate transpose of matrix $A$ and vector $v$, respectively. Given $A \in \mathbb{R}^{n \times m}$, $A^{\mathrm{D}}$ is the Drazin inverse of matrix $A$, $A^{\#}$ is the group inverse of matrix $A$, and $A^{\dagger}$ is the Moore–Penrose pseudoinverse of matrix $A$. If $\mathrm{Col}\,(A) = \mathrm{Col}\,(A^{\top})$, then $A^{\mathrm{D}} = A^{\dagger}$.

Given an $n \times n$ square matrix $A$ with eigenvalue $\lambda$, $v_\lambda$ is an eigenvector of $A$ whose related eigenvalue is $\lambda$; $\sigma(A)$ is the spectrum of matrix $A$ (the set of its eigenvalues); $\rho(A)$ is the spectral radius of $A$ (the largest absolute value of the eigenvalues); and $\Re(\lambda)$ represents the real part of complex number $\lambda$. We call a matrix $A$ **positive stable** (resp. **non-negative stable**) if the real part of each eigenvalue of $A$ is positive (resp. non-negative), and we call it **positive semi-stable** if the real part of each nonzero eigenvalue is positive. $A$ is **inverse-positive** when $A^{-1}$ exists and $A^{-1} \geq 0$. we define $I$ as the identity matrix, $\mathbf{Index}\,(A)$ denotes the index of $A$, which is the smallest positive integer $k$ s.t. $\mathbb{R}^{n \times n} = \mathrm{Col}\,(A^k) \oplus \mathrm{Ker}\,(A^k)$ (or, equivalently, $\mathrm{Col}\,(A^k) \cap \mathrm{Ker}\,(A^k) = 0$) holds, where the symbol $\oplus$ represents the direct sum of two subspaces. The index of a nonsingular matrix is always 0. When $\mathbf{Index}\,(A) = 1$, $A^{\mathrm{D}} = A^{\#}$. The index of an eigenvalue $\lambda \in \sigma\,(A)$ for a matrix $A$ is defined to be the index of the matrix $(A - \lambda I)$: $\mathrm{index}\,(\lambda) = \mathbf{Index}\,(A - \lambda I)$.

The dimension $\dim\,(\mathcal{V})$ of a vector space $\mathcal{V}$ is defined to be the number of vectors in any basis for $\mathcal{V}$. Given a vector $v \in \mathcal{V}$, $\|v\|_2$ denotes the $\ell_2$-norm for $v$ and $\|v\|_\mu$ denotes the $\mu$-weighted norm for $v$. $\mathrm{alg\,mult}_{\mathbf{A}}(\lambda)$ and $\mathrm{geo\,mult}_{\mathbf{A}}(\lambda)$ are the algebraic and geometric multiplicities, respectively, of eigenvalue $\lambda \in \sigma\,(A)$. If $\mathrm{alg\,mult}_{\mathbf{A}}(\lambda) = 1$, we say that eigenvalue $\lambda$ is **simple**, and if $\mathrm{alg\,mult}_{\mathbf{A}}(\lambda) = \mathrm{geo\,mult}_{\mathbf{A}}(\lambda)$, we say that eigenvalue $\lambda$ is **semisimple**.

**Lemma A.1.** *Given a matrix $A \in \mathbb{C}^{n \times n}$, the spectrum $\sigma\,(I - A)$ of the matrix $(I - A)$ is given by $\{1 - \lambda \mid \forall \lambda \in \sigma\,(A)\}$, and $\forall \lambda \in \sigma\,(A)$, $\mathrm{alg\,mult}_{\mathbf{A}}(\lambda) = \mathrm{alg\,mult}_{\mathbf{I-A}}(1 - \lambda)$ and $\mathrm{geo\,mult}_{\mathbf{A}}(\lambda) = \mathrm{geo\,mult}_{\mathbf{I-A}}(1 - \lambda)$.*

This lemma is proved in Appendix A.1.1. Every theorem, lemma, proposition, and corollary in this paper is accompanied by a complete mathematical proof in the appendix, regardless of whether we provide an intuitive explanation for its validity in the main body of the paper.

**Linear systems:**   Given a matrix $A \in \mathbb{R}^{n \times m}$ and a vector $b \in \mathbb{R}^n$, if there exists $x \in \mathbb{R}^m$ such that $Ax = b$, the linear system $Ax = b$ is called **consistent**. Given a vector $\bar{b} \in \mathbb{R}^{\bar{n}}$ and a matrix $B \in \mathbb{R}^{n \times \bar{n}}$, if the linear system $Ax = B\bar{b}$ is consistent for any $\bar{b} \in \mathbb{R}^{\bar{n}}$, we call this linear system **universally consistent**. If $A$ can be split into two matrices $M$ and $N$, such that $A = M - N$, $M^{-1} \geq 0$, and $N \geq 0$, the splitting is called a **regular splitting** [5, Chapter 5, Note 8.5] [41, 35]. If $M^{-1} \geq 0$ and $M^{-1}N \geq 0$, it is referred to as a **weak regular splitting** [42, Page 95, Definition 3.28] [29]. Lastly, if $A$ can be split into matrices $M$ and $N$ such that $A = M - N$, and additionally $\mathrm{Col}\,(A) = \mathrm{Col}\,(M)$ and $\mathrm{Ker}\,(A) = \mathrm{Ker}\,(M)$, the splitting is called a **proper splitting** [4].

**Positive definite matrices:**   The definition of a positive definite matrix varies slightly throughout the literature. The following definition is consistent with all the papers cited herein:

**Definition A.2.** The matrix $A \in \mathbb{C}^{n \times n}$ is called *positive definite* if $\Re\,(x^{\mathrm{H}} Ax) > 0$, for all $x \in \mathbb{C}^n \backslash \{0\}$.

**Lemma A.3.** *For a $A \in \mathbb{R}^{n \times n}$, $\Re\,(x^{\mathrm{H}} Ax) > 0$, for all $x \in \mathbb{C}^n \backslash \{0\}$, is equivalent to $x^{\top} Ax > 0$, for all $x \in \mathbb{R}^n \backslash \{0\}$.*

From Lemma A.3, we know that a matrix $A \in \mathbb{R}^{n \times n}$ is also *positive definite* if $x^{\top} Ax > 0$, for all $x \in \mathbb{R}^n \backslash \{0\}$.

**Property A.4.** For any positive definite matrix $A \in \mathbb{C}^{n \times n}$, every eigenvalue of $A$ has positive real part, i.e., $\forall \lambda \in \sigma\,(A)$, $\Re(\lambda) > 0$.

Sometimes the definition of a positive definite matrix includes symmetry, leading to the statement that a positive definite matrix has only real positive eigenvalues and is necessarily diagonalizable. However, in this paper, the definition of a positive definite matrix does not require symmetry. Consequently, a positive definite matrix may not have only real positive eigenvalues (as shown in Appendix A.1.2) or be necessarily diagonalizable (as demonstrated in Appendix A.1.3).

**Range Perpendicular to Nullspace (RPN) Matrices**   **RPN matrix** is a class of square matrices whose column space is perpendicular to its nullspace: $\{A \in \mathbb{C}^{n \times n} \mid \text{Col}(A) \perp \text{Ker}(A)\}$ where $\perp$ denotes perpendicularity. this is also equivalent to $\text{Col}(A) = \text{Col}(A^\top) = \text{Row}(A)$, and $\text{Ker}(A) = \text{Ker}(A^\top)$ [26, Page 408], so sometimes it also called Range-Symmetric or EP Matrices. As shown in Property A.5, any RPN matrix necessarily has index less than or equal to 1. The following Lemma A.6 shows the tight connection between RPN matrices and positive definite matrices.

**Property A.5.** If $A \in \mathbb{C}^{n \times n}$ is a singular RPN matrix, then $\textbf{Index}(A) = 1$.

**Lemma A.6.** *For any positive definite matrix $A \in \mathbb{R}^{n \times n}$ and any matrix $X \in \mathbb{R}^{n \times m}$, $X^\top A X$ is an RPN matrix.*

**Semiconvergent matrices**   Definition A.7 provides the definition of semiconvergent matrix, while Proposition A.8 characterizes the conditions under which a matrix qualifies as semiconvergent matrix in terms of its spectral radius and eigenvalues.

**Definition A.7.** [5, Chapter 6, Definition 4.8] A matrix $A \in R^{n \times n}$ is said to be **semiconvergent** whenever $\lim_{j \to \infty} A^j$ exists.

**Proposition A.8.** *[26, Page 630] A matrix $A$ is semiconvergent if and only if $\rho(\mathbf{A}) < 1$ or $\rho(\mathbf{A}) = 1$, where $\lambda = 1$ is the only eigenvalue on the unit circle, and $\lambda = 1$ is semisimple.*

**Z-matrix, M-matrix, and nonnegative matrices**   Definition A.9 provides the definition of a Z-matrix, while an M-matrix is a specific type of Z-matrix, with its definition given in Definition A.10. Notably, the inverse of a nonsingular M-matrix is known to be a nonnegative matrix [5].

**Definition A.9** ($Z$-matrix [5])**.** The class of $Z$-matrices are those matrices whose off-diagonal entries are less than or equal to zero, i.e., matrices of the form: $Z = (z_{ij})$, where $z_{ij} \leq 0$, for all $i \neq j$.

**Definition A.10** (M-matrix [5])**.** Let $A$ be a $n \times n$ real $Z$-matrix. Matrix $A$ is also an M-matrix if it can be expressed in the form $A = sI - B$, where $B = (b_{ij})$, with $b_{ij} \geq 0$, for all $1 \leq i, j \leq n$, and $s \geq \rho(B)$.

### A.1.1   Proof of Lemma A.1

*Proof.* Given a matrix $A \in \mathbb{C}^{n \times n}$, denote its Jordan form as $J$, so there is a nonsingular matrix $P$ that:
$$A = P^{-1}JP.$$
Therefore,
$$I - A = I - P^{-1}JP = P^{-1}P - P^{-1}JP = P^{-1}(I - J)P.$$
Since diagonal entries of matrix $J$ are the eigenvalues of matrix $A$, and from above we can see that $(I - J)$ and $(I - A)$ are similar to each other, the two matrices share the same Jordan form. Moreover, because the Jordan form of $I - J$ is itself, $(I - J)$ is the Jordan form of $(I - A)$, then we know that diagonal entries of matrix $(I - J)$ are the eigenvalues of matrix $(I - A)$, so $\sigma(I - A) = \{1 - \lambda \mid \forall \lambda \in \sigma(A)\}$, and since the size of every Jordan block of $(I - J)$ is the same as of $J$, we have $\forall \lambda \in \sigma(A), \text{alg mult}_{\mathbf{A}}(\lambda) = \text{alg mult}_{\mathbf{I-A}}(1 - \lambda), \text{geo mult}_{\mathbf{A}}(\lambda) = \text{geo mult}_{\mathbf{I-A}}(1 - \lambda)$.   □

### A.1.2   Counterexample for real positive definite matrix having only real positive eigenvalue

Consider the matrix $A = \begin{pmatrix} 2 & -1 \\ 1 & 2 \end{pmatrix}$.

1. **Quadratic Form**: Checking the quadratic form $x^T A x$:
$$x = \begin{pmatrix} x_1 \\ x_2 \end{pmatrix}, \quad x^T A x = (x_1 \quad x_2) \begin{pmatrix} 2 & -1 \\ 1 & 2 \end{pmatrix} \begin{pmatrix} x_1 \\ x_2 \end{pmatrix},$$

$$x^T Ax = 2x_1^2 - x_1 x_2 + x_1 x_2 + 2x_2^2 = 2x_1^2 + 2x_2^2 > 0 \text{ for all } x \neq 0.$$

The quadratic form is positive for all non-zero $x$.

2. **Eigenvalues**: To find the eigenvalues of $A$, solve the characteristic equation $\det(A - \lambda I) = 0$:

$$\det \begin{pmatrix} 2 - \lambda & -1 \\ 1 & 2 - \lambda \end{pmatrix} = (2 - \lambda)(2 - \lambda) - (-1)(1) = \lambda^2 - 4\lambda + 5 = 0.$$

The solutions to the characteristic equation are:

$$\lambda = \frac{4 \pm \sqrt{16 - 20}}{2} = \frac{4 \pm \sqrt{-4}}{2} = 2 \pm i.$$

The eigenvalues are $2 + i$ and $2 - i$, which are complex.

Thus, $A$ is an example of a non-symmetric matrix with a positive quadratic form but having complex eigenvalues.

### A.1.3 Counterexample of a real positive definite matrix as necessarily diagonalizable

An example of a positive definite but non-symmetric matrix that is not diagonalizable is:

$$A = \begin{pmatrix} 1 & 1 \\ 0 & 1 \end{pmatrix}.$$

This matrix is positive definite because:

$$x^\top Ax = \begin{pmatrix} x_1 & x_2 \end{pmatrix} \begin{pmatrix} 1 & 1 \\ 0 & 1 \end{pmatrix} \begin{pmatrix} x_1 \\ x_2 \end{pmatrix} = x_1^2 + (x_1 + x_2)x_2 = x_1^2 + x_1 x_2 + x_2^2 > 0$$

for all $x_1 \neq 0$ and $x_2 \neq 0$. However, $A$ is not diagonalizable because it has a single eigenvalue, $\lambda = 1$, with algebraic multiplicity 2 but geometric multiplicity 1. Thus, it does not have a full set of linearly independent eigenvectors.

Therefore, while $A$ being positive definite implies certain spectral properties, it does not guarantee that $A$ is diagonalizable if $A$ is not symmetric.

### A.1.4 Proof of Lemma A.3

**Lemma A.11** (Restatement of Lemma A.3). *For a $A \in \mathbb{R}^{n \times n}$, $\Re\left(x^H Ax\right) > 0$ for all $x \in \mathbb{C}^n \backslash \{0\}$ is equivalent to $\left(x^\top Ax\right) > 0$ for all $x \in \mathbb{R}^n \backslash \{0\}$.*

*Proof.* Assume $\left(x^\top Ax\right) > 0$ for all $x \in \mathbb{R}^n \backslash \{0\}$, then we know $\left(x^\top A^\top x\right) = \left(x^\top Ax\right)^\top > 0$ for all $x \in \mathbb{R}^n \backslash \{0\}$, so we have $\left(x^\top (A + A^\top)x\right) > 0$ for all $x \in \mathbb{R}^n \backslash \{0\}$. It is clear that $(A + A^\top)$ is a symmetric, real matrix, and by Lemma A.12 we know that this implies $\left(x^H (A + A^\top)x\right) > 0$ for all $x \in \mathbb{C}^n \backslash \{0\}$. Then by Lemma A.13 and the fact that $A$ is real matrix, we obtain $\Re\left(x^H Ax\right) > 0$ for all $x \in \mathbb{C}^n \backslash \{0\}$.

Conversely, assume $\Re\left(x^H Ax\right) > 0$ for all $x \in \mathbb{C}^n \backslash \{0\}$. Since $\left(x^\top Ax\right)$ is real number for all $x \in \mathbb{R}^n \backslash \{0\}$, it follows that $\left(x^\top Ax\right) > 0$

Hence, the proof is complete. $\qquad\square$

**Lemma A.12.** *Given a symmetric matrix $A \in \mathbb{R}^{n \times n}$, if $x^\top Ax > 0$ for all $x \in \mathbb{R}^n \backslash \{0\}$, then $x^H Ax > 0$ for all $x \in \mathbb{C}^n \backslash \{0\}$.*

*Proof.* Given that $A$ is a symmetric real matrix and $x^\top Ax > 0$ for all $x \in \mathbb{R}^n \backslash \{0\}$, we need to show that $x^H Ax > 0$ for all $x \in \mathbb{C}^n \backslash \{0\}$.

Let $x \in \mathbb{C}^n$ be an arbitrary nonzero complex vector. We can write $x$ as $x = \mathbf{u} + i\mathbf{v}$, where $\mathbf{u}$ and $\mathbf{v}$ are real vectors in $\mathbb{R}^n$.

The quadratic form in the complex case is $x^H Ax$:

$$x^H Ax = (\mathbf{u} - i\mathbf{v})^\top A(\mathbf{u} + i\mathbf{v})$$

Expanding the expression, we get:

$$x^H A x = \mathbf{u}^\top A \mathbf{u} - i\mathbf{v}^\top A \mathbf{u} + i\mathbf{u}^\top A \mathbf{v} + \mathbf{v}^\top A \mathbf{v}.$$

Since $A$ is symmetric, $\mathbf{v}^\top A \mathbf{u} = (\mathbf{u}^\top A^\top \mathbf{v})^\top = (\mathbf{u}^\top A \mathbf{v})^\top = \mathbf{u}^\top A \mathbf{v}$. Therefore:

$$x^H A x = \mathbf{u}^\top A \mathbf{u} + \mathbf{v}^\top A \mathbf{v}.$$

Since $\mathbf{u}$ and $\mathbf{v}$ are real vectors, and $A$ is positive definite, we have:

$$\mathbf{u}^\top A \mathbf{u} > 0 \quad \text{for } \mathbf{u} \neq 0$$

$$\mathbf{v}^\top A \mathbf{v} > 0 \quad \text{for } \mathbf{v} \neq 0.$$

For $\mathbf{x} \neq 0$, either $\mathbf{u} \neq 0$ or $\mathbf{v} \neq 0$ (or both). Therefore:

$$\mathbf{u}^\top A \mathbf{u} + \mathbf{v}^\top A \mathbf{v} > 0.$$

Thus, $x^H A x = \mathbf{u}^\top A \mathbf{u} + \mathbf{v}^\top A \mathbf{v} > 0$ for all $x \in \mathbb{C}^n \backslash \{0\}$.

$\square$

**Lemma A.13.** *Given a matrix $A \in C^{n \times n}$ and for all $x \in C^n \backslash \{0\}$, $(x^H (A + A^H)x)$ is real number, and $(x^H (A + A^H)x)$ has the same sign as $\Re (x^H A x)$.*

*Proof.* Define the quadratic form of $A$ as $x^H A x = a + bi$ where $a$ and $b$ are the real part and imaginary part of the complex number, then we have $x^H A^H x = (x^H A x)^H = a - bi$, and we know that

$$x^H (A + A^H) x = x^H A x + x^H A^H x = a + bi + a - bi = 2a,$$

so quadratic form $x^H (A + A^H) x$ is always real for any $x \in \mathbb{C}^n$, and it shares the same sign with $\Re(x^H A x) = a$.

$\square$

**Lemma A.14.** *If a matrix $A \in C^{n \times n}$ is Hermitian, then it is positive definite if and only if $x^H A x > 0$ for all $x \in C^n \backslash \{0\}$.*

*Proof.* Define quadratic form of $A$ as $x^H A x = a + bi$ where $a$ and $b$ are the real part and imaginary part of the complex number, then we have $x^H A^H x = (x^H A x)^H = a - bi$. Because $A$ is Hermitian, $a + bi = a - bi$, which implies $b = 0$, so $x^H A^H x = a$, meaning the quadratic form is always a real number. If $x^H A^H x > 0$, we have:

$$\begin{aligned}
x^H A x &= x^H \left( \frac{1}{2} A + \frac{1}{2} A + \frac{1}{2} A^H - \frac{1}{2} A^H \right) x \\
&= \frac{1}{2} x^H \left( A + A + A^H - A^H \right) x \\
&= \frac{1}{2} x^H \left( A + A^H \right) x + \frac{1}{2} x^H \left( A - A^H \right) x.
\end{aligned} \tag{12}$$

Because $A$ is Hermitian,

$$x^H \left( A - A^H \right) x = x^H A x - x^H A^H x = 0.$$

Therefore, we obtain

$$x^H A x = \frac{1}{2} x^H \left( A + A^H \right) x = \frac{1}{2} (a + bi + a - bi) = a,$$

so $x^H A x$ is real number for all $x \in \mathbb{C}^n$. Subsequently, $\Re(x^H A x) > 0$ for all $x \in \mathbb{C}^n \backslash \{0\}$ is equivalent to $x^H A x > 0$ for all $x \in \mathbb{C}^n \backslash \{0\}$.
$\square$

### A.1.5 Proof of Lemma A.6

**Lemma A.15** (Restatement of Lemma A.6). *For any positive definite matrix $A \in \mathbb{R}^{n \times n}$ and any matrix $X \in \mathbb{R}^{n \times m}$, $X^\top A X$ is an RPN matrix.*

*Proof.* Given a positive definite matrix $A \in \mathbb{R}^{n \times n}$ and a matrix $X \in \mathbb{R}^{n \times m}$, then by the definition of an RPN matrix, we know that $X^\top A X$ is an RPN matrix if and only if

$$\mathrm{Ker}\left(X^\top A X\right) = \mathrm{Ker}\left(\left[X^\top A X\right]^\top\right).$$

First, by Lemma A.16, we know that

$$\mathrm{Ker}\left(X^\top A X\right) = \mathrm{Ker}\left(X\right).$$

Second, by Lemma A.17, it is clear that $X^\top$ is also a positive definite matrix, then the following holds:

$$\mathrm{Ker}\left(X^\top A^\top X\right) = \mathrm{Ker}\left(X\right).$$

Therefore,

$$\mathrm{Ker}\left(X^\top A X\right) = \mathrm{Ker}\left(\left[X^\top A X\right]^\top\right).$$

Hence, $X^\top A X$ is an RPN matrix. $\qquad\square$

**Lemma A.16.** *Given any positive definite matrix $A \in \mathbb{R}^{n \times n}$ and any matrix $X \in \mathbb{R}^{n \times m}$, $\mathrm{Ker}\left(X^\top A X\right) = \mathrm{Ker}\left(X\right)$*

*Proof.*

$$\mathrm{Ker}\left(X^\top A X\right) = \{x \in \mathbb{C}^m | \left(X^\top A X\right)x = 0\} \tag{13}$$
$$\subseteq \{x \in \mathbb{C}^m | x^\mathrm{H} X^\top A X x = 0\} \tag{14}$$
$$= \{x \in \mathbb{C}^m | X x = 0\} \tag{15}$$
$$= \mathrm{Ker}\left(X\right) \tag{16}$$

The step from Equation (14) to Equation (15), is because $A$ is positive definite, and by definition

$$\forall x \in \mathbb{C}^m \setminus \{0\}, \quad \Re(x^\mathrm{H} A x) > 0.$$

So $x^\mathrm{H} X^\top A X x = 0$ iff vector $X x = 0$, so Equation (14)- Equation (15) holds. Next, it is easy to see that

$$\forall x \in \mathrm{Ker}\left(X\right), \left[X^\top A X\right] x =,$$

which means $\mathrm{Ker}\left(X\right) \subseteq \mathrm{Ker}\left(X^\top A X\right)$, so together with $\mathrm{Ker}\left(X^\top A X\right) \subseteq \mathrm{Ker}\left(X\right)$, we can get $\mathrm{Ker}\left(X^\top A X\right) = \mathrm{Ker}\left(X\right)$. $\qquad\square$

**Lemma A.17.** *A conjugate transpose of a positive definite matrix is also a positive definite matrix.*

*Proof.* Given a $n \times n$ positive definite matrix $A$, define the quadratic form of $A$ as $x^\mathrm{H} A x = a + bi$, where $a$ is real part and $b$ is imaginary part. Then, we have $x^\mathrm{H} A^\mathrm{H} x = (x^\mathrm{H} A x)^\mathrm{H} = a - bi$; therefore $\Re(A) = \Re(A^\mathrm{H})$.

Hence, if $\forall x \in \mathbb{C}^n, \Re(x^\mathrm{H} A x) > 0$ then $\forall x \in \mathbb{C}^n, \Re(x^\mathrm{H} A x) > 0$, vice versa. $\qquad\square$

### A.1.6 Proof of Property A.5

**Property A.18** (Restatement of Property A.5). *Given any singular RPN matrix $A \in \mathbb{C}^{n \times n}$,*

$$\mathbf{Index}\left(A\right) = 1.$$

*Proof.* Given an singular RPN matrix $A \in \mathbb{R}^{n \times n}$, by its definition, we have

$$\mathrm{Col}\left(A\right) \perp \mathrm{Ker}\left(A\right),$$

which implies $\mathrm{Col}\left(A\right) \cap \mathrm{Ker}\left(A\right) = 0$. By definition of index of singular matrix, we know that

$$\mathbf{Index}\left(A\right) = 1.$$

$\qquad\square$

## A.2 MDPs

An MDP is classically defined as a tuple, $(\mathcal{S}, \mathcal{A}, P, R, \gamma)$, where $\mathcal{S}$ is a finite state space, $\mathcal{A}$ is a finite action space, $P : \mathcal{S} \times \mathcal{A} \to \Delta(\mathcal{A})$ is a Markovian transition model defining the conditional distribution over next states given the current state and action, where we denote $\Delta(\mathcal{X})$ as the set of probability distributions over a finite set $\mathcal{X}$. $P(s' \mid s, a)$, $R : \mathcal{S} \times \mathcal{A} \to \mathbb{R}$ is a reward function, and $0 < \gamma \le 1$ is a discount factor. The $\gamma = 1$ case requires special treatment, both algorithmically and theoretically, so we focus on the common $0 < \gamma < 1$ case. A *Q-function* $Q_\pi : \mathcal{S} \times \mathcal{A} \to \mathbb{R}$ for a given policy $\pi : \mathcal{S} \to \Delta(\mathcal{A})$ assigns a value to every state-action pair $(s, a) \in \mathcal{S} \times \mathcal{A}$. This value, called the *Q-value*, represents the expected cumulative rewards starting from the given state-action pair. Q-functions can also be represented as a vector $Q_\pi \in \mathbb{R}^h$, where $h = \mid \mathcal{S} \times \mathcal{A} \mid$. The Q-function satisfies the Bellman equation:

$$Q_\pi = R + \gamma \mathbf{P}_\pi Q_\pi = (I - \gamma \mathbf{P}_\pi)^{-1} R,$$

where $\mathbf{P}_\pi \in \mathbb{R}^{h \times h}$ is the Markovian, row-stochastic transition matrix induced by policy $\pi$. The entries of $\mathbf{P}_\pi$ represent the state-action transition probabilities under policy $\pi$, defined as $\mathbf{P}_\pi((s, a), (s', a')) = P(s' \mid s, a)\pi(a' \mid s')$, and $R \in \mathbb{R}^h$ is the reward function in vector form.

**Policy evaluation**  *Policy evaluation* refers to the problem of computing the expected discounted value of a given policy, such as estimating the Q-value for each state-action pair. In *on-policy* policy evaluation, data (state-action pairs) are sampled following the policy being evaluated. Conversely, In *off-policy* policy evaluation, the data sampling does not need to follow the policy being evaluated and is often based on a different behavior policy. State-action pairs are visited according to a distribution $\mu(s, a)$, which can be uniform, user-provided, or implicit in a sampling distribution. For example, $\mu(s, a)$ could be the stationary distribution of an ergodic Markov chain induced by a behavior policy. It is worthwhile to mention that in the on-policy setting, any state-action visited with nonzero probability can be removed from the problem, and in the off-policy setting, it would be impossible to estimate the values under $\pi$ if the state-action pairs would be visited under $\pi$ could never be sampled according to $\mu$ and their consequences were never observed. Therefore, we assume that $\mu(s, a) > 0$ for every state-action pair that would be visited under $\pi$. This assumption is referred to as the *assumption of coverage* [39]. Accordingly, we define $\mu$ as a distribution vector, $\mu \in \mathbb{R}^h$, where each entry represents the sampling probability of a state-action pair. Subsequently, we define the distribution matrix $\mathbf{D} = \text{diag}(\mu)$, which is a nonsingular diagonal matrix with diagonal entries corresponding to the sampling probabilities of each state-action pair. In particular, in an *on-policy setting*, the relationship $\mu \mathbf{P}_\pi = \mu$ holds, meaning that the distribution $\mu$ aligns with the stationary distribution induced by the target policy $\pi$. In contrast, in an *off-policy setting*, $\mu \mathbf{P}_\pi = \mu$ does not necessarily hold, as the sampling distribution $\mu$ may be influenced by a behavior policy that differs from $\pi$.

**Function approximation**  Although the state and action sets $\mathcal{S}$ and $\mathcal{A}$ are assumed to be finite, but the states-action space usually is very large, so it is unrealistic to use a table to represent the value of every state-action (which is known as the tabular setting[14, 21]), so use of function approximation to represent the Q-function is necessary. In such cases, some form of parametric function approximation is frequently used. *Linear Function Approximation* is the most extensively studied form because it is both amenable to analysis and computationally tractable. An additional motivation for studying linear function approximation, despite the growing success and popularity of non-linear methods such as neural networks, is that the final layers of such networks are often linear. Thus, understanding linear function approximation, while of interest in own right, can also be viewed as a stepping stone towards understanding more complex methods.

When function approximation is used, each state-action pair is featurized with a $d$-dimensional feature vector, $\phi(s, a) \to \mathbb{R}^d$, and corresponding feature matrix:

$$\Phi := \begin{bmatrix} \phi((s,a)_1)^\top \\ \phi((s,a)_2)^\top \\ \vdots \\ \phi((s,a)_h)^\top \end{bmatrix} \in \mathbb{R}^{|\mathcal{S} \times \mathcal{A}| \times d}. \tag{17}$$

Given this feature matrix, for some finite-dimensional parameter vector $\theta \in \mathbb{R}^d$, we can build a linear model of the $Q$ function as $Q_\theta(s, a) = \phi(s, a)^T \theta$, for all state-action pairs $(s, a)$. The goal of linear

function approximation is to find $\theta$ such that $\Phi\theta = Q_\theta \approx Q$. In this paper, we focus on a family of commonly used algorithms that can be interpreted as solving for a $\theta$ which satisfies a linear fixed point equation known as LSTD[10, 9, 28]. In the following, we introduce several quantities arising from linear function approximation. The state-action covariance matrix, $\Sigma_{cov}$, and the cross-covariance matrix, $\Sigma_{cr}$, are defined as:

$$\Sigma_{\text{cov}} := \mathop{\mathbb{E}}_{(s,a)\sim\mu} \left[\phi(s,a)\phi(s,a)^\top\right] = \Phi^\top \mathbf{D}\Phi,$$

$$\Sigma_{\text{cr}} := \mathop{\mathbb{E}}_{\substack{(s,a)\sim\mu \\ s'\sim P(\cdot|s,a),a'\sim\pi(s')}} \left[\phi(s,a)\phi(s',a')^\top\right] = \Phi^\top \mathbf{D}\mathbf{P}_\pi\Phi.$$

Additionally, the mean feature-reward vector, $\theta_{\phi,r}$, is given by:

$$\theta_{\phi,r} := \mathop{\mathbb{E}}_{(s,a)\sim\mu} \left[\phi(s,a)r(s,a)\right] = \Phi^\top \mathbf{D}R.$$

### A.3  Introduction to algorithms

#### A.3.1  FQI

Fitted Q-iteration[15, 31, 24] is one of the most popular algorithms for policy evaluation in practice. While typically applied in a batch setting, the expected or population level behavior of FQI is modeled below. In full generality, in every iteration, FQI uses an arbitrary, parametric function approximator, $Q_\theta(s,a)$, and uses some function "Fit" which is an arbitrary regressor to choose parameters, $\theta$ to optimize fit to a target function:

$$\theta_{k+1} = \text{Fit}(\gamma\mathbf{P}_\pi Q_{\theta_k} + R).$$

The more detailed form as:

$$\theta_{k+1} = \arg\min_\theta \mathop{E}_{\substack{(s,a)\sim\mu \\ s'\sim P(\cdot|s,a),a'\sim\pi(s')}} \left[\left(Q_\theta(s,a) - \gamma Q_{\theta_k}(s',a') - r(s,a)\right)^2\right]. \tag{18}$$

When using a linear function approximator $Q_\theta(s,a) = \phi(s,a)^\top\theta$ the update is a shown below. For the detailed derivation from Equation (19) to Equation (20) please see Appendix A.3.4:

$$\theta_{k+1} = \arg\min_\theta \mathop{E}_{\substack{(s,a)\sim\mu \\ s'\sim P(\cdot|s,a),a'\sim\pi(s')}} \left[\left(\phi(s,a)^\top\theta - \gamma\phi(s',a')^\top\theta_k - r(s,a)\right)^2\right] \tag{19}$$

$$= \gamma\Sigma_{cov}^\dagger\Sigma_{cr}\theta_k + \Sigma_{cov}^\dagger\theta_{\phi,r}. \tag{20}$$

**FQI in the batch setting**   Given the datset $\{(s_i, a_i, r_i(s_i, a_i), s_i', a_i')\}_{i=1}^n$, with linear function approximation, at every iteration, the update of FQI involves iterative solving a least squares regression problem. The update equation is:

$$\theta_{k+1} = \arg\min_\theta \sum_{i=1}^n \left(\phi(s_i,a_i)^\top\theta - r(s_i,a_i) - \gamma\phi(s_i',a_i')^\top\theta_k\right)^2 \tag{21}$$

$$= \gamma\widehat{\Sigma}_{cov}^\dagger\widehat{\Sigma}_{cr}\theta_k + \widehat{\Sigma}_{cov}^\dagger\widehat{\theta}_{\phi,r}. \tag{22}$$

#### A.3.2  TD

Temporal Difference Learning (TD)[36, 38] is the progenitor of modern reinforcement learning algorithms. Originally presented as a stochastic approximation algorithm for evaluating state values, it has been extended the evaluate state-action values, and its behavior has been studied in the batch and expected settings as well. When a tabular representation is used, TD is known to converge to the true state values. We review various formulations of TD with linear approximation below.

**Stochastic TD**    TD is known as an iterative stochastic approximation method. Its update equation is Equation (23). When using linear function approximator $Q_\theta(s,a) = \phi(s,a)^\top \theta$, the update equation becomes Equation (25), where $\alpha \in \mathbb{R}^+$ is the learning rate:

$$\theta_{k+1} = \theta_k - \alpha \left[ \nabla_{\theta_k} Q_{\theta_k}(s,a) \left( Q_{\theta_k}(s,a) - \gamma Q_{\theta_k}(s',a') - r(s,a) \right) \right] \tag{23}$$

$$\text{where } (s,a) \sim \mu, s' \sim P(\cdot|s,a), a' \sim \pi(s') \tag{24}$$

$$= \theta_k - \alpha \left[ \phi(s,a) \left( \phi(s,a)^\top \theta_k - \gamma \phi(s',a')^\top \theta_k - r(s,a) \right) \right]. \tag{25}$$

**Batch TD**    In the batch setting / offline policy evaluation setting, TD uses the entire dataset instead of stochastic samples to update:

$$\theta_{k+1} = \theta_k - \alpha \cdot \frac{1}{n} \sum_{i=1}^{n} \left[ \nabla_{\theta_k} Q_{\theta_k}(s,a) \left( Q_{\theta_k}(s,a) - \gamma Q_{\theta_k}(s',a') - r(s,a) \right) \right] \tag{26}$$

$$= \theta_k - \alpha \cdot \frac{1}{n} \sum_{i=1}^{n} \left[ \phi(s,a) \left( \phi(s,a)^\top \theta_k - \gamma \phi(s',a')^\top \theta_k - r(s,a) \right) \right] \tag{27}$$

$$= \theta_k - \alpha \left[ \left( \widehat{\Sigma}_{cov} - \widehat{\Sigma}_{cr} \right) \theta_k - \widehat{\theta}_{\phi,r} \right]. \tag{28}$$

**Expected TD**    This paper largely focuses on expected TD, which can be understood as modeling the behavior of batch TD in expectation. This abstracts away sample complexity considerations, and focuses attention on mathematical and algorithmic properties rather than statistical ones. The expected TD update equation is:

$$\theta_{k+1} = \theta_k - \alpha \underset{\substack{(s,a)\sim\mu \\ s'\sim P(\cdot|s,a), a'\sim\pi(s')}}{E} \left[ \nabla_{\theta_k} Q_{\theta_k}(s,a) \left( Q_{\theta_k}(s,a) - \gamma Q_{\theta_k}(s',a') - r(s,a) \right) \right]. \tag{29}$$

With a linear function approximator $Q_\theta(s,a) = \phi(s,a)^\top \theta$:

$$\theta_{k+1} = \theta_k - \alpha \underset{\substack{(s,a)\sim\mu \\ s'\sim P(\cdot|s,a), a'\sim\pi(s')}}{E} \left[ \phi(s,a) \left( \phi(s,a)^\top \theta_k - \gamma \phi(s',a')^\top \theta_k - r(s,a) \right) \right] \tag{30}$$

$$= \left( I - \alpha(\Sigma_{cov} - \gamma\Sigma_{cr}) \right) \theta_k + \alpha\theta_{\phi,r} \tag{31}$$

$$= \theta_k - \alpha \left( \Phi^\top D\Phi\theta_k - \gamma\Phi^\top D\mathbf{P}_\pi\Phi\theta_k - \Phi DR \right). \tag{32}$$

$$\tag{33}$$

### A.3.3    PFQI

PFQI differs from FQI (Equation (18)) and TD (Equation (29)) by employing two distinct sets of parameters: target parameters $\theta_k$ and learning parameters $\theta_{k,t}$[16]. The target parameters $\theta_k$ parameterize the TD target $[\gamma Q_{\theta_k}(s',a') - r(s,a)]$, while the learning parameters $\theta_{k,t}$ parameterize the learning Q-function $Q_{\theta_{k,t}}$. While $\theta_{k,t}$ is updated at every timestep, $\theta_k$ is updated only every $t$ timesteps. In this context, $Q_{\theta_k}$ in the TD target is referred to as the *target value function*, and its value $Q_{\theta_k}(s,a)$ is called the *target value*. Under a fixed TD target, the expected update equation at each timestep is:

$$\theta_{k,t+1} = \theta_{k,t} - \alpha \underset{\substack{(s,a)\sim\mu \\ s'\sim P(\cdot|s,a), a'\sim\pi(s')}}{E} \left[ \nabla_{\theta_{k,t}} Q_{\theta_{k,t}}(s,a) \left( Q_{\theta_{k,t}}(s,a) - \gamma Q_{\theta_k}(s',a') - r(s,a) \right) \right].$$
$$\tag{34}$$

After $t$ timesteps, we update the target parameters $\theta_t$ with the current learning parameters $\theta_{k,t}$:

$$\theta_k = \theta_{k,t}. \tag{35}$$

DQN [27] famously popularized this two-parameter approach, using neural networks as function approximators. In this case, the function approximator for the TD target is known as the *Target Network*. This technique of increasing the number of updates under each TD target (or target value function) while using two separate parameter sets to stabilize the algorithm is often referred to as the target network approach [16].

When using a linear function approximator $Q_\theta(s,a) = \phi(s,a)^\top \theta$, the update equation at each timestep becomes:

$$\theta_{k,t+1} = \theta_{k,t} - \alpha \underset{\substack{(s,a)\sim\mu \\ s'\sim P(\cdot|s,a),a'\sim\pi(s')}}{E} \left[ \phi(s,a)^\top \left( \phi(s,a)^\top \theta_{k,t} - \gamma\phi(s',a')^\top \theta_k - r(s,a) \right) \right] \quad (36)$$

$$= \theta_{k,t} - \alpha \left( \Phi^\top D\Phi\theta_{k,t} - \Phi^\top D\mathbf{P}_\pi\Phi\theta_k - \Phi DR \right) \quad (37)$$

$$= (I - \alpha\Sigma_{cov})\theta_{k,t} + \alpha(\gamma\Sigma_{cr}\theta_k + \theta_{\phi,r}). \quad (38)$$

Therefore, the update equation for every $t$ timesteps, or in other words, the target parameter update equation is the following:

$$\theta_{k+1} = \left( \alpha \sum_{i=0}^{t-1} (1 - \alpha\Sigma_{cov})^i \gamma\Sigma_{cr} + (I - \alpha\Sigma_{cov})^t \right) \theta_k + \alpha \sum_{i=0}^{t-1} (1 - \alpha\Sigma_{cov})^i \cdot \theta_{\phi,r}. \quad (39)$$

### A.3.4 Derivation of the FQI update equation

$$\theta_{k+1} = \arg\min_\theta \underset{\substack{(s,a)\sim\mu \\ s'\sim P(\cdot|s,a),a'\sim\pi(s')}}{E} \left[ (Q_\theta(s,a) - \gamma Q_{\theta_k}(s',a') - r(s,a))^2 \right] \quad (40)$$

With linear function approximator $Q_\theta(s,a) = \phi(s,a)^\top\theta$:

$$\theta_{k+1} = \arg\min_\theta \underset{\substack{(s,a)\sim\mu \\ s'\sim P(\cdot|s,a),a'\sim\pi(s')}}{E} \left[ \left( \phi(s,a)^\top\theta - \gamma\phi(s',a')^\top \theta_k - r(s,a) \right)^2 \right] \quad (41)$$

$$= \arg\min_\theta \|\Phi\theta - \gamma\mathbf{P}_\pi\Phi\theta_k - R\|_\mu^2 \quad (42)$$

$$= \arg\min_\theta \left\| \underbrace{\mathbf{D}^{\frac{1}{2}}\Phi}_{A} \underbrace{\theta}_{x} - \underbrace{\left( \gamma\mathbf{D}^{\frac{1}{2}}\mathbf{P}_\pi\Phi\theta_k + \mathbf{D}^{\frac{1}{2}}R \right)}_{b} \right\|_2^2. \quad (43)$$

There are two common approaches to minimizing $\|Ax - b\|_2$: solving the projection equation and solving the normal equation. As shown in [26, Page 438], these methods are equivalent for solving this minimization problem. Below, we present the methodology of both approaches.

**The projection equation approach** The projection equation is:

$$Ax = \mathbf{P}_{\text{Col}(A)}b = \left( AA^\dagger \right) b, \quad (44)$$

where $\mathbf{P}_{\text{Col}(A)}$ is the orthogonal projector onto $\text{Col}(A)$, equal to $\left( AA^\dagger \right)$. This method involves first computing the orthogonal projection of $b$ onto $\text{Col}(A)$, namely $\left( AA^\dagger \right) b$, and then finding the coordinates of this projection (i.e., $x$) in the column space of $A$. If we use the projection equation approach to solve Equation (43), we know that the update of $\theta_k$ is:

$$\theta_{k+1} = \{\theta \in \mathbb{R}^d | \mathbf{D}^{\frac{1}{2}}\Phi\theta = \mathbf{D}^{\frac{1}{2}}\Phi(\mathbf{D}^{\frac{1}{2}}\Phi)^\dagger \left( \gamma\mathbf{D}^{\frac{1}{2}}\mathbf{P}_\pi\Phi\theta_k + \mathbf{D}^{\frac{1}{2}}R \right)\} \quad (45)$$

$$= \{\gamma(\mathbf{D}^{\frac{1}{2}}\Phi)^\dagger\mathbf{D}^{\frac{1}{2}}\mathbf{P}_\pi\Phi\theta_k + (\mathbf{D}^{\frac{1}{2}}\Phi)^\dagger\mathbf{D}^{\frac{1}{2}}R + \left( I - (\mathbf{D}^{\frac{1}{2}}\Phi)^\dagger\mathbf{D}^{\frac{1}{2}}\Phi \right) v \mid v \in \mathbb{R}^h\}. \quad (46)$$

The minimal norm solution is:

$$\theta_{k+1} = \gamma(\mathbf{D}^{\frac{1}{2}}\Phi)^\dagger\mathbf{D}^{\frac{1}{2}}\mathbf{P}_\pi\Phi\theta_k + (\mathbf{D}^{\frac{1}{2}}\Phi)^\dagger\mathbf{D}^{\frac{1}{2}}R \quad (47)$$

$$= \gamma \left( \Phi^\top\mathbf{D}\Phi \right)^\dagger \Phi^\top\mathbf{D}\mathbf{P}_\pi\Phi\theta_k + \left( \Phi^\top\mathbf{D}\Phi \right)^\dagger \Phi^\top\mathbf{D}R \quad (48)$$

$$= \gamma\Sigma_{cov}^\dagger\Sigma_{cr}\theta_k + \Sigma_{cov}^\dagger\theta_{\phi,r}. \quad (49)$$

**The normal equation approach** The second method for solving this minimization problem is tosolve the normal equation $A^\top A x = A^\top b$ directly. Therefore, When using the normal equation approach to solve Equation (43), we know that the update of $\theta_k$ is:

$$\theta_{k+1} = \{\theta \in \mathbb{R}^d | \Phi^\top \mathbf{D} \Phi \theta = \gamma \Phi^\top \mathbf{D} \mathbf{P}_\pi \Phi \theta_k + \Phi^\top \mathbf{D} R\} \tag{50}$$

$$= \{\gamma (\Phi^\top \mathbf{D} \Phi)^\dagger \Phi^\top \mathbf{D} \mathbf{P}_\pi \Phi \theta_k + (\Phi^\top \mathbf{D} \Phi)^\dagger \Phi^\top \mathbf{D} R + \left(I - (\mathbf{D}^{\frac{1}{2}} \Phi)^\dagger \mathbf{D}^{\frac{1}{2}} \Phi\right) v \mid v \in \mathbb{R}^h\} \tag{51}$$

$$= \{\gamma (\mathbf{D}^{\frac{1}{2}} \Phi)^\dagger \mathbf{D}^{\frac{1}{2}} \mathbf{P}_\pi \Phi \theta_k + (\mathbf{D}^{\frac{1}{2}} \Phi)^\dagger \mathbf{D}^{\frac{1}{2}} R + \left(I - (\mathbf{D}^{\frac{1}{2}} \Phi)^\dagger \mathbf{D}^{\frac{1}{2}} \Phi\right) v \mid v \in \mathbb{R}^h\}. \tag{52}$$

The minimal norm solution is:

$$\theta_{k+1} = \left(\Phi^\top \mathbf{D} \Phi\right)^\dagger \gamma \Phi^\top \mathbf{D} \mathbf{P}_\pi \Phi \theta_k + \left(\Phi^\top \mathbf{D} \Phi\right)^\dagger \Phi^\top \mathbf{D} R \tag{53}$$

$$= \gamma \Sigma_{cov}^\dagger \Sigma_{cr} \theta_k + \Sigma_{cov}^\dagger \theta_{\phi,r} \tag{54}$$

$$= \gamma \left(\mathbf{D}^{\frac{1}{2}} \Phi\right)^\dagger \mathbf{D}^{\frac{1}{2}} \mathbf{P}_\pi \Phi \theta_k + \left(\mathbf{D}^{\frac{1}{2}} \Phi\right)^\dagger \mathbf{D}^{\frac{1}{2}} R. \tag{55}$$

In summary, as shown above, without assumptions on the chosen features (i.e., on feature matrix $\Phi$), the update at each iteration is not uniquely determined. From Equation (46) and Equation (52), we know that any vector in the set formed by the sum of the minimum norm solution and any vector from the nullspace of $\mathbf{D}^{\frac{1}{2}} \Phi$ can serve as a valid update. In this paper, we choose the minimum norm solution as the update at each iteration. As shown in Equation (49) and Equation (54), this leads to the following FQI update equation:

$$\theta_{k+1} = \gamma \Sigma_{cov}^\dagger \Sigma_{cr} \theta_k + \Sigma_{cov}^\dagger \theta_{\phi,r}. \tag{56}$$

Consequently, we know that when $\Phi$ is full column rank, the FQI update equation is:

$$\theta_{k+1} = \gamma \Sigma_{cov}^{-1} \Sigma_{cr} \theta_k + \Sigma_{cov}^{-1} \theta_{\phi,r}. \tag{57}$$

When $\Phi$ is full row rank in the over-parameterized setting($d \geq h$), with detailed derivations appearing in Lemma A.19, the update equation becomes:

$$\theta_{k+1} = \gamma \Phi^\dagger \mathbf{P}_\pi \Phi \theta_k + \Phi^\dagger R. \tag{58}$$

**Lemma A.19.** *When $\Phi$ is full row rank in over-parameterized setting($d \geq h$), the FQI update equation is:*

$$\theta_{k+1} = \gamma \Phi^\dagger \mathbf{P}_\pi \Phi \theta_k + \Phi^\dagger R. \tag{59}$$

*Proof.* In the setting where $\Phi$ is full row rank, in the over-parameterized setting($d \geq h$), we know that $\left(\mathbf{D}^{\frac{1}{2}}\right)^{-1} \mathbf{D}^{\frac{1}{2}} \Phi \Phi^\top \mathbf{D}^{\frac{1}{2}} = \Phi \Phi^\top \mathbf{D}^{\frac{1}{2}}$ and because $\Phi$ is full row rank, $\Phi \Phi^\dagger = I$, then $\Phi \Phi^\dagger \mathbf{D}^{\frac{1}{2}} \mathbf{D}^{\frac{1}{2}} \Phi = \mathbf{D}^{\frac{1}{2}} \mathbf{D}^{\frac{1}{2}} \Phi$. By Greville [18, Theorem 1], we can get that $\left(\mathbf{D}^{\frac{1}{2}} \Phi\right)^\dagger = \Phi^\dagger \left(\mathbf{D}^{\frac{1}{2}}\right)^{-1}$. Combining this with update equation (Equation (55)), we can rewrite the update equation as:

$$\theta_{k+1} = \gamma \left(\mathbf{D}^{\frac{1}{2}} \Phi\right)^\dagger \mathbf{D}^{\frac{1}{2}} \mathbf{P}_\pi \Phi \theta_k + \left(\mathbf{D}^{\frac{1}{2}} \Phi\right)^\dagger \mathbf{D}^{\frac{1}{2}} R \tag{60}$$

$$= \gamma \Phi^\dagger \left(\mathbf{D}^{\frac{1}{2}}\right)^{-1} \mathbf{D}^{\frac{1}{2}} \mathbf{P}_\pi \Phi \theta_k + \Phi^\dagger \left(\mathbf{D}^{\frac{1}{2}}\right)^{-1} \mathbf{D}^{\frac{1}{2}} R \tag{61}$$

$$= \gamma \Phi^\dagger \mathbf{P}_\pi \Phi \theta_k + \Phi^\dagger R. \tag{62}$$

$\square$

# B   Unified view: preconditioned iterative method for solving the linear system

## B.1   Unified view

One of the key contributions of this work is to show that the three algorithms—TD, FQI, and Partial FQI—are the same iterative method for solving the same target linear system / fixed point equation (Equation (67)), as they share the same coefficient matrix $A = (\Sigma_{cov} - \gamma \Sigma_{cr})$ and coefficient vector $b = \theta_{\phi,r}$.Their only difference is that they rely on different preconditioners $M$, a choice which impacts the ensuing algorithm's convergence properties. the following will connect to each algorithm's update equation to such perspective.

**TD**

$$\underbrace{\theta_{k+1}}_{x_{k+1}} = \underbrace{\left[ I - \underbrace{\alpha I}_{M} \underbrace{(\Sigma_{cov} - \gamma\Sigma_{cr})}_{A} \right]}_{H} \underbrace{\theta_k}_{x_k} + \underbrace{\underbrace{\alpha I}_{M} \underbrace{\theta_{\phi,r}}_{b}}_{c} \tag{63}$$

We denote the preconditioner $M$ of TD as $M_{\text{TD}} = \alpha I$ and define $H_{\text{TD}} = [I - \alpha(\Sigma_{cov} - \gamma\Sigma_{cr})]$.

**FQI**

$$\underbrace{\theta_{k+1}}_{x_{k+1}} = \underbrace{\left[ I - \underbrace{\Sigma_{cov}^{-1}}_{M} \underbrace{(\Sigma_{cov} - \gamma\Sigma_{cr})}_{A} \right]}_{H} \underbrace{\theta_k}_{x_k} + \underbrace{\underbrace{\Sigma_{cov}^{-1}}_{M} \underbrace{\theta_{\phi,r}}_{b}}_{c} \tag{64}$$

We denote the preconditioner $M$ of FQI [12] as $M_{\text{FQI}} = \Sigma_{cov}^{-1}$ and define $H_{\text{FQI}} = \gamma\Sigma_{cov}^{-1}\Sigma_{cr}$.

**PFQI**

$$\underbrace{\theta_{k+1}}_{x_{k+1}} = \underbrace{\left[ I - \underbrace{\alpha\sum_{i=0}^{t-1}(I - \alpha\Sigma_{cov})^i}_{M} \underbrace{(\Sigma_{cov} - \gamma\Sigma_{cr})}_{A} \right]}_{H} \underbrace{\theta_k}_{x_k} + \underbrace{\underbrace{\alpha\sum_{i=0}^{t-1}(I - \alpha\Sigma_{cov})^i}_{M} \underbrace{\theta_{\phi,r}}_{b}}_{c} \tag{65}$$

We denote the preconditioner $M$ of PFQI as $M_{\text{PFQI}} = \alpha\sum_{i=0}^{t-1}(I - \alpha\Sigma_{cov})^i$ and define $H_{\text{PFQI}} = I - \alpha\sum_{i=0}^{t-1}(I - \alpha\Sigma_{cov})^i(\Sigma_{cov} - \gamma\Sigma_{cr})$.[13] Proposition B.1 details the transformation of the traditional the PFQI update equation (Equation (39)) into this form (Equation (65)).

**Preconditioned target linear system (preconditioned fixed point equation):** From above we can easily see that the the fixed point equations of TD, PFQI, and FQI are in form of Equation (66), which is a preconditioned linear system, As previously demonstrated, solving this preconditioned linear system is equivalent to solving the original linear system as it only multiply a nonsingular matrix $M$ on both sides of the original linear system.

$$M \underbrace{(\Sigma_{cov} - \gamma\Sigma_{cr})}_{A} \underbrace{\theta^\star}_{x} = M \underbrace{\theta_{\phi,r}}_{b} \tag{66}$$

**Target linear system (fixed point equation):** Equation (67) presents the original linear system, therefore as well as the fixed point equations of TD, PFQI, and FQI. We refer to this linear system as the **target linear system**.

$$\underbrace{(\Sigma_{cov} - \gamma\Sigma_{cr})}_{A} \underbrace{\theta}_{x} = \underbrace{\theta_{\phi,r}}_{b} \tag{67}$$

**Non-iterative method to solve fixed point equation (LSTD):** From Equation (68), it is evident that if target linear system is consistent, the matrix inversion method used to solve it is exactly LSTD. therefore, we denote the $A$ matrix and vector $b$ of the target linear system as $A_{\text{LSTD}} = (\Sigma_{cov} - \gamma\Sigma_{cr})$ and $b_{\text{LSTD}} = \theta_{\phi,r}$, and $\Theta_{\text{LSTD}}$ as set of solutions of the target linear system, $\Theta_{\text{LSTD}} = \{\theta \in \mathbb{R}^d \mid (\Sigma_{cov} - \gamma\Sigma_{cr})\theta = \theta_{\phi,r}\}$.

$$\underbrace{\theta_{LSTD}}_{x} = \underbrace{(\Sigma_{cov} - \gamma\Sigma_{cr})^\dagger}_{A^\dagger} \underbrace{\theta_{\phi,r}}_{b} \tag{68}$$

---

[12]here we assumed invertibility of $\Sigma_{cov}$, in Section 3 we provide analysis for FQI without this assumption

[13]here we assume $\left(\sum_{i=0}^{t-1}(I - \alpha\Sigma_{cov})^i\right)$ is nonsingular just for clarity of presentation, but it doesn't lose generality, as $\Sigma_{cov}$ is symmetric positive semidefinite, we can easily find a $\alpha$ that $(\alpha\Sigma_{cov})$ have no eigenvalue equal to 1 and 2, in that case Lemma B.2 show us $\left(\sum_{i=0}^{t-1}(I - \alpha\Sigma_{cov})^i\right)$ is nonsingular

**Preconditioner transformation**   From above, we can see that TD, FQI, and PFQI differ only in their choice of preconditioners, while other components in their update equations remain the same—they all use $A_{\text{LSTD}}$ as their $A$ matrix and $b_{\text{LSTD}}$ as their $b$ matrix. Looking at the preconditioner matrix ($M$) of each algorithm, it is evident that these preconditioners are strongly interconnected, as demonstrated in Equation (69). When $t = 1$, the preconditioner of TD equals that of PFQI. However, as $t$ increases, the preconditioner of PFQI converges to the preconditioner of FQI. Therefore, we can clearly see that increasing the number of updates toward the target value function (denoted by $t$)—a technique known as target network [27]—essentially transforms the algorithm from using a constant preconditioner to using the inverse of the covariance matrix as preconditioner, in the context of linear function approximation.

$$\underbrace{\alpha I}_{\text{TD}} \underset{t=1}{\rightleftharpoons} \underbrace{\alpha \sum_{i=0}^{t-1} (I - \alpha\Sigma_{cov})^i}_{\text{PFQI}} \xrightarrow{t \to \infty} \underbrace{\Sigma_{cov}^{-1}}_{\text{FQI}} \tag{69}$$

## B.2   FQI without assuming invertible covariance matrix

We peviously showed that FQI is a iterative method utilizing $\Sigma_{cov}^{-1}$ as preconditioner to solve the target linear system, but which require $\Phi$ have full column rank. We now study the case without assuming $\Phi$ is full column rank. From Equation (20) , we know general form FQI update equation is:

$$\theta_{k+1} = \gamma\Sigma_{cov}^{\dagger}\Sigma_{cr}\theta_k + \Sigma_{cov}^{\dagger}\theta_{\phi,r},$$

Interestingly, which can be seen as :

$$\underbrace{\theta_{k+1}}_{x_{k+1}} = \left[ I - \underbrace{\left( I - \gamma\Sigma_{cov}^{\dagger}\Sigma_{cr} \right)}_{A} \right] \underbrace{\theta_k}_{x_k} + \underbrace{\Sigma_{cov}^{\dagger}\theta_{\phi,r}}_{b} .$$

which is a vanilla iterative method to solve the linear system:

$$\underbrace{\left( I - \gamma\Sigma_{cov}^{\dagger}\Sigma_{cr} \right)}_{A} \underbrace{\theta}_{x} = \underbrace{\Sigma_{cov}^{\dagger}\theta_{\phi,r}}_{b} . \tag{70}$$

We call this linear system,Equation (70), the **FQI linear system**, and denote the solution set of this linear system, $\Theta_{\text{FQI}}$, with $A$ matrix: $A_{\text{FQI}} = \left( I - \gamma\Sigma_{cov}^{\dagger}\Sigma_{cr} \right)$ and $b_{\text{FQI}} = \Sigma_{cov}^{\dagger}\theta_{\phi,r}$. If we multiply $\Sigma_{cov}$ on both side of linear system, we get a new linear system and this new linear system is our target linear system, and show in Equation (71) (Detailed calculations in Proposition B.4):

$$\Sigma_{cov} \left( I - \gamma\Sigma_{cov}^{\dagger}\Sigma_{cr} \right) \theta = \Sigma_{cov}\Sigma_{cov}^{\dagger}\theta_{\phi,r} \Leftrightarrow \left( \Sigma_{cov} - \gamma\Sigma_{cr} \right) \theta = \theta_{\phi,r} \tag{71}$$

Therefore, we know the target linear system is the projected FQI linear system . Naturally, we have the Proposition 3.1, which shows that any solution of FQI linear system must also be solution of target linear system and what is necessary and sufficient condition that solution set of FQI linear system is exactly equal to solution set of target linear system, and from which we prove that when chosen features are linearly independent($\Phi$ is full column rank), the solution set of FQI linear system is exactly equal to solution set of target linear system.

## B.3   PFQI

**Proposition B.1.** *PFQI update can be expressed as:*

$$\theta_{k+1} = \left( I - \alpha \underbrace{\sum_{i=0}^{t-1}(I - \alpha\Sigma_{cov})^i}_{M} \underbrace{\left( \Sigma_{cov} - \gamma\Sigma_{cr} \right)}_{A} \right) \theta_k + \underbrace{\alpha \sum_{i=0}^{t-1}(I - \alpha\Sigma_{cov})^i}_{M} \underbrace{\theta_{\phi,r}}_{b} \tag{72}$$

*Proof.* As $\Sigma_{cov}$ is symmetric positive semidefinite matrix, it can be diagonalized into:

$$\Sigma_{cov} = Q^{-1} \begin{bmatrix} 0 & 0 \\ 0 & K_{r \times r} \end{bmatrix} Q$$

where $K_{r \times r}$ is full rank diagonal matrix whose diagonal entries are all positive numbers, and $r = \mathrm{Rank}(\Sigma_{cov})$, and $Q$ is the matrix of eigenvectors. It is straightforward to choose a scalar $\alpha$ such that $(I - \alpha K_{r \times r})$ nonsingular, so we will assume $(I - \alpha K_{r \times r})$ as nonsingular matrix for rest of proof. For notational simplicity, we will henceforth denote $K_{r \times r}$ as $K$.

From above, we can derive that

$$\alpha \sum_{i=0}^{t-1} (I - \alpha \Sigma_{cov})^i = Q^{-1} \begin{bmatrix} (\alpha t) I & 0 \\ 0 & (I - (I - \alpha K)^t) K^{-1} \end{bmatrix} Q$$

By Lemma B.2 we know that $\alpha \sum_{i=0}^{t-1} (I - \alpha \Sigma_{cov})^i$ is invertible, subsequently its inverse is:

$$\left( \alpha \sum_{i=0}^{t-1} (I - \alpha \Sigma_{cov})^i \right)^{-1} = Q^{-1} \begin{bmatrix} \frac{1}{\alpha t} I & 0 \\ 0 & K (I - (I - \alpha K)^t)^{-1} \end{bmatrix} Q$$

Therefore, the PFQI update can be rewritten as:

$$\theta_{k+1} = \left( \alpha \gamma \sum_{i=0}^{t-1} (1 - \alpha \Sigma_{cov})^i \Sigma_{cr} + (I - \alpha \Sigma_{cov})^t \right) \theta_k + \alpha \sum_{i=0}^{t-1} (1 - \alpha \Sigma_{cov})^i \cdot \theta_{\phi,r} \tag{73}$$

$$= \left[ \alpha \sum_{i=0}^{t-1} (I - \alpha \Sigma_{cov})^i \left( \gamma \Sigma_{cr} + \left( \alpha \sum_{i=0}^{t-1} (I - \alpha \Sigma_{cov})^i \right)^{-1} (I - \alpha \Sigma_{cov})^t \right) \right] \theta_k \tag{74}$$

$$+ \alpha \sum_{i=0}^{t-1} (I - \alpha \Sigma_{cov})^i \theta_{\phi,r} \tag{75}$$

$$= \left\{ Q^{-1} \begin{bmatrix} \alpha t I & 0 \\ 0 & (I - (I - \alpha K)^t) K^{-1} \end{bmatrix} Q \right. \tag{76}$$

$$\left. \cdot \left( \gamma \Sigma_{cr} + Q^{-1} \begin{bmatrix} \frac{1}{\alpha t} I & 0 \\ 0 & K (I - (I - \alpha K)^t)^{-1} (I - \alpha K)^t \end{bmatrix} Q \right) \right\} \theta_k \tag{77}$$

$$+ \alpha \sum_{i=0}^{t-1} (I - \alpha \Sigma_{cov})^i \theta_{\phi,r} \tag{78}$$

$$= Q^{-1} \begin{bmatrix} \alpha t I & 0 \\ 0 & (I - (I - \alpha K)^t) K^{-1} \end{bmatrix} Q \cdot \left\{ Q^{-1} \begin{bmatrix} \frac{1}{\alpha t} I & 0 \\ 0 & K (I - (I - \alpha K)^t)^{-1} \end{bmatrix} Q \right. \tag{79}$$

$$\left. - \left( Q^{-1} \begin{bmatrix} 0 & 0 \\ 0 & K \end{bmatrix} Q - \gamma \Sigma_{cr} \right) \right\} \cdot \theta_k + \alpha \sum_{i=0}^{t-1} (I - \alpha \Sigma_{cov})^i \theta_{\phi,r} \tag{80}$$

$$= \alpha \sum_{i=0}^{t-1} (I - \alpha \Sigma_{cov})^i \left[ \left( \alpha \sum_{i=0}^{t-1} (I - \alpha \Sigma_{cov})^i \right)^{-1} - (\Sigma_{cov} - \gamma \Sigma_{cr}) \right] \theta_k \tag{81}$$

$$+ \alpha \sum_{i=0}^{t-1} (I - \alpha \Sigma_{cov})^i \theta_{\phi,r} \tag{82}$$

$$= \left( I - \underbrace{\alpha \sum_{i=0}^{t-1} (I - \alpha \Sigma_{cov})^i}_{M} \underbrace{(\Sigma_{cov} - \gamma \Sigma_{cr})}_{A} \right) \theta_k + \underbrace{\alpha \sum_{i=0}^{t-1} (I - \alpha \Sigma_{cov})^i}_{M} \underbrace{\theta_{\phi,r}}_{b} \tag{83}$$

$\square$

**Lemma B.2.** *Given any symmetric positive semidefinite matrix $A$ and scalar $\alpha > 0$, if $(I - \alpha A)$ is invertible and $(\alpha A)$ have no eigenvalue equal to 2, then $\sum_{i=0}^{t-1} (I - \alpha A)^i$ is invertible for any positive integer $t$.*

*Proof.* Given any symmetric positive semidefinite matrix $A$ and $(I - \alpha A)$ is invertible, it can be diagonalized into the form:

$$A = Q^{-1} \begin{bmatrix} 0 & 0 \\ 0 & K_{r \times r} \end{bmatrix} Q$$

where $K$ is positive definite diagonal matrix with no eigenvalue equal to 2, and $r = \text{Rank}(A)$, so

$$\sum_{i=0}^{t-1} (I - \alpha A)^i = Q^{-1} \begin{bmatrix} tI & 0 \\ 0 & (I - (I - \alpha K)^t) K^{-1} \end{bmatrix} Q$$

and by Lemma B.3 we know that $(I - (I - \alpha K)^t)$ is invertible, then clearly

$$\begin{bmatrix} tI & 0 \\ 0 & (I - (I - \alpha K)^t) K^{-1} \end{bmatrix}$$

is full rank, therefore, $\sum_{i=0}^{t-1} (I - \alpha A)^i$ is invertible. $\square$

**Lemma B.3.** *Given any positive definite diagonal matrix $K$ and scalar $\alpha > 0$ and nonnegative integer t, if $(\alpha) K$ have no eigenvalue equal to 2, $(I - (I - \alpha K)^t)$ is invertible.*

*Proof.* Since $K$ is positive definite, it has no eigenvalue equal to 0. By Lemma A.1, it follows that $(I - \alpha K)$ has no eigenvalue equal to 1, Consequently, $(I - \alpha K)^t$ have no eigenvalue equal to 1. Applying Lemma A.1 once more, we can see that $(I - (I - \alpha K)^t)$ has no eigenvalue equal to 0, therefore, it is full rank and hence invertible. $\square$

**Proposition B.4.** *FQI using the minimal norm solution as the update is a vanilla iterative method solving the linear system:*

$$\left(I - \gamma \Sigma_{cov}^{\dagger} \Sigma_{cr}\right) \theta = \Sigma_{cov}^{\dagger} \theta_{\phi,r}$$

*and whose projected linear system(multiplying both sides of this equation by $\Sigma_{cov}$) is target linear system:$(\Sigma_{cov} - \gamma \Sigma_{cr}) \theta = \theta_{\phi,r}$*

*Proof.* When FQI use the minimal norm solution as the update, based on the minimal norm solution in Equation (49) and Equation (54), we knwo that the FQI update is:

$$\theta_{k+1} = \gamma \Sigma_{cov}^{\dagger} \Sigma_{cr} \theta_k + \Sigma_{cov}^{\dagger} \theta_{\phi,r} \tag{84}$$

We can rewrite this update as

$$\underbrace{\theta_{k+1}}_{x_{k+1}} = \left[I - \underbrace{\left(I - \gamma \Sigma_{cov}^{\dagger} \Sigma_{cr}\right)}_{A}\right] \underbrace{\theta_k}_{x_k} + \underbrace{\Sigma_{cov}^{\dagger} \theta_{\phi,r}}_{b} \tag{85}$$

and thus interpret Equation (84) as a vanilla iterative method to solve the linear system:

$$\underbrace{\left(I - \gamma \Sigma_{cov}^{\dagger} \Sigma_{cr}\right)}_{A} \underbrace{\theta}_{x} = \underbrace{\Sigma_{cov}^{\dagger} \theta_{\phi,r}}_{b} \tag{86}$$

Left multiplying both sides of this equation by $\Sigma_{cov}$ yields a new linear system, the projected FQI linear system:

$$\Sigma_{cov} \left(I - \gamma \Sigma_{cov}^{\dagger} \Sigma_{cr}\right) \theta = \Sigma_{cov} \Sigma_{cov}^{\dagger} \theta_{\phi,r}$$

By Lemma B.5, we know that

$$\text{Col}\left(\Phi^{\top}\right) = \text{Col}\left(\Sigma_{cov}\right) \supseteq \text{Col}\left(\Sigma_{cr}\right)$$

and

$$\left(\Phi^{\top} \mathbf{D} R\right) \in \text{Col}\left(\Phi^{\top}\right) = \text{Col}\left(\Sigma_{cov}\right)$$

so $\Sigma_{cov} \Sigma_{cov}^{\dagger} \Sigma_{cr} = \Sigma_{cr}$ and $\Sigma_{cov} \Sigma_{cov}^{\dagger} \theta_{\phi,r} = \theta_{\phi,r}$. Therefore, this new linear system can be rewritten as:

$$\left(\Sigma_{cov} - \gamma \Sigma_{cr}\right) \theta = \theta_{\phi,r}$$

which is target linear system.

$\square$

**Lemma B.5.**

$$\text{Col}\left(\Phi^\top \mathbf{D}\Phi\right) = \text{Col}\left(\Phi^\top\right) \supseteq \text{Col}\left(\Phi^\top \mathbf{D}\mathbf{P}_\pi\Phi\right) \tag{87}$$

$$\text{Col}\left(\Phi^\top \mathbf{D}\Phi\right) = \text{Col}\left(\Phi^\top\right) \supseteq \text{Col}\left(\Phi^\top \mathbf{D}(I - \gamma\mathbf{P}_\pi)\Phi\right) \tag{88}$$

$$\text{Ker}\left(\Phi^\top \mathbf{D}\Phi\right) = \text{Ker}\left(\Phi\right) \subseteq \text{Ker}\left(\Phi^\top \mathbf{D}\mathbf{P}_\pi\Phi\right) \tag{89}$$

$$\text{Ker}\left(\Phi^\top \mathbf{D}\Phi\right) = \text{Ker}\left(\Phi\right) \subseteq \text{Ker}\left(\Phi^\top \mathbf{D}(I - \gamma\mathbf{P}_\pi)\Phi\right) \tag{90}$$

*Proof.* By Lemma B.6, we know that

$$\text{Col}\left(\Phi^\top \mathbf{D}\Phi\right) = \text{Col}\left(\Phi^\top \mathbf{D}^{\frac{1}{2}}\right)$$

Since $\mathbf{D}^{\frac{1}{2}}$ is full rank and $\text{Col}\left(\Phi^\top\right) \supseteq \text{Col}\left(\Phi^\top \mathbf{D}\mathbf{P}_\pi\Phi\right)$ naturally holds, we get:

$$\text{Col}\left(\Phi^\top \mathbf{D}^{\frac{1}{2}}\right) = \text{Col}\left(\Phi^\top\right) \supseteq \text{Col}\left(\Phi^\top \mathbf{D}\mathbf{P}_\pi\Phi\right)$$

Next, by Lemma B.6, we know that $\text{Rank}\left(\Phi^\top \mathbf{D}\Phi\right) = \text{Rank}\left(\Phi^\top \mathbf{D}^{\frac{1}{2}}\right) = \text{Rank}\left(\Phi\right)$, which means

$$\dim\left(\text{Ker}\left(\Phi^\top \mathbf{D}\Phi\right)\right) = \dim\left(\text{Ker}\left(\Phi\right)\right)$$

Additionally, we know that $\text{Ker}\left(\Phi^\top \mathbf{D}\Phi\right) \supseteq \text{Ker}\left(\Phi\right)$ and $\text{Ker}\left(\Phi\right) \subseteq \text{Ker}\left(\Phi^\top \mathbf{D}\mathbf{P}_\pi\Phi\right)$ naturally hold, therefore we can conclude that:

$$\text{Ker}\left(\Phi^\top \mathbf{D}\Phi\right) = \text{Ker}\left(\Phi\right) \subseteq \text{Ker}\left(\Phi^\top \mathbf{D}\mathbf{P}_\pi\Phi\right)$$

Since $\text{Col}\left(\Phi^\top \mathbf{D}\Phi\right) \supseteq \text{Col}\left(\Phi^\top \mathbf{D}\mathbf{P}_\pi\Phi\right)$ and $\text{Ker}\left(\Phi^\top \mathbf{D}\Phi\right) \subseteq \text{Ker}\left(\Phi^\top \mathbf{D}\mathbf{P}_\pi\Phi\right)$, naturally,

$$\text{Col}\left(\Phi^\top \mathbf{D}\Phi\right) \supseteq \text{Col}\left(\Phi^\top \mathbf{D}(I - \gamma\mathbf{P}_\pi)\Phi\right) \text{ and } \text{Ker}\left(\Phi^\top \mathbf{D}\Phi\right) \subseteq \text{Ker}\left(\Phi^\top \mathbf{D}(I - \gamma\mathbf{P}_\pi)\Phi\right)$$

$\square$

**Lemma B.6.** *Given any matrix $A \in \mathbb{R}^{n \times m}$,*

$$\text{Col}\left(AA^\top\right) = \text{Col}\left(A\right)$$

*Proof.* Since $\text{Col}\left(A^\top\right) = \text{Row}\left(A\right) \perp \text{Ker}\left(A\right)$ and $\text{Rank}\left(A\right) = \text{Rank}\left(A^\top\right)$, by Lemma B.7 we know that $\text{Rank}\left(AA^\top\right) = \text{Rank}\left(A^\top\right) = \text{Rank}\left(A\right)$, and $\text{Col}\left(AA^\top\right) \subseteq \text{Col}\left(A\right)$ naturally holds. Hence,

$$\text{Col}\left(AA^\top\right) = \text{Col}\left(A\right)$$

$\square$

**Lemma B.7.** *Given any two matrices $A \in \mathbb{R}^{n \times m}$ and $B \in \mathbb{R}^{m \times n}$, Assuming $\text{Rank}\left(A\right) \geq \text{Rank}\left(B\right)$, then*

$$\text{Rank}\left(AB\right) = \text{Rank}\left(B\right)$$

*if and only if $\text{Ker}\left(A\right) \cap \text{Col}\left(B\right) = \{0\}$.*

*Proof.* By [26, Page 210], we know that

$$\text{Rank}\left(AB\right) = \text{Rank}\left(B\right) - \dim\left(\text{Ker}\left(A\right) \cap \text{Col}\left(B\right)\right)$$

Therefore, if and only if $\text{Ker}\left(A\right) \cap \text{Col}\left(B\right) = \{0\}$, $\text{Rank}\left(AB\right) = \text{Rank}\left(B\right)$ $\square$

## B.4 Proof of Proposition 3.1

**Proposition B.8** (Restatement of Proposition 3.1).        • $\Theta_{LSTD} \supseteq \Theta_{FQI}$.

- *if and only if* $\mathrm{Rank}\,(\Sigma_{cov} - \gamma\Sigma_{cr}) = \mathrm{Rank}\,(I - \gamma\Sigma_{cov}^{\dagger}\Sigma_{cr})$, $\Theta_{LSTD} = \Theta_{FQI}$.

- *when* $\Sigma_{cov}$ *is full rank(or* $\Phi$ *is full column rank),* $\Theta_{LSTD} = \Theta_{FQI}$

*Proof.* As we show in Section 3, the target linear system is projected FQI linear system (multiplying both sides of FQI linear system by $\Sigma_{cov}$), every solution of FQI linear system must also be solution of target linear system, which means:

$$\Theta_{\mathrm{LSTD}} \supseteq \Theta_{\mathrm{FQI}}$$

By Lemma C.10, we know that $\Theta_{\mathrm{LSTD}} = \Theta_{\mathrm{FQI}}$ if and only if

$$\mathrm{Rank}\,(\Sigma_{cov} - \gamma\Sigma_{cr}) = \mathrm{Rank}\,(I - \gamma\Sigma_{cov}^{\dagger}\Sigma_{cr})$$

Therefore, when $\Sigma_{cov}$ is full rank, we know that

$$\mathrm{Rank}\,(\Sigma_{cov} - \gamma\Sigma_{cr}) = \mathrm{Rank}\,(\Sigma_{cov}\,(I - \gamma\Sigma_{cov}^{\dagger}\Sigma_{cr})) = \mathrm{Rank}\,(I - \gamma\Sigma_{cov}^{\dagger}\Sigma_{cr})$$

hence $\Theta_{\mathrm{LSTD}} = \Theta_{\mathrm{FQI}}$. □

# C   Singularity and consistency of the linear system

## C.1   Rank invariance and linearly independent features are distinct conditions

The commonly assumed condition for algorithms like TD and FQI — that the features are linearly independent, meaning $\Phi$ has full column rank (Condition 4.3) — does not necessarily imply rank invariance (Condition 4.1), which, by Lemma C.1, is equivalent to:

$$\mathrm{Ker}\,(\Phi^{\top}) \cap \mathrm{Col}\,(\mathbf{D}(I - \gamma\mathbf{P}_{\pi})\Phi) = \{0\} \tag{91}$$

Conversely, rank invariance (Condition 4.1) does not imply $\Phi$ has full column rank. The intuition behind this distinction lies in the fact that $\mathrm{Ker}\,(\Phi^{\top}) \cap \mathrm{Row}\,(\Phi^{\top}) = \{0\}$ naturally holds, leading to $\mathrm{Ker}\,(\Phi^{\top}) \cap \mathrm{Col}\,(\Phi) = \{0\}$. However, the relationship $\mathrm{Col}\,(\mathbf{D}(I - \gamma\mathbf{P}_{\pi})\Phi) = \mathrm{Col}\,(\Phi)$ does not necessarily hold, regardless of whether $\Phi$ has full column rank. Consequently, there is no guarantee that $\mathrm{Ker}\,(\Phi^{\top}) \cap \mathrm{Col}\,(\mathbf{D}(I - \gamma\mathbf{P}_{\pi})\Phi) = \{0\}$ will hold, irrespective of the rank of $\Phi$. Thus, linearly independent features (Condition 4.3) and rank invariance (Condition 4.1) are distinct conditions, with neither necessarily implying the other. Since rank invariance (Condition 4.1) is necessary and sufficient condition for the target linear system to be universally consistent (Proposition 4.2), the existence of a solution to the target linear system system cannot be guaranteed solely from the assumption of linearly independent features (Condition 4.3). Consequently, these iterative algorithms such as TD, FQI, and PFQI that are designed to solve the target linear system does not necessarily have fixed point just under the assumption of linearly independent features.

**Lemma C.1.** *These following conditions are equivalent to rank invariance (Condition 4.1):*

$$\mathrm{Rank}\,(\Sigma_{cov}) = \mathrm{Rank}\,(\Sigma_{cov} - \gamma\Sigma_{cr}) \tag{92}$$

$$\mathrm{Col}\,(\Sigma_{cov}) = \mathrm{Col}\,(\Sigma_{cov} - \gamma\Sigma_{cr}) \tag{93}$$

$$\mathrm{Ker}\,(\Sigma_{cov}) = \mathrm{Ker}\,(\Sigma_{cov} - \gamma\Sigma_{cr}) \tag{94}$$

$$\mathrm{Col}\,(\Phi^{\top}) = \mathrm{Col}\,(\Sigma_{cov} - \gamma\Sigma_{cr}) \tag{95}$$

$$\mathrm{Ker}\,(\Phi) = \mathrm{Ker}\,(\Sigma_{cov} - \gamma\Sigma_{cr}) \tag{96}$$

$$\mathrm{Rank}\,(\Phi) = \mathrm{Rank}\,(\Sigma_{cov} - \gamma\Sigma_{cr}) \tag{97}$$

$$\mathrm{Ker}\,(\Phi^{\top}) \cap \mathrm{Col}\,(\mathbf{D}(I - \gamma\mathbf{P}_{\pi})\Phi) = \{0\} \tag{98}$$

$$\mathrm{Ker}\,(\Phi^{\top}\mathbf{D}^{\frac{1}{2}}) \cap \mathrm{Col}\,(\mathbf{D}^{\frac{1}{2}}(I - \gamma\mathbf{P}_{\pi})\Phi) = \{0\} \tag{99}$$

$$\mathrm{Ker}\,(\Phi^{\top}\mathbf{D}(I - \gamma\mathbf{P}_{\pi})) \cap \mathrm{Col}\,(\Phi) = \{0\} \tag{100}$$

*Proof.* From Lemma B.5, we know that

$$\text{Col}\left(\Phi^\top \mathbf{D}\Phi\right) = \text{Col}\left(\Phi^\top\right) \supseteq \text{Col}\left(\Phi^\top \mathbf{D}(I - \gamma \mathbf{P}_\pi)\Phi\right)$$

and

$$\text{Ker}\left(\Phi^\top \mathbf{D}\Phi\right) = \text{Ker}\left(\Phi\right) \subseteq \text{Ker}\left(\Phi^\top \mathbf{D}(I - \gamma \mathbf{P}_\pi)\Phi\right)$$

Therefore,

$$\text{Rank}\left(\Phi\right) = \text{Rank}\left(\Phi^\top \mathbf{D}(I - \gamma \mathbf{P}_\pi)\Phi\right)$$

if and only if the following hold

$$\text{Col}\left(\Phi^\top \mathbf{D}\Phi\right) = \text{Col}\left(\Phi^\top\right) = \text{Col}\left(\Phi^\top \mathbf{D}(I - \gamma \mathbf{P}_\pi)\Phi\right)$$

and

$$\text{Ker}\left(\Phi^\top \mathbf{D}\Phi\right) = \text{Ker}\left(\Phi\right) = \text{Ker}\left(\Phi^\top \mathbf{D}(I - \gamma \mathbf{P}_\pi)\Phi\right)$$

Hence, Equations (93) to (97) are equivalent. Subsequently, together with

$$\text{Rank}\left(\Phi^\top \mathbf{D}\Phi\right) = \text{Rank}\left(\Phi\right)$$

we can obtain that Equation (92) is equivalent to Equation (97).

Next, since $\mathbf{D}(I - \gamma \mathbf{P}_\pi)$ is nonsingular matrix,

$$\text{Rank}\left(\Phi^\top\right) = \text{Rank}\left(\mathbf{D}(I - \gamma \mathbf{P}_\pi)\Phi\right)$$

and

$$\text{Rank}\left(\mathbf{D}^{\frac{1}{2}}\Phi\right) = \text{Rank}\left(\mathbf{D}^{\frac{1}{2}}(I - \gamma \mathbf{P}_\pi)\Phi\right)$$

and

$$\text{Rank}\left(\Phi^\top \mathbf{D}(I - \gamma \mathbf{P}_\pi)\right) = \text{Rank}\left(\Phi\right)$$

Consequently, by Lemma B.7 we know that $\text{Rank}\left(\Phi\right) = \text{Rank}\left(\Phi^\top \mathbf{D}(I - \gamma \mathbf{P}_\pi)\Phi\right)$ if and only if

$$\text{Ker}\left(\Phi^\top\right) \cap \text{Col}\left(\mathbf{D}(I - \gamma \mathbf{P}_\pi)\Phi\right) = \{0\}$$

or

$$\text{Ker}\left(\Phi^\top \mathbf{D}^{\frac{1}{2}}\right) \cap \text{Col}\left(\mathbf{D}^{\frac{1}{2}}(I - \gamma \mathbf{P}_\pi)\Phi\right) = \{0\}$$

or

$$\text{Ker}\left(\Phi^\top \mathbf{D}(I - \gamma \mathbf{P}_\pi)\right) \cap \text{Col}\left(\Phi\right) = \{0\}$$

So Equations (97) to (100) are equivalent. Hence the proof is complete. $\square$

## C.2  Rank Invariance is a mild condition and should widely exist

From Lemma C.2, we can see that the condition of $\gamma \Sigma_{cov}^\dagger \Sigma_{cr}$ having no eigenvalue equal to 1 is equivalent to rank invariance (Condition 4.1) holding. Even if $\gamma \Sigma_{cov}^\dagger \Sigma_{cr}$ has an eigenvalue equal to 1, by slightly changing the value of $\gamma$, we can ensure that $\gamma \Sigma_{cov}^\dagger \Sigma_{cr}$ no longer has 1 as an eigenvalue. In such a case, rank invariance (Condition 4.1) will hold. Therefore, we can conclude that rank invariance (Condition 4.1) can be easily achieved and should widely exist.

**Lemma C.2.** $\left(\gamma \Sigma_{cov}^\dagger \Sigma_{cr}\right)$ *have no eigenvalue equal to 1 if and only if rank invariance (Condition 4.1) holds.*

*Proof.* Assuming rank invariance (Condition 4.1) does not holds, by Lemma C.1 we know that

$$\text{Ker}\left(\Sigma_{cov} - \gamma \Sigma_{cr}\right) \neq \text{Ker}\left(\Sigma_{cov}\right)$$

then together with Lemma B.5 we know

$$\text{Ker}\left(\Sigma_{cov} - \gamma \Sigma_{cr}\right) \supset \text{Ker}\left(\Sigma_{cov}\right)$$

so

$$\text{Ker}\left(\Sigma_{cov} - \gamma \Sigma_{cr}\right) \cap \text{Row}\left(\Sigma_{cov}\right) \neq \{0\}$$

Moreover, since $\Sigma_{cov}$ is symmetric matrix, we know that

$$\text{Ker}\left(\Sigma_{cov} - \gamma \Sigma_{cr}\right) \cap \text{Col}\left(\Sigma_{cov}\right) \neq \{0\}.$$

Therefore, for a nonzero vector $v \in \mathrm{Ker}\left(\Sigma_{cov} - \gamma\Sigma_{cr}\right) \cap \mathrm{Col}\left(\Sigma_{cov}\right)$, we have:

$$\left(\Sigma_{cov} - \gamma\Sigma_{cr}\right)v = 0$$

which is equal to $\Sigma_{cov}v = \gamma\Sigma_{cr}v$. By multiplying $\Sigma_{cov}^{\dagger}$ on both sides of equation we can get:

$$\gamma\Sigma_{cov}^{\dagger}\Sigma_{cr}v = \Sigma_{cov}^{\dagger}\Sigma_{cov}v. \tag{101}$$

Since $\left(\Sigma_{cov}^{\dagger}\Sigma_{cov}\right)$ is orthogonal projector onto $\mathrm{Col}\left(\Sigma_{cov}^{\dagger}\right)$ and by Lemma C.3 we know

$$\mathrm{Col}\left(\Sigma_{cov}^{\dagger}\right) = \mathrm{Col}\left(\Sigma_{cov}^{\top}\right)$$

Additionally, $\Sigma_{cov}$ is symmetric, so $\mathrm{Col}\left(\Sigma_{cov}^{\top}\right) = \mathrm{Col}\left(\Sigma_{cov}\right)$, then since $v \in \mathrm{Col}\left(\Sigma_{cov}\right)$ we can obtain that $\Sigma_{cov}^{\dagger}\Sigma_{cov}v = v$, therefore, we have:

$$\gamma\Sigma_{cov}^{\dagger}\Sigma_{cr}v = v,$$

which means $v$ is eigenvector of $\left(\gamma\Sigma_{cov}^{\dagger}\Sigma_{cr}\right)$ and whose corresponding eigenvalue is 1. We can conclude that when rank invariance (Condition 4.1) does not hold, matrix $\left(\gamma\Sigma_{cov}^{\dagger}\Sigma_{cr}\right)$ must have eigenvalue equal to 1.

Next, assuming $\left(\gamma\Sigma_{cov}^{\dagger}\Sigma_{cr}\right)$ has eigenvalue equal to 1, then there exist a nonzero vector $v$ that

$$\gamma\Sigma_{cov}^{\dagger}\Sigma_{cr}v = v$$

and

$$v \in \mathrm{Col}\left(\Sigma_{cov}^{\dagger}\right)$$

By Lemma C.3 we know $\mathrm{Col}\left(\Sigma_{cov}^{\dagger}\right) = \mathrm{Col}\left(\Sigma_{cov}^{\top}\right)$ and $\Sigma_{cov}$ is symmetric so

$$v \in \mathrm{Col}\left(\Sigma_{cov}^{\top}\right) = \mathrm{Col}\left(\Sigma_{cov}\right)$$

Furthermore, by Lemma B.5, we know that $\mathrm{Col}\left(\Phi^{\top}\mathbf{D}\Phi\right) = \mathrm{Col}\left(\Phi^{\top}\right) \supseteq \mathrm{Col}\left(\Phi^{\top}\mathbf{D}\mathbf{P}_{\pi}\Phi\right)$, which implies $\mathrm{Col}\left(\Sigma_{cov}\right) \supseteq \mathrm{Col}\left(\Sigma_{cr}\right)$. Therefore multiplying by $\Sigma_{cov}$ on both sides of $\gamma\Sigma_{cov}^{\dagger}\Sigma_{cr}v = v$, we get

$$\gamma\Sigma_{cov}\Sigma_{cov}^{\dagger}\Sigma_{cr}v = \Sigma_{cov}v$$

which is equal to

$$\left(\Sigma_{cov} - \gamma\Sigma_{cr}\right)v = 0$$

Thus, $v \in \mathrm{Ker}\left(\Sigma_{cov} - \gamma\Sigma_{cr}\right)$, implying that

$$\mathrm{Col}\left(\Sigma_{cov}\right) \cap \mathrm{Ker}\left(\Sigma_{cov} - \gamma\Sigma_{cr}\right) \neq \emptyset$$

Since $\mathrm{Col}\left(\Sigma_{cov}\right) = \mathrm{Row}\left(\Sigma_{cov}\right)$, we conclude that

$$\mathrm{Ker}\left(\Sigma_{cov}\right) \neq \mathrm{Ker}\left(\Sigma_{cov} - \gamma\Sigma_{cr}\right)$$

By Lemma C.1, this shows that rank invariance (Condition 4.1) does not hold.

Hence the proof is complete. $\qquad\square$

**Lemma C.3.** *Given a matrix $A \in \mathbb{R}^{n \times m}$, $\mathrm{Col}\left(A^{\top}\right) = \mathrm{Col}\left(A^{\dagger}\right)$*

*Proof.* Since $AA^{\dagger}$ is orthogonal projector onto $\mathrm{Col}(A)$ and $A^{\dagger}A$ is orthogonal projector onto $\mathrm{Col}\left(A^{\dagger}\right)$, by Meyer [26, Page 386, 5.9.11], we know that

$$\mathrm{Col}\left(AA^{\dagger}\right) = \mathrm{Col}(A) \text{ and } \mathrm{Col}\left(A^{\dagger}A\right) = \mathrm{Col}\left(A^{\dagger}\right)$$

therefore, we have:

$$\mathrm{Col}\left(A^{\top}\right) = \mathrm{Col}\left(A^{\top}(A^{\top})^{\dagger}\right) = \mathrm{Col}\left(((A^{\top})^{\dagger})^{\top}A\right) = \mathrm{Col}\left(A^{\dagger}A\right) = \mathrm{Col}\left(A^{\dagger}\right) \tag{102}$$

$$\qquad\square$$

## C.3 Consistency of the target linear system

### C.3.1 Proof of Proposition 4.2

**Proposition C.4** (Restatement of Proposition 4.2). *The target linear system:*

$$\left(\Phi^\top \mathbf{D}(I - \gamma \mathbf{P}_\pi)\Phi\right)\theta = \Phi^\top \mathbf{D}R$$

*is consistent for any $R \in \mathbb{R}^h$ if and only if*

$$\mathrm{Rank}\left(\Sigma_{cov}\right) = \mathrm{Rank}\left(\Sigma_{cov} - \gamma\Sigma_{cr}\right)$$

.

*Proof.* For any $R \in \mathbb{R}^h$, the target linear system:

$$\left(\Phi^\top \mathbf{D}(I - \gamma \mathbf{P}_\pi)\Phi\right)\theta = \Phi^\top \mathbf{D}R$$

is consistent if and only if for any $R \in \mathbb{R}^h$,

$$\left(\Phi^\top \mathbf{D}R\right) \in \mathrm{Col}\left(\Phi^\top \mathbf{D}(I - \gamma \mathbf{P}_\pi)\Phi\right),$$

which is equivalent to

$$\mathrm{Col}\left(\Phi^\top \mathbf{D}\right) \subseteq \mathrm{Col}\left(\Phi^\top \mathbf{D}(I - \gamma \mathbf{P}_\pi)\Phi\right).$$

From Lemma B.5, we know that

$$\mathrm{Col}\left(\Phi^\top \mathbf{D}\right) = \mathrm{Col}\left(\Phi^\top\right) \supseteq \mathrm{Col}\left(\Phi^\top \mathbf{D}(I - \gamma \mathbf{P}_\pi)\Phi\right).$$

Therefore, $\mathrm{Col}\left(\Phi^\top \mathbf{D}\right) \subseteq \mathrm{Col}\left(\Phi^\top \mathbf{D}(I - \gamma \mathbf{P}_\pi)\Phi\right)$ holds if and only if

$$\mathrm{Col}\left(\Phi^\top \mathbf{D}\right) = \mathrm{Col}\left(\Phi^\top \mathbf{D}(I - \gamma \mathbf{P}_\pi)\Phi\right).$$

Since $\mathrm{Col}\left(\Phi^\top \mathbf{D}\right) = \mathrm{Col}\left(\Phi^\top\right)$, by Lemma C.1 we know that

$$\mathrm{Col}\left(\Phi^\top \mathbf{D}\right) = \mathrm{Col}\left(\Phi^\top \mathbf{D}(I - \gamma \mathbf{P}_\pi)\Phi\right)$$

holds if and only if $\mathrm{Rank}\left(\Sigma_{cov}\right) = \mathrm{Rank}\left(\Sigma_{cov} - \gamma\Sigma_{cr}\right)$.

Hence, the target linear system

$$\left(\Phi^\top \mathbf{D}(I - \gamma \mathbf{P}_\pi)\Phi\right)\theta = \Phi^\top \mathbf{D}R$$

is consistent for any $R \in \mathbb{R}^h$ if and only if $\mathrm{Rank}\left(\Sigma_{cov}\right) = \mathrm{Rank}\left(\Sigma_{cov} - \gamma\Sigma_{cr}\right)$. $\qquad \square$

## C.4 Nonsingularity the target linear system

### C.4.1 Proof of Proposition 4.5

**Proposition C.5** (Restatement of Proposition 4.5). *$(\Sigma_{cov} - \gamma\Sigma_{cr})$ is nonsingular if and only if*

$$\Phi \text{ is full column rank and } \mathrm{Rank}\left(\Sigma_{cov}\right) = \mathrm{Rank}\left(\Sigma_{cov} - \gamma\Sigma_{cr}\right)$$

.

*Proof.* If $\Phi^\top \mathbf{D}(I - \gamma \mathbf{P}_\pi)\Phi$ is full rank, by Fact C.6, it is clear that $\Phi$ must be full column rank.

Next, assuming $\Phi$ is full column rank, we know that $\mathrm{Rank}\left(\Phi^\top \mathbf{D}(I - \gamma \mathbf{P}_\pi)\Phi\right)$ is full rank if and only if

$$\mathrm{Rank}\left(\Phi^\top \mathbf{D}(I - \gamma \mathbf{P}_\pi)\Phi\right) = \mathrm{Rank}\left(\Phi\right).$$

Also, by Lemma B.5 we know that

$$\mathrm{Rank}\left(\Sigma_{cov}\right) = \mathrm{Rank}\left(\Phi\right).$$

Therefore, $\mathrm{Rank}\left(\Phi^\top \mathbf{D}(I - \gamma \mathbf{P}_\pi)\Phi\right)$ is full rank if and only if

$$\mathrm{Rank}\left(\Sigma_{cov}\right) = \mathrm{Rank}\left(\Sigma_{cov} - \gamma\Sigma_{cr}\right).$$

and $\Phi$ is full column rank. $\qquad \square$

**Fact C.6.** Let $A$ be a $K \times L$ matrix and $B$ an $L \times M$ matrix. Then,

$$\mathrm{rank}(AB) \leq \min(\mathrm{rank}(A), \mathrm{rank}(B)).$$

## C.5 Nonsingularity of the FQI linear system

**Proposition C.7** (Restatement of Proposition 4.6). *$I - \gamma \Sigma_{cov}^{\dagger} \Sigma_{cr}$ is nonsingular if and only if rank invariance (Condition 4.1) holds*

*Proof.* By Lemma C.2, we know that rank invariance (Condition 4.1) holds if and only if $\gamma \Sigma_{cov}^{\dagger} \Sigma_{cr}$ has no eigenvalue equal to 1. Consequently, by Lemma A.1, this is equivalent to $I - \gamma \Sigma_{cov}^{\dagger} \Sigma_{cr}$ having no eigenvalue equal to 0, which in turn it is equivalent to $I - \gamma \Sigma_{cov}^{\dagger} \Sigma_{cr}$ being nonsingular. $\qquad \square$

## C.6 On-policy setting

### C.6.1 Proof of Proposition 4.7

**Proposition C.8** (Restatement of Proposition 4.7). *In the on-policy setting,*

$$\mathrm{Rank}\left(\Sigma_{cov}\right) = \mathrm{Rank}\left(\Sigma_{cov} - \gamma \Sigma_{cr}\right).$$

*Proof.* In the on-policy setting, from Tsitsiklis and Van Roy [40] we know that $\mathbf{D}(I - \gamma \mathbf{P}_\pi)$ is a positive definite matrix, then by Lemma A.16, we know that

$$\mathrm{Ker}\left(\Phi^\top \mathbf{D}(I - \gamma \mathbf{P}_\pi)\Phi\right) = \mathrm{Ker}\left(\Phi\right).$$

Therefore, by Lemma C.1 we know that

$$\mathrm{Rank}\left(\Sigma_{cov}\right) = \mathrm{Rank}\left(\Sigma_{cov} - \gamma \Sigma_{cr}\right).$$

$\qquad \square$

## C.7 Linear realizability

### C.7.1 Proof of Proposition 4.9

**Proposition C.9** (Restatement of Proposition 4.9). *When linear realizability holds (Assumption 4.8),*

- *$\Theta_{LSTD} \supseteq \Theta_\pi$ always holds*
- *$\Theta_{LSTD} = \Theta_\pi$ holds if and only if rank invariance (Condition 4.1) holds.*

*Proof.* Since $\Theta_{\mathrm{LSTD}}$ is the solution set of the target linear system:

$$\left(\Phi^\top \mathbf{D}(I - \gamma \mathbf{P}_\pi)\Phi\right)\Phi\theta = \Phi^\top \mathbf{D}R$$

and $\Theta_\pi$ is equal to the solution set of linear system:

$$(I - \gamma \mathbf{P}_\pi)\theta = R,$$

we know that

$$\Theta_{\mathrm{LSTD}} \supseteq \Theta_\pi.$$

Then, by Lemma C.10 we know that $\Theta_{\mathrm{LSTD}} = \Theta_\pi$ holds if and only if

$$\mathrm{Rank}\left(\Phi^\top \mathbf{D}(I - \gamma \mathbf{P}_\pi)\Phi\right) = \mathrm{Rank}\left((I - \gamma \mathbf{P}_\pi)\Phi\right),$$

and since $(I - \gamma \mathbf{P}_\pi)$ is full rank matrix and $\mathrm{Rank}\left(\Phi^\top \mathbf{D}\Phi\right) = \mathrm{Rank}\left(\Phi\right)$, from Lemma B.5, we know that

$$\mathrm{Rank}\left((I - \gamma \mathbf{P}_\pi)\Phi\right) = \mathrm{Rank}\left(\Phi\right).$$

Therefore, we know that $\Theta_{\mathrm{LSTD}} = \Theta_\pi$ holds if and only if

$$\mathrm{Rank}\left(\Phi^\top \mathbf{D}(I - \gamma \mathbf{P}_\pi)\Phi\right) = \mathrm{Rank}\left(\Phi^\top \mathbf{D}\Phi\right),$$

which is $\mathrm{Rank}\left(\Sigma_{cov}\right) = \mathrm{Rank}\left(\Sigma_{cov} - \gamma \Sigma_{cr}\right).$ $\qquad \square$

**Lemma C.10.** *Given two matrices $A \in \mathbb{R}^{n \times m}$ and $B \in \mathbb{R}^{p \times n}$, and a vector $b \in \mathrm{Col}(A)$, we denote the $\mathcal{S}_A$ the solution set for linear system: $Ax = b$ and $\mathcal{S}_{BA}$ the solution set for linear system: $BAx = Bb$. the following holds:*

$$\mathcal{S}_A \supseteq \mathcal{S}_{BA} \text{ only holds when } \mathcal{S}_A = \mathcal{S}_{BA} \tag{103}$$

*and*

$$\mathcal{S}_A = \mathcal{S}_{BA} \text{ if and only if } \mathrm{Rank}(BA) = \mathrm{Rank}(A). \tag{104}$$

*Proof.* It is clear that any $x$ satisfying $Ax = b$ also satisfies $BAx = Bb$, so $\mathcal{S}_A \subseteq \mathcal{S}_{BA}$. Therefore, if $\mathcal{S}_A \supseteq \mathcal{S}_{BA}$, $\mathcal{S}_A = \mathcal{S}_{BA}$.

Next, as $b \in \mathrm{Col}(A)$ we know that

$$\mathcal{S}_A = \{A^\dagger b + (I - A^\dagger A)v \mid \forall v \in \mathbb{R}^m)\},$$

where $\{(I - A^\dagger A)v \mid \forall v \in \mathbb{R}^m)\} = \mathrm{Ker}(A)$ and $(A^\dagger b) \notin \mathrm{Ker}(A)$. Also,

$$\mathcal{S}_{BA} = \{(BA)^\dagger Bb + (I - (BA)^\dagger BA)w \mid \forall w \in \mathbb{R}^m)\},$$

where $\{(I - (BA)^\dagger BA)w \mid \forall w \in \mathbb{R}^m)\} = \mathrm{Ker}(BA)$ and $((BA)^\dagger Bb) \notin \mathrm{Ker}(BA)$. Additionally,

$$\mathrm{Ker}(A) \subseteq \mathrm{Ker}(BA).$$

First, we will prove that if $\mathcal{S}_A = \mathcal{S}_{BA}$, then $\mathrm{Rank}(A) = \mathrm{Rank}(BA)$.

Since $(A^\dagger b) \notin \mathrm{Ker}(A)$ and $((BA)^\dagger Bb) \notin \mathrm{Ker}(BA)$, from above we know if $\mathcal{S}_A = \mathcal{S}_{BA}$,

$$\dim(\mathrm{Ker}(A)) = \dim(\mathrm{Ker}(BA)),$$

which is equivalent to $\mathrm{Ker}(A) = \mathrm{Ker}(BA)$ since $\mathrm{Ker}(A) \subseteq \mathrm{Ker}(BA)$. From that we can get that $\mathrm{Rank}(A) = \mathrm{Rank}(BA)$ by the Rank-Nullity Theorem.

Now we need to prove that if $\mathrm{Rank}(A) = \mathrm{Rank}(BA)$, then $\mathcal{S}_A = \mathcal{S}_{BA}$. We know that

$$\mathrm{Ker}(A) \subseteq \mathrm{Ker}(BA),$$

so when $\mathrm{Rank}(A) = \mathrm{Rank}(BA)$, $\mathrm{Ker}(A) = \mathrm{Ker}(BA)$ and

$$\{(I - A^\dagger A)v \mid \forall v \in \mathbb{R}^m)\} = \{(I - (BA)^\dagger BA)w \mid \forall w \in \mathbb{R}^m)\}.$$

Also, we have that:

$$A^\dagger b - (BA)^\dagger Bb = \left(A^\dagger - (BA)^\dagger B\right) b \tag{105}$$

$$= \left(I - (BA)^\dagger BA\right) A^\dagger b \tag{106}$$

$$\in \{(I - (BA)^\dagger BA)w \mid \forall w \in \mathbb{R}^m)\} = \mathrm{Ker}(BA) = \mathrm{Ker}(A). \tag{107}$$

Therefore,

$$A^\dagger b \in \{(BA)^\dagger Bb + (I - (BA)^\dagger BA)w \mid \forall w \in \mathbb{R}^m)\},$$

and

$$((BA)^\dagger Bb) \in \{A^\dagger b - (I - A^\dagger A)v \mid \forall v \in \mathbb{R}^m)\},$$

which is equal to

$$((BA)^\dagger Bb) \in \{A^\dagger b + (I - A^\dagger A)v \mid \forall v \in \mathbb{R}^m)\}.$$

Then, we know that

$$\{(BA)^\dagger Bb + (I - (BA)^\dagger BA)w \mid \forall w \in \mathbb{R}^m)\} = \{A^\dagger b + (I - A^\dagger A)v \mid \forall v \in \mathbb{R}^m)\}.$$

Hence we can conclude that if $\mathrm{Rank}(A) = \mathrm{Rank}(BA)$, $\mathcal{S}_{BA} = \mathcal{S}_A$.

$\square$

**Fact C.11.** If $X_{t+1} = AX_t + B$, then if update starts from $X_0$, we have:

$$X_{t+1} = \sum_{i=0}^{t} A^i B + A^{t+1} X_0$$

# D  The convergence of FQI

## D.1  Interpretation of convergence condition and fixed point for FQI

First, Theorem 5.1 provides a general necessary and sufficient condition for the convergence of FQI without imposing any additional assumptions, such as $\Phi$ being full rank. Later, we will demonstrate how the convergence conditions vary under different assumptions.

**Theorem D.1** (Restatement of Theorem 5.1). *FQI converges for any initial point $\theta_0$ if and only if $\left(\Sigma_{cov}^{\dagger}\theta_{\phi,r}\right) \in \text{Col}\left(I - \gamma\Sigma_{cov}^{\dagger}\Sigma_{cr}\right)$ and $\left(\gamma\Sigma_{cov}^{\dagger}\Sigma_{cr}\right)$ is semiconvergent. It converges to*

$$\left[\left(I - \gamma\Sigma_{cov}^{\dagger}\Sigma_{cr}\right)^{\text{D}}\Sigma_{cov}^{\dagger}\theta_{\phi,r} + \left(I - (I - \gamma\Sigma_{cov}^{\dagger}\Sigma_{cr})\left(I - \gamma\Sigma_{cov}^{\dagger}\Sigma_{cr}\right)^{\text{D}}\right)\theta_0\right] \in \Theta_{LSTD}.$$

As previously defined in Section 3, we have $b_{\text{FQI}} = \Sigma_{cov}^{\dagger}\theta_{\phi,r}$, $A_{\text{FQI}} = I - \gamma\Sigma_{cov}^{\dagger}\Sigma_{cr}$, and $H_{\text{FQI}} = \gamma\Sigma_{cov}^{\dagger}\Sigma_{cr}$. From Theorem 5.1, we can see that the necessary and sufficient condition for FQI convergence consists of two conditions:

$$(b_{\text{FQI}}) \in \text{Col}\left(A_{\text{FQI}}\right) \text{ and } H_{\text{FQI}} \text{ being semiconvergent.}$$

First, $(b_{\text{FQI}}) \in \text{Col}\left(A_{\text{FQI}}\right)$ ensures that the FQI linear system is consistent, which means that a fixed point for FQI exists. Second, $H_{\text{FQI}}$ being semiconvergent implies that $H_{\text{FQI}}$ converges on $\overline{\text{Ker}}\left(A_{\text{FQI}}\right)$, and acts as an identity matrix on $\text{Ker}\left(A_{\text{FQI}}\right)$ if $\text{Ker}\left(A_{\text{FQI}}\right) \neq \{0\}$. Since any vector can be decomposed into two components — one from $\text{Ker}\left(A_{\text{FQI}}\right)$ and one from $\overline{\text{Ker}}\left(A_{\text{FQI}}\right)$ — the above condition ensures that iterations converge to a fixed point for the component in $\overline{\text{Ker}}\left(A_{\text{FQI}}\right)$, while maintaining stability for the component in $\text{Ker}\left(A_{\text{FQI}}\right)$ without amplification. This stability is crucial because $H_{\text{FQI}} = I - A_{\text{FQI}}$, and if $\text{Ker}\left(A_{\text{FQI}}\right) \neq \{0\}$, then $H_{\text{FQI}}$ necessarily has an eigenvalue equal to 1, whose associated component can easily diverge within $\text{Ker}\left(A_{\text{FQI}}\right)$. Consequently, preventing amplification of $H_{\text{FQI}}$ in $\text{Ker}\left(A_{\text{FQI}}\right)$ during iterations is essential.

The fixed point to which FQI converges consists of two components:

$$\left(A_{\text{FQI}}\right)^{\text{D}}b_{\text{FQI}} \quad \text{and} \quad \left(I - A_{\text{FQI}}\left(A_{\text{FQI}}\right)^{\text{D}}\right)\theta_0. \tag{108}$$

The term $\left(I - (A_{\text{FQI}})\left(A_{\text{FQI}}\right)^{\text{D}}\right)\theta_0$ represents any vector from

$$\text{Ker}\left(A_{\text{FQI}}\right)$$

because $\left[(A_{\text{FQI}})\left(A_{\text{FQI}}\right)^{\text{D}}\right]$ is a projector onto

$$\text{Col}\left((A_{\text{FQI}})^{k}\right) \text{ along } \text{Ker}\left((A_{\text{FQI}})^{k}\right),$$

while $\left(I - (A_{\text{FQI}})\left(A_{\text{FQI}}\right)^{\text{D}}\right)$ is the complementary projector onto

$$\text{Ker}\left((A_{\text{FQI}})^{k}\right) \text{ along } col(A_{\text{FQI}})^{k},$$

where $k = \textbf{Index}\left(A_{\text{FQI}}\right)$. Consequently,

$$\text{Col}\left(I - (A_{\text{FQI}})\left(A_{\text{FQI}}\right)^{\text{D}}\right) = \text{Ker}\left((A_{\text{FQI}})^{k}\right).$$

Since $H_{\text{FQI}}$ is semiconvergent, $\textbf{Index}\left(I - H_{\text{FQI}}\right) \leq 1$ and $A_{\text{FQI}} = I - H_{\text{FQI}}$, we know that

$$\text{Col}\left(I - (A_{\text{FQI}})\left(A_{\text{FQI}}\right)^{\text{D}}\right) = \text{Ker}\left(A_{\text{FQI}}\right)$$

Therefore, $\left(I - (A_{\text{FQI}})\left(A_{\text{FQI}}\right)^{\text{D}}\right)\theta_0$ can be any vector in $\text{Ker}\left(A_{\text{FQI}}\right)$. Additionally, for the term $(A_{\text{FQI}})^{\text{D}}b_{\text{FQI}}$ in Equation (108), since $\textbf{Index}\left(A_{\text{FQI}}\right) \leq 1$, it follows that

$$(A_{\text{FQI}})^{\text{D}}b_{\text{FQI}} = (A_{\text{FQI}})^{\#}b_{\text{FQI}}.$$

In summary, we conclude that any fixed point to which FQI converges is the sum of the group inverse solution of the FQI linear system, denoted by $(A_{\text{FQI}})^{\#} b_{\text{FQI}}$, and a vector from the null space of $A_{\text{FQI}}$, i.e., $\text{Ker}\,(A_{\text{FQI}})$. Additionally, since $\Sigma_{cov} A_{\text{FQI}} = A_{\text{LSTD}}$ and $\Sigma_{cov} b_{\text{FQI}} = b_{\text{LSTD}}$ (Section 3) and the FQI linear system is consistent, i.e., $(b_{\text{FQI}}) \in \text{Col}\,(A_{\text{FQI}})$, it follows that $(A_{\text{FQI}})^{\#} b_{\text{FQI}}$ is also a solution to target linear system[14]. Moreover, as $\text{Ker}\,(A_{\text{FQI}}) \subseteq \text{Ker}\,(A_{\text{LSTD}})$, the sum of $(A_{\text{FQI}})^{\#} b_{\text{FQI}}$ and any vector from $\text{Ker}\,(A_{\text{FQI}})$ is also a solution to target linear system. In other words, any fixed point to which FQI converges is also a solution to the target linear system. This conclusion aligns with the results presented in Section 3, where it is shown that target linear system represents the projected version of the FQI linear system.

## D.2  Proof of Theorem 5.1

**Theorem D.2** (Restatement of Theorem 5.1). *FQI converges for any initial point $\theta_0$ if and only if*

$$\left(\Sigma_{cov}^{\dagger} \theta_{\phi,r}\right) \in \text{Col}\left(I - \gamma \Sigma_{cov}^{\dagger} \Sigma_{cr}\right) \ \textit{and} \ \left(\gamma \Sigma_{cov}^{\dagger} \Sigma_{cr}\right) \ \textit{is semiconvergent.}$$

*It converges to* $\left[\left(I - \gamma \Sigma_{cov}^{\dagger} \Sigma_{cr}\right)^{\text{D}} \Sigma_{cov}^{\dagger} \theta_{\phi,r} + \left(I - (I - \gamma \Sigma_{cov}^{\dagger} \Sigma_{cr})\left(I - \gamma \Sigma_{cov}^{\dagger} \Sigma_{cr}\right)^{\text{D}}\right) \theta_0\right].$

*Proof.* From Section 3 we know that FQI is fundamentally a iterative method to solve the FQI linear system

$$(I - \gamma \Sigma_{cov}^{\dagger} \Sigma_{cr})\theta = \Sigma_{cov}^{\dagger} \theta_{\phi,r}.$$

Therefore, without assuming singularity of the linear system, by Berman and Plemmons [5, Pages 198, lemma 6.13][15], we know that this iterative method converges if and only if the FQI linear system is consistent:

$$\left(\Sigma_{cov}^{\dagger} \theta_{\phi,r}\right) \in \text{Col}\left(I - \gamma \Sigma_{cov}^{\dagger} \Sigma_{cr}\right),$$

and $\gamma \Sigma_{cov}^{\dagger} \Sigma_{cr}$ is semiconvergent. It converges to

$$\left[\left[\left(I - \gamma \Sigma_{cov}^{\dagger} \Sigma_{cr}\right)^{\text{D}} \Sigma_{cov}^{\dagger} \theta_{\phi,r} + \left(I - (I - \gamma \Sigma_{cov}^{\dagger} \Sigma_{cr})\left(I - \gamma \Sigma_{cov}^{\dagger} \Sigma_{cr}\right)^{\text{D}}\right) \theta_0\right]\right] \in \Theta_{\text{LSTD}}.$$

$\square$

## D.3  Linearly independent features

Proposition D.3 examines how linearly independent features affect the convergence of FQI. As shown in Section 3, when $\Phi$ is full rank (linearly independent features (Condition 4.3)), the FQI linear system that FQI solves is exactly equal to the target linear system. Consequently, the consistency condition changes from $(b_{\text{FQI}}) \in \text{Col}\,(A_{\text{FQI}})$ to $b_{\text{LSTD}} \in \text{Col}\,(A_{\text{LSTD}})$, and the covariance matrix $\Sigma_{cov}$ becomes invertible. FQI can then be viewed as an iterative method using $\Sigma_{cov}^{-1}$ as a preconditioner to solve target linear system, with $M_{\text{FQI}} = \Sigma_{cov}^{-1}$ and $H_{\text{FQI}} = I - M_{\text{FQI}} A_{\text{LSTD}}$. Beyond these adjustments, the convergence conditions for FQI remain largely unchanged compared to the general convergence conditions for FQI (Theorem 5.1), which does not make the linearly independent features assumption. Thus, we conclude that the linearly independent features assumption does not play a crucial role in FQI's convergence but instead determines the specific linear system that FQI is iteratively solving.

**Proposition D.3.** *Given $\Phi$ is full column rank(Condition 4.3 holds), FQI converges for any initial point $\theta_0$ if and only if*

$$\theta_{\phi,r} \in \text{Col}\left(\Sigma_{cov} - \gamma \Sigma_{cr}\right)$$

*and $\left(\gamma \Sigma_{cov}^{-1} \Sigma_{cr}\right)$ is semiconvergent. it converges to*

$$\left[\left(I - \gamma \Sigma_{cov}^{-1} \Sigma_{cr}\right)^{\text{D}} \Sigma_{cov}^{-1} \theta_{\phi,r} + \left(I - (I - \gamma \Sigma_{cov}^{-1} \Sigma_{cr})\left(I - \gamma \Sigma_{cov}^{-1} \Sigma_{cr}\right)^{\text{D}}\right) \theta_0\right] \in \Theta_{LSTD}.$$

---

[14]Brief proof: $A_{\text{LSTD}} (A_{\text{FQI}})^{\#} b_{\text{FQI}} = \Sigma_{cov} A_{\text{FQI}} (A_{\text{FQI}})^{\#} b_{\text{FQI}} = \Sigma_{cov} b_{\text{FQI}} = b_{\text{LSTD}}$.

[15]We note that the first printing of this text contained an error in this theorem, by which the contribution of the initial point, $x_0$, was expressed as $(I - H)(I - H)^{D} x_0$ rather than $I - (I - H)(I - H)^{D} x_0$. This was corrected by the fourth printing.

*Proof.* From Section 3 we know that when $\Phi$ is full column rank ($\Sigma_{cov}$ is full rank), FQI is exactly iterative method to solve target linear system and the FQI linear system is equivalent to the target linear system. Therefore, the consistency condition of FQI linear system:

$$\left(\Sigma_{cov}^\dagger \theta_{\phi,r}\right) \in \mathrm{Col}\left(I - \gamma \Sigma_{cov}^\dagger \Sigma_{cr}\right)$$

is equivalent to the consistency condition of target linear system:

$$\theta_{\phi,r} \in \mathrm{Col}\left(\Sigma_{cov} - \gamma \Sigma_{cr}\right),$$

and we have $\Sigma_{cov}^\dagger = \Sigma_{cov}^{-1}$. Then, from Theorem 5.1, we know that in such a setting, FQI converges for any initial point $\theta_0$ if and only if

$$\theta_{\phi,r} \in \mathrm{Col}\left(\Sigma_{cov} - \gamma \Sigma_{cr}\right) \text{ and } \left(\gamma \Sigma_{cov}^{-1} \Sigma_{cr}\right) \text{ are semiconvergent,}$$

and it converges to

$$\left[\left(I - \gamma \Sigma_{cov}^{-1} \Sigma_{cr}\right)^{\mathrm{D}} \Sigma_{cov}^{-1} \theta_{\phi,r} + \left(I - (I - \gamma \Sigma_{cov}^{-1} \Sigma_{cr})\left(I - \gamma \Sigma_{cov}^{-1} \Sigma_{cr}\right)^{\mathrm{D}}\right) \theta_0\right] \in \Theta_{\mathrm{LSTD}}.$$

$\square$

## D.4  Rank Invariance

### D.4.1  Proof of Lemma 5.2

**Lemma D.4** (Restatement of Lemma 5.2). *If rank invariance (Condition 4.1) holds, $\Sigma_{cov}$ and $\Sigma_{cr}$ are a proper splitting of $(\Sigma_{cov} - \gamma \Sigma_{cr})$.*

*Proof.* When $\mathrm{Rank}\left(\Sigma_{cov}\right) = \mathrm{Rank}\left(\Sigma_{cov} - \gamma \Sigma_{cr}\right)$, by Lemma C.1 we know that

$$\mathrm{Col}\left(\Sigma_{cov}\right) = \mathrm{Col}\left(\Sigma_{cov} - \gamma \Sigma_{cr}\right) \text{ and } \mathrm{Ker}\left(\Sigma_{cov}\right) = \mathrm{Ker}\left(\Sigma_{cov} - \gamma \Sigma_{cr}\right).$$

Then, by definition of a proper splitting (in Appendix A.1), $\Sigma_{cov}$ and $\Sigma_{cr}$ are a proper splitting of

$$\left(\Sigma_{cov} - \gamma \Sigma_{cr}\right).$$

$\square$

### D.4.2  Proof of Corollary 5.3

**Corollary D.5** (Restatement of Corollary 5.3). *Assuming that rank invariance (Condition 4.1) holds, FQI converges for any initial point $\theta_0$ if and only if*

$$\rho\left(\gamma \Sigma_{cov}^\dagger \Sigma_{cr}\right) < 1.$$

*It converges to $\left[(I - \gamma \Sigma_{cov}^\dagger \Sigma_{cr})^{-1} \Sigma_{cov}^\dagger \theta_{\phi,r}\right] \in \Theta_{LSTD}$.*

*Proof.* From Lemma 5.2 we know when rank invariance (Condition 4.1) holds, $\Sigma_{cov}$ and $\Sigma_{cr}$ is proper splitting of $(\Sigma_{cov} - \gamma \Sigma_{cr})$. By the property of a proper splitting [4, Theorem 1], we know that $\left(I - \gamma \Sigma_{cov}^\dagger \Sigma_{cr}\right)$ is a nonsingular matrix. Then by Lemma A.1 we know that $\gamma \Sigma_{cov}^\dagger \Sigma_{cr}$ has no eigenvalue equal to 1; therefore, $\gamma \Sigma_{cov}^\dagger \Sigma_{cr}$ is semiconvergent if and only if $\rho\left(\gamma \Sigma_{cov}^\dagger \Sigma_{cr}\right) < 1$.

Morever, since FQI linear system is nonsingular, $\left(\Sigma_{cov}^\dagger \theta_{\phi,r}\right) \in \mathrm{Col}\left(I - \gamma \Sigma_{cov}^\dagger \Sigma_{cr}\right)$ naturally holds. Additionally,

$$\left(I - \gamma \Sigma_{cov}^\dagger \Sigma_{cr}\right)^{\mathrm{D}} = \left(I - \gamma \Sigma_{cov}^\dagger \Sigma_{cr}\right)^{-1}.$$

Hence, by Theorem 5.1, we know that in such a setting, FQI converges for any initial point $\theta_0$ if and only if

$$\rho\left(\gamma \Sigma_{cov}^\dagger \Sigma_{cr}\right) < 1.$$

It converges to $\left[(I - \gamma \Sigma_{cov}^\dagger \Sigma_{cr})^{-1} \Sigma_{cov}^\dagger \theta_{\phi,r}\right] \in \Theta_{\mathrm{LSTD}}$. $\square$

### D.5 Nonsingular target linear system

**Corollary D.6.** *Assuming $A_{FQI}$ is full rank, FQI converges for any initial point $\theta_0$ if and only if*

$$\rho\left(\gamma \Sigma_{cov}^{-1} \Sigma_{cr}\right) < 1$$

*It converges to*

$$\left[\left(\Sigma_{cov} - \gamma \Sigma_{cr}\right)^{-1} \theta_{\phi,r}\right] = \Theta_{LSTD}$$

*Proof.* By Proposition 4.6, we know that $A_{\mathrm{FQI}}$ is full rank if and only if rank invariance (Condition 4.1) holds, therefore, it is clear that it share the same convergence result with Corollary 5.3. □

## E   The convergence of TD

**Definition E.1.** TD is stable if there exists a step size $\alpha > 0$ such that for any initial parameter $\theta_0 \in \mathbb{R}^d$, when taking updates according to the TD update equation (Equation (8)), the sequence $\{\theta_k\}_{k=0}^{\infty}$ converges, i.e., $\lim_{k \to \infty} \theta_k$ exists.

### E.1   Interpretation of convergence condition and fixed point for TD

**Theorem E.2** (Restatement of Theorem 6.1). *TD converges for any initial point $\theta_0$ if and only if $b_{LSTD} \in \mathrm{Col}\,(A_{LSTD})$, and $H_{TD}$ is semiconvergent. It converges to*

$$\left[(A_{LSTD})^{\mathrm{D}}\, b_{LSTD} + \left(I - (A_{LSTD})(A_{LSTD})^{\mathrm{D}}\right)\theta_0\right] \in \Theta_{LSTD}. \tag{109}$$

As presented in Section 3, TD is an iterative method that uses a positive constant as a preconditioner to solve the target linear system. Its convergence depends solely on the consistency of the target linear system and the properties of $H_{\mathrm{TD}}$. In Theorem 6.1, we establish the necessary and sufficient condition for TD convergence. Using the notation defined in Section 3, where $b_{\mathrm{LSTD}} = \theta_{\phi,r}$, $A_{\mathrm{LSTD}} = (\Sigma_{cov} - \gamma \Sigma_{cr})$, and $H_{\mathrm{TD}} = (I - \alpha A_{\mathrm{LSTD}})$, we know that the necessary and sufficient conditions are composed of two conditions:

$$b_{\mathrm{LSTD}} \in \mathrm{Col}\,(A_{\mathrm{LSTD}}) \text{ and } H_{\mathrm{TD}} = (I - \alpha A_{\mathrm{LSTD}}) \text{ is semiconvergent.}$$

First, $b_{\mathrm{LSTD}} \in \mathrm{Col}\,(A_{\mathrm{LSTD}})$ is the necessary and sufficient condition for target linear system being consistent, meaning that a fixed point of TD exists. Second, $H_{\mathrm{TD}}$ being semiconvergent implies that $H_{\mathrm{TD}}$ is convergent on $\overline{\mathrm{Ker}}\,(A_{\mathrm{LSTD}})$ and acts as the identity on $\mathrm{Ker}\,(A_{\mathrm{LSTD}})$ if $\mathrm{Ker}\,(A_{\mathrm{LSTD}}) \neq \{0\}$.

This means that the iterations converge a fixed point on $\overline{\mathrm{Ker}}\,(A_{\mathrm{LSTD}})$ while remaining stable on $\mathrm{Ker}\,(A_{\mathrm{LSTD}})$ without amplification. Since $H_{\mathrm{TD}} = I - \alpha A_{\mathrm{LSTD}}$, if $\mathrm{Ker}\,(A_{\mathrm{LSTD}}) \neq \{0\}$, then $H_{\mathrm{TD}}$ will necessarily have an eigenvalue equal to 1, and we want to prevent amplification of this part through iterations. From Theorem 6.1, we can also see that the fixed point to which TD converges has two components:

$$(A_{\mathrm{LSTD}})^{\mathrm{D}} b_{\mathrm{LSTD}} \text{ and } \left(I - (A_{\mathrm{LSTD}})\,(A_{\mathrm{LSTD}})^{\mathrm{D}}\right)\theta_0.$$

The term $\left(I - (A_{\mathrm{LSTD}})\,(A_{\mathrm{LSTD}})^{\mathrm{D}}\right)\theta_0$ represents any vector from $\mathrm{Ker}\,(A_{\mathrm{LSTD}})$, because

$$\left((A_{\mathrm{LSTD}})\,(A_{\mathrm{LSTD}})^{\mathrm{D}}\right) \text{ is a projector onto } \mathrm{Col}\left((A_{\mathrm{LSTD}})^k\right) \text{ along } \mathrm{Ker}\left((A_{\mathrm{LSTD}})^k\right),$$

while $\left(I - (A_{\mathrm{LSTD}})\,(A_{\mathrm{LSTD}})^{\mathrm{D}}\right)$ is the complementary projector onto

$$\mathrm{Ker}\left((A_{\mathrm{LSTD}})^k\right) \text{ along } \mathrm{Col}\left((A_{\mathrm{LSTD}})^k\right),$$

where $k = \mathbf{Index}\,(A_{\mathrm{LSTD}})$. Consequently, we know

$$\mathrm{Col}\left(I - (A_{\mathrm{LSTD}})\,(A_{\mathrm{LSTD}})^{\mathrm{D}}\right) = \mathrm{Ker}\left((A_{\mathrm{LSTD}})^k\right).$$

Since $H_{\text{TD}} = I - A_{\text{LSTD}}$ is semiconvergent, we know

$$\textbf{Index} \left(A_{\text{LSTD}}\right) \leq 1,$$

giving us

$$\text{Col}\left(I - \left(A_{\text{LSTD}}\right)\left(A_{\text{LSTD}}\right)^{\text{D}}\right) = \text{Ker}\left(A_{\text{LSTD}}\right).$$

Therefore, $\left(I - \left(A_{\text{LSTD}}\right)\left(A_{\text{LSTD}}\right)^{\text{D}}\right)\theta_0$ can be any vector in $\text{Ker}\left(A_{\text{LSTD}}\right)$. Additionally, because $\textbf{Index}\left(A_{\text{LSTD}}\right) \leq 1$, we have

$$\left(A_{\text{LSTD}}\right)^{\text{D}} b_{\text{LSTD}} = \left(A_{\text{LSTD}}\right)^{\#} b_{\text{LSTD}}.$$

In summary, we conclude that any fixed point to which TD converges is the sum of the group inverse solution of the target linear system, denoted by $\left(A_{\text{LSTD}}\right)^{\#} b_{\text{LSTD}}$, and a vector from the null space of $A_{\text{LSTD}}$, i.e., $\text{Ker}\left(A_{\text{LSTD}}\right)$.

### E.2 Proof of Theorem 6.1

**Theorem E.3** (Restatement of Theorem 6.1). *TD converges for any initial point $\theta_0$ if and only if the target linear sytem is consistent:*

$$\theta_{\phi,r} \in \text{Col}\left(\Sigma_{cov} - \gamma\Sigma_{cr}\right)$$

*and semiconvergent:*

$$\rho\left(I - \alpha\left(\Sigma_{cov} - \gamma\Sigma_{cr}\right)\right) < 1,$$

*or else*

$$\rho(I - \alpha\left(\Sigma_{cov} - \gamma\Sigma_{cr}\right)) = 1,$$

*where $\forall\lambda \in \sigma\left(I - \alpha\left(\Sigma_{cov} - \gamma\Sigma_{cr}\right)\right), \lambda = 1$ is the only eigenvalue on the unit circle, and $\lambda = 1$ is semisimple.*

*It converges to* $\left[\left(\Sigma_{cov} - \gamma\Sigma_{cr}\right)^{\text{D}}\theta_{\phi,r} + \left(I - \left(\Sigma_{cov} - \gamma\Sigma_{cr}\right)\left(\Sigma_{cov} - \gamma\Sigma_{cr}\right)^{\text{D}}\right)\theta_0\right] \in \Theta_{\text{LSTD}}$.

*Proof.* As we show in Section 3, TD is fundamentally an iterative method to solve its target linear system:

$$\left(\Sigma_{cov} - \gamma\Sigma_{cr}\right)\theta = \theta_{\phi,r}.$$

When the target linear system is not consistent, this means there is no solution, and naturally TD will not converge. $\theta_{\phi,r} \in \text{Col}\left(\Sigma_{cov} - \gamma\Sigma_{cr}\right)$ is a necessary and sufficient condition for the existence of a solution to the linear system $\left(\Sigma_{cov} - \gamma\Sigma_{cr}\right)\theta = \theta_{\phi,r}$, making it a necessary condition for TD convergence.

From Berman and Plemmons [5, chapter 7, lemma 6.13] or Hensel [20], we know the general necessary and sufficient conditions of convergence of an iterative method for a consistent linear system. We know that given a consistent target linear system, TD converges for any initial point $\theta_0$ if and only if $\left(I - \alpha\left(\Sigma_{cov} - \gamma\Sigma_{cr}\right)\right)$ is semiconvergent.

Therefore, we know TD converges for any initial point $\theta_0$ if and only if (1) the target linear system is consistent:

$$\theta_{\phi,r} \in \text{Col}\left(\Sigma_{cov} - \gamma\Sigma_{cr}\right)$$

and (2)

$$\rho\left(I - \alpha\left(\Sigma_{cov} - \gamma\Sigma_{cr}\right)\right) < 1,$$

or else

$$\rho(I - \alpha\left(\Sigma_{cov} - \gamma\Sigma_{cr}\right)) = 1$$

where $\forall\lambda \in \sigma\left(I - \alpha\left(\Sigma_{cov} - \gamma\Sigma_{cr}\right)\right), \lambda = 1$, is the only eigenvalue on the unit circle, and $\lambda = 1$ is semisimple. and when it converges, it will converges to

$$\left[\left(\Sigma_{cov} - \gamma\Sigma_{cr}\right)^{\text{D}}\theta_{\phi,r} + \left(I - \left(\Sigma_{cov} - \gamma\Sigma_{cr}\right)\left(\Sigma_{cov} - \gamma\Sigma_{cr}\right)^{\text{D}}\right)\theta_0\right] \in \Theta_{\text{LSTD}}.$$

$\square$

### E.3 Proof of Corollary 6.2

**Corollary E.4** (Restatement of Corollary 6.2). *TD is stable if and only if the following conditions hold:*

- $\theta_{\phi,r} \in \mathrm{Col}\,(\Sigma_{cov} - \gamma\Sigma_{cr})$

- $(\Sigma_{cov} - \gamma\Sigma_{cr})$ *is positive semi-stable.*

- $\mathbf{Index}\,(\Sigma_{cov} - \gamma\Sigma_{cr}) \le 1$

*Additionally, if* $(\Sigma_{cov} - \gamma\Sigma_{cr})$ *is M-matrix, positive semi-stable condition can be relaxed to:* $(\Sigma_{cov} - \gamma\Sigma_{cr})$ *is nonnegative stable.*

*Proof.* First, from Lemma E.5 we know that when $(\Sigma_{cov} - \gamma\Sigma_{cr})$ is full rank, there exists $\alpha > 0$ that

$$(I - \alpha\,(\Sigma_{cov} - \gamma\Sigma_{cr}))\ \text{is semiconvergent}$$

if and only if $(\Sigma_{cov} - \gamma\Sigma_{cr})$ is positive stable.

Second, from Lemma E.6 we know that when $(\Sigma_{cov} - \gamma\Sigma_{cr})$ is not full rank, there exists $\alpha > 0$ that $(I - \alpha\,(\Sigma_{cov} - \gamma\Sigma_{cr}))$ is semiconvergent if and only if $(\Sigma_{cov} - \gamma\Sigma_{cr})$ is positive semi-stable and the eigenvalue $\lambda_{(\Sigma_{cov}-\gamma\Sigma_{cr})} = 0 \in \sigma\,(\Sigma_{cov} - \gamma\Sigma_{cr})$ is semisimple. Moreover, from Lemma E.24, we know "the eigenvalue $\lambda_{(\Sigma_{cov}-\gamma\Sigma_{cr})} = 0 \in \sigma\,(\Sigma_{cov} - \gamma\Sigma_{cr})$ is semisimple" is equivalent to $\mathbf{Index}\,(\Sigma_{cov} - \gamma\Sigma_{cr}) = 1$.

Combining two above cases where $\Sigma_{cov} - \gamma\Sigma_{cr}$ is full rank and not full rank, we conclude that there exists $\alpha > 0$ such that $(I - \alpha\,(\Sigma_{cov} - \gamma\Sigma_{cr}))$ is semiconvergent if and only if $(\Sigma_{cov} - \gamma\Sigma_{cr})$ is positive semi-stable and $\mathbf{Index}\,(\Sigma_{cov} - \gamma\Sigma_{cr}) \le 1$.

Finally, by Theorem 6.1, we know that there exists $\alpha > 0$ such that TD converges for any initial point $\theta_0$ if and only if

$$\theta_{\phi,r} \in \mathrm{Col}\,(\Sigma_{cov} - \gamma\Sigma_{cr})$$

and

$$(\Sigma_{cov} - \gamma\Sigma_{cr})\ \text{is positive semi-stable}$$

and

$$\mathbf{Index}\,(\Sigma_{cov} - \gamma\Sigma_{cr}) \le 1.$$

Additionally, When $(\Sigma_{cov} - \gamma\Sigma_{cr})$ is a singular M-matrix, by Berman and Plemmons [5, Chapter 6, Page 150, E11,F12], we know that if $(\Sigma_{cov} - \gamma\Sigma_{cr})$ is positive semi-stable, it must be nonnegative stable. Hence, the proof complete. $\square$

**Lemma E.5.** *Given a square full rank matrix $A$ and a positive scalar $\alpha$, $(I - \alpha A)$ is semiconvergent if and only if $A$ is positive stable and $\alpha \in (0, \epsilon)$ where $\epsilon = \min_{\lambda \in \sigma(A)} \frac{2 \cdot \Re(\lambda)}{|\lambda|^2}$.*

*Proof.* Since $A$ is full rank it has no eigenvalue $\lambda = 0 \in \sigma\,(A)$, therefore, by Lemma A.1 we know that it is impossible that $(I - \alpha A)$ have eigenvalue equal to 1 for any eligible $\alpha$.

By Proposition A.8, we know that $(I - \alpha A)$ is semiconvergent if and only if

$$\rho(I - \alpha A) < 1.$$

Additionally, because

$$\sigma\,(I - \alpha A) \setminus \{1\} = \sigma\,(I - \alpha A)$$

and $\lambda_{(A)} \in \sigma\,(A)$, by Lemma E.7, we know that

$$\forall \lambda_{(I-\alpha A)} \in \sigma\,(I - \alpha A)\,, |\lambda_{(I-\alpha A)}| < 1$$

if and only if

$$\forall \lambda_{(A)} \in \sigma\,(A)\,, \Re\left(\lambda_{(A)}\right) > 0$$

and $\alpha \in (0, \epsilon)$ where $\epsilon = \min_{\lambda \in \sigma(A)} \frac{2 \cdot \Re(\lambda)}{|\lambda|^2}$.

Hence, We can conclude that $(I - \alpha A)$ is semiconvergent if and only if $A$ is positive stable and $\alpha \in (0, \epsilon)$, where

$$\epsilon = \min_{\lambda \in \sigma(A)} \frac{2 \cdot \Re(\lambda)}{|\lambda|^2}.$$

$\square$

**Lemma E.6.** *Given a square, rank deficient matrix $A$ and a positive scalar $\alpha$, $(I - \alpha A)$ is semiconvergent if and only if*

- *$A$ is positive semi-stable*

- *the eigenvalue $\lambda_{(A)} = 0 \in \sigma(A)$ is semisimple or $\mathbf{Index}(A) = 1$*

- *$\alpha \in (0, \epsilon)$, where $\epsilon = \min_{\lambda \in \sigma(A) \backslash \{0\}} \frac{2 \cdot \Re(\lambda)}{|\lambda|^2}$.*

*Proof.* Since $A$ is not full rank it must have eigenvalue $\lambda_{(A)} = 0 \in \sigma(A)$. Then, by Proposition A.8, we know that $(I - \alpha A)$ is semiconvergent if and only if

$$\rho(I - \alpha A) = 1$$

where $\lambda_{(I-\alpha A)} = 1$ is the only eigenvalue on the unit circle, and $\lambda_{(I-\alpha A)} = 1$ is semisimple.

Next, by Lemma E.7, we know that

$$\forall \lambda_{(I-\alpha A)} \in \sigma(I - \alpha A) \backslash \{1\}, |\lambda_{(I-\alpha A)}| < 1$$

if and only if

$$\forall \lambda_{(A)} \in \sigma(A) \backslash \{0\}, \Re(\lambda_{(A)}) > 0$$

and $\alpha \in (0, \epsilon)$ where $\epsilon = \min_{\lambda \in \sigma(A) \backslash \{0\}} \frac{2 \cdot \Re(\lambda)}{|\lambda|^2}$.

Thus, $\forall \lambda_{(A)} \in \sigma(A) \backslash \{0\}$,

$$\Re(\lambda_{(A)}) > 0 \text{ and } \alpha \in (0, \epsilon) \text{ where } \epsilon = \min_{\lambda \in \sigma(A) \backslash \{0\}} \frac{2 \cdot \Re(\lambda)}{|\lambda|^2}$$

are necessary and sufficient condition for $\rho(I - \alpha A) = 1$ where $\lambda_{(I-\alpha A)} = 1$ is the only eigenvalue on the unit circle.

Then by Lemma A.1 we know $\lambda_{(I-\alpha A)} = 1$ is semisimple if and only if $\lambda_{(A)} = 0$ is semisimple.

Therefore, we can conclude that $(I - \alpha A)$ is semiconvergent if and only if $A$ is positive semi-stable and its eigenvalue $\lambda_{(A)} = 0 \in \sigma(A)$ is semisimple and $\alpha \in (0, \epsilon)$ where $\epsilon = \min_{\lambda_{(A)} \in \sigma(A) \backslash \{0\}} \frac{\Re(\lambda_{(A)})}{|\lambda_{(A)}|^2}$.

From Lemma E.24 we know that the eigenvalue

$$\lambda_{(A)} = 0 \in \sigma(A) \text{ being semisimple}$$

is equivalent to $\mathbf{Index}(A) = 1$. Hence, the poof is complete.

$\square$

**Lemma E.7.** *Given a positive scalar $\alpha$ and matrix $A \in \mathbb{R}^{n \times n}$,*

$$\forall \lambda_{(I-\alpha A)} \in \sigma(I - \alpha A) \backslash \{1\}, |\lambda_{(I-\alpha A)}| < 1$$

*if and only if $\forall \lambda_{(A)} \in \sigma(A) \backslash \{0\}$,*

$$\Re(\lambda_{(A)}) > 0 \text{ and } \alpha \in (0, \epsilon) \text{ where } \epsilon = \min_{\lambda_{(A)} \in \sigma(A) \backslash \{0\}} \left( \frac{2 \cdot \Re(\lambda_{(A)})}{|\lambda_{(A)}|^2} \right).$$

*Proof.* Assume that there exists an $\alpha > 0$ such that $\forall \lambda_{(I-\alpha A)} \in \sigma(I - \alpha A) \backslash \{1\}, |\lambda_{(I-\alpha A)}| < 1$. This means that for every nonzero eigenvalue $\lambda_{(A)} \neq 0$ of $A$, the inequality $|1 - \alpha \lambda_{(A)}| < 1$ holds. Define any nonzero eigenvalue of matrix $A$ as $\lambda_{(A)} = a + bi$, where $a$ and $b$ are real numbers, and $i$ is the imaginary unit. Using Lemma A.1, the condition $|1 - \alpha \lambda_{(A)}| < 1$ can be rewritten as:

$$\sqrt{(1 - \alpha a)^2 + (-\alpha b)^2} < 1.$$

Squaring both sides and simplifying, we get:

$$\alpha^2(a^2 + b^2) - 2\alpha a + 1 < 1,$$

which further simplifies to:

$$\alpha^2(a^2 + b^2) - 2\alpha a < 0, \tag{110}$$

and since $\left(a^2 + b^2\right) > 0$, we know that there exists $\alpha$ make Equation (110) hold only if quadratic equation Equation (111) has two roots:

$$(a^2 + b^2)\alpha^2 - 2a\alpha = 0, \tag{111}$$

which means the discriminant of Equation (111): $(-2a)^2 > 0$, so $a \neq 0$. When $\alpha = 0$ and $\frac{2a}{a^2+b^2}$ (they are two roots), the Equation (111) holds. Therefore,

- Assuming $a < 0$, then $\frac{2a}{a^2+b^2} < 0$, so Equation (110) holds if and only if $\alpha \in \left(\frac{2a}{a^2+b^2}, 0\right)$. However, this contradicts the fact that $\alpha > 0$, so it cannot hold.

- Assuming $a > 0$, then $\frac{2a}{a^2+b^2} > 0$, so Equation (110) holds if and only if $\alpha \in \left(0, \frac{2a}{a^2+b^2}\right)$.

Therefore, we can see that $A \in \mathbb{R}^{n \times n}, \forall \lambda_{(I-\alpha A)} \in \sigma\left(I - \alpha A\right) \setminus \{1\}, |\lambda_{(I-\alpha A)}| < 1$ if and only if $\forall \lambda_{(A)} \in \sigma\left(A\right) \setminus \{0\}, a > 0$ and $\alpha \in (0, \epsilon)$ where

$$\epsilon = \min_{\lambda_{(A)} \in \sigma(A) \setminus \{0\}} \left(\frac{2a}{a^2 + b^2}\right) \tag{112}$$

$$= \min_{\lambda_{(A)} \in \sigma(A) \setminus \{0\}} \left(\frac{2 \cdot \Re(\lambda_{(A)})}{|\lambda_{(A)}|^2}\right) \tag{113}$$

Hence, the proof is complete.

$\square$

### E.4 Proof of Corollary 6.3

**Corollary E.8** (Restatement of Corollary 6.3). *When TD is stable, TD converges if and only if learning rate $\alpha \in (0, \epsilon)$ where*

$$\epsilon = \min_{\lambda \in \sigma(\Sigma_{cov} - \gamma \Sigma_{cr}) \setminus \{0\}} \left(\frac{2 \cdot \Re(\lambda)}{|\lambda|^2}\right).$$

*Proof.* When TD is stable, from Corollary 6.2, we know that

$$\theta_{\phi, r} \in \mathrm{Col}\left(\Sigma_{cov} - \gamma \Sigma_{cr}\right)$$

and

$$\left(\Sigma_{cov} - \gamma \Sigma_{cr}\right) \text{ is positive semi-stable}$$

and

$$\mathbf{Index}\left(\Sigma_{cov} - \gamma \Sigma_{cr}\right) \leq 1.$$

In such a case, by Theorem 6.1, we know that TD converges for any initial point if and only if $\left(I - \alpha\left(\Sigma_{cov} - \gamma \Sigma_{cr}\right)\right)$ is semiconvergent.

Next, given the above, by Lemma E.5 and Lemma E.6, we know that $\left(I - \alpha\left(\Sigma_{cov} - \gamma \Sigma_{cr}\right)\right)$ is semiconvergent if and only if

$$\alpha \in (0, \epsilon) \text{ where } \epsilon = \min_{\lambda \in \sigma(\Sigma_{cov} - \gamma \Sigma_{cr}) \setminus \{0\}} \left(\frac{2 \cdot \Re(\lambda)}{|\lambda|^2}\right).$$

Hence, the proof is complete. $\square$

## E.5 Encoder-decoder view

To understand the matrix $A_{\text{LSTD}} = \left[ \Phi^\top \mathbf{D}(I - \gamma \mathbf{P}_\pi)\Phi \right]$, we begin by analyzing the matrix $\mathbf{D}(I - \gamma \mathbf{P}_\pi)$, referred to as the **system's dynamics**, which captures the dynamics of the system (state action temporal difference and the importance of each state). As established in Proposition E.9, $\mathbf{D}(I - \gamma \mathbf{P}_\pi)$ is a nonsingular M-matrix. Being positive stable is an important property of a nonsingular M-matrix[5, Chapter 6, Theorem 2.3, G20]. Moreover, since $\Phi^\top \mathbf{D}(I - \gamma \mathbf{P}_\pi)\Phi$ shares the same nonzero eigenvalues as $\mathbf{D}(I - \gamma \mathbf{P}_\pi)\Phi\Phi^\top$ (Lemma E.18), positive semi-stability of one implies the same for the other. Interestingly, the matrix $\mathbf{D}(I - \gamma \mathbf{P}_\pi)\Phi\Phi^\top$ acts as an encoding-decoding process, as shown in Equation (114). This encoding-decoding process involves two transformations: First, $\Phi$ serves as an encoder, mapping the system's dynamics into a $d$-dimensional feature space; then, $\Phi^\top$ acts as a decoder, transforming it back to the $|\mathcal{S} \times \mathcal{A}|$-dimensional space. The dimensions of these transformations are explicitly marked in Equation (114). Since from Corollary 6.2 we know that $\Phi^\top \mathbf{D}(I - \gamma \mathbf{P}_\pi)\Phi$ being positive semi-stable is one of the necessary conditions for convergence of TD. Therefore, whether this encoding-decoding process can preserve the positive semi-stability of the system's dynamics determines whether this necessary condition for convergence can be satisfied.

$$
\overbrace{\mathbf{D}(I - \gamma \mathbf{P}_\pi)}^{|\mathcal{S} \times \mathcal{A}|} \underbrace{\overbrace{\Phi}^{d}}_{\text{Encoder}} \underbrace{\overbrace{\Phi^\top}^{|\mathcal{S} \times \mathcal{A}|}}_{\text{Decoder}}
\tag{114}
$$

**Proposition E.9.** $(I - \gamma \mathbf{P}_\pi)$ *and* $\mathbf{D}(I - \gamma \mathbf{P}_\pi)$ *are both non-singular M-matrices and strictly diagonally dominant.*

## E.6 TD in the over-parameterized setting

**Over-parameterized orthogonal state-action feature vectors** To gain a more concrete understanding of the Encoder-Decoder View, consider an extreme setting where the abstraction and compression effects of the encoding-decoding process are entirely eliminated, and with no additional constraints imposed. In this scenario, all information from the system's dynamics should be fully retained, and if the Encoder-Decoder view is valid, the positive semi-stability of the system's dynamics should be preserved. This setting corresponds to $| \mathcal{S} \times \mathcal{A} | \le d$ (overparameterization), and more importantly each state-action pair is represented by a different, orthogonal feature vector[16], mathematically, $\phi(s_i, a_i)^\top \phi(s_j, a_j) = 0, \forall i \ne j$. In this case, we prove that $\left[ \mathbf{D}(I - \gamma \mathbf{P}_\pi)\Phi\Phi^\top \right]$ is also a nonsingular M-matrix[17], just like $\mathbf{D}(I - \gamma \mathbf{P}_\pi)$, ensuring that positive semi-stability is perfectly preserved during the encoding-decoding process. Furthermore, we show that in this case, the other convergence conditions required by Corollary 6.2 are also satisfied. Thus, TD is stable under this scenario, as formally stated in Proposition E.10.

**Proposition E.10.** *TD is stable when the feature vectors of distinct state-action pairs are orthogonal, i.e.,*

$$
\phi(s_i, a_i)^\top \phi(s_j, a_j) = 0, \quad \forall (s_i, a_i) \ne (s_j, a_j).
$$

**Over-parameterized linearly independent state-action feature vectors** Now, consider a similar over-parameterized setting to the previous one, but without excluding the abstraction and compression effects of the encoding-decoding process process. This assumes a milder condition, where state-action feature vectors are linearly independent (Condition J.1) rather than orthogonal. In this scenario, feature vectors may still exhibit correlation, potentially leading to abstraction or compression in the encoder-decoder process. The ability of this process to preserve the positive semi-stability of system's dynamics depends on the choice of features. Not all features guarantee this unless the system's dynamics possesses specific structural properties (for example, in the on-policy setting, any features can preserves positive semi-stability in system's dynamics). We provide necessary and sufficient condition of TD convergence for this setting in Corollary E.11. These results show that both the consistency condition and index condition in Corollary 6.2 are satisfied in this setting. Only the positive semi-stability condition cannot be guaranteed, which aligns with our previous discussion. Additionally, the star MDP from Baird [2] is a notable example demonstrating that TD can diverge with an over-parameterized linear function approximator, where each state is represented by different,

---

[16]In this paper, "orthogonal" does not imply "orthonormal," as the latter imposes an additional norm constraint.

[17]The proof is included in proof of Proposition E.10

linearly independent feature vectors. Xiao et al. [44, Theorems 2] further investigate the necessary and sufficient conditions for the convergence of TD with an over-parameterized linear approximation in the batch setting, assuming that each state's feature vector is linearly independent. However, the proposed conditions for TD are neither sufficient nor necessary. A detailed analysis is provided in Appendix I. Che et al. [11, Proposition 3.1] attempts to refine the TD convergence results in Xiao et al. [44], providing sufficient conditions for the convergence of TD under the same setting. However, as we explain in Appendix I, this condition, as presented, cannot hold. The results in this section provide the correct necessary and sufficient condition.

If we take a further step and remove the assumption that the feature vectors for each state-action pair are linearly independent, while still operating in over-parameterized setting (i.e., $| \mathcal{S} \times \mathcal{A} | \leq d$, but $\Phi$ is not necessarily full row rank), the consistency of the target linear system (i.e., the existence of a fixed point) can no longer be guaranteed, as demonstrated earlier in Section 4. Naturally, this leads to stricter convergence conditions for TD compared to under the previous assumption.

**Corollary E.11.** *Let $\Phi$ be full row rank. Then TD is stable if and only if either $\left[\Phi^\top \mathbf{D}(I - \gamma \mathbf{P}_\pi)\Phi\right]$ is positive semi-stable or $\left[\mathbf{D}(I - \gamma \mathbf{P}_\pi)\Phi\Phi^\top\right]$ is positive stable.*

## E.7  Proof of Proposition E.9

*Proof.* As $\mathbf{P}_\pi$ is row stochastic matrix, we know that $0 \leq \gamma \mathbf{P}_\pi \leq 1$, we obtain that $(I - \gamma \mathbf{P}_\pi)$ is Z-matrix by Definition A.9. As $\mathbf{D}$ is positive diagonal matrix then $\mathbf{D}(I - \gamma \mathbf{P}_\pi)$ is also an Z-matrix. From below and by Berman and Plemmons [5, page 137, N38] that any inverse-positive Z-matrix is nonsingular M-matrix, we can see that $(I - \gamma \mathbf{P}_\pi)$ and $\mathbf{D}(I - \gamma \mathbf{P}_\pi)$ is nonsingular M-matrix:

$$
\begin{aligned}
(I - \gamma \mathbf{P}_\pi)^{-1} &= (I - \gamma \mathbf{P}_\pi)^{-1} \\
&= \sum_{i=0}^{\infty} (\gamma \mathbf{P}_\pi)^i \quad \text{(convergence of matrix power series due to } \rho(\gamma \mathbf{P}_\pi) < 1) \quad (115) \\
&\geq 0 \quad (\gamma \mathbf{P}_\pi \geq 0 \text{ and } \geq 0),
\end{aligned}
$$

so $(I - \gamma \mathbf{P}_\pi)$ is nonsingular M-matrix, then since $D$ is positive definite diagonal matrix, using Lemma E.12 we know $\mathbf{D}(I - \gamma \mathbf{P}_\pi)$ is also nonsingular M-matrix. $\square$

**Lemma E.12.** *Given any positive definite diagonal matrix $G$, if A is an nonsingular M-matrix, then $GA$ and $AG$ are also nonsingular M-matrices.*

*Proof.* If $A$ is nonsingular M-matrix, then for any positive definite diagonal matrix $G$, off-diagonal entries of matrix $GA$ or $AG$ is also non-positive, means they are also Z-matrix. Furthermore, since by property of nonsingular M-matrix[5, Chapter 6, Page 137, N38], $A^{-1} \geq 0$, then we can see that $(GA)^{-1} = A^{-1}G^{-1} \geq 0$ and $(AG)^{-1} = G^{-1}A^{-1} \geq 0$, therefore we know that $GA$ and $AG$ are both Z-matrix and inverse-positive, so they are nonsingular M-matrix. $\square$

## E.8  Linearly independent features, rank invariance, and nonsingularity

While there may be an expectation that if $\Phi$ is full column rank, TD is more stable, full column rank does not guarantee any of the conditions of Corollary 6.2. The stability conditions for the full rank case are not relaxed from Corollary 6.2, which is reflected in Proposition E.13. Additionally, in Proposition E.14, we see that rank invariance ensures only the consistency of the target linear system but does not relax other stability conditions.

**Proposition E.13.** *When $\Phi$ has full column rank (satisfying Condition 4.3), TD is stable if and only if the following conditions hold*

    *1. $\left(\Phi^\top \mathbf{D}R\right) \in \mathrm{Col}\left(\Phi^\top \mathbf{D}(I - \gamma \mathbf{P}_\pi)\Phi\right)$*

    *2. $\left[\Phi^\top \mathbf{D}(I - \gamma \mathbf{P}_\pi)\Phi\right]$ is positive semi-stable*

    *3. $\mathbf{Index}\left(\Phi^\top \mathbf{D}(I - \gamma \mathbf{P}_\pi)\Phi\right) \leq 1$.*

*If $(\Phi^\top \mathbf{D}(I - \gamma \mathbf{P}_\pi)\Phi)$ is an M-matrix, the positive semi-stable condition can be relaxed to: $\left(\Phi^\top \mathbf{D}(I - \gamma \mathbf{P}_\pi)\Phi\right)$ is nonnegative stable.*

**Proposition E.14.** *Assuming rank invariance (Condition 4.1) holds, TD is stable if and if only the following 2 conditions hold: (1)$(\Sigma_{cov} - \gamma\Sigma_{cr})$ is positive semi-stable. (2) $\mathbf{Index}\,(\Sigma_{cov} - \gamma\Sigma_{cr}) \leq 1$.*

**Nonsingular linear system** When the target linear system is nonsingular, the solution of target linear system (the fixed point of TD) must exist and be unique. Additionally, the necessary and sufficient condition for TD to be stable reduces to the condition that $A_{\mathrm{LSTD}}$ is positive stable, as concluded in Corollary E.15. Interestingly, if $(\Phi\Phi^\top)$ is a Z-matrix, meaning that the feature vectors of all state-action pairs have non-positive correlation (i.e., $\forall i \neq j, \phi(s_i, a_i)^\top \phi(s_j, a_j) \leq 0$), and its product with another Z-matrix, $\mathbf{D}(I - \gamma\mathbf{P}_\pi)$, is also a Z-matrix, then $(\mathbf{D}(I - \gamma\mathbf{P}_\pi)\Phi\Phi^\top)$ is a nonsingular M-matrix. In this case, using the encoder-decoder perspective we presented earlier, we can easily prove that TD is stable. This result is formalized in Corollary E.16.

**Corollary E.15.** *When $(\Sigma_{cov} - \gamma\Sigma_{cr})$ is nonsingular (satisfying Condition 4.4), TD is stable if and only if $(\Sigma_{cov} - \gamma\Sigma_{cr})$ is positive stable.*

**Corollary E.16.** *When $(\Sigma_{cov} - \gamma\Sigma_{cr})$ is nonsingular (satisfying Condition 4.4) and two matrices:*

$$\Phi\Phi^\top, \left(\mathbf{D}(I - \gamma\mathbf{P}_\pi)\Phi\Phi^\top\right)$$

*are Z-matrices, TD is stable.*

### E.9 Linearly independent features

#### E.9.1 Proof of Proposition E.13

*Proof.* Since $\Phi$ is full column rank does not necessarily imply any of three conditions in Corollary 6.2, therefore, its existence will not alter the condition of TD being stable. When $\Phi$ is full column rank , TD is stable if and only if the three conditions in Corollary 6.2 hold. $\qquad\square$

### E.10 Rank invariance

#### E.10.1 Proof of Proposition E.14

*Proof.* When $\mathrm{Rank}\,(\Sigma_{cov}) = \mathrm{Rank}\,(\Sigma_{cov} - \gamma\Sigma_{cr})$, from Proposition 4.2 we know that it implies $\theta_{\phi,r} \in \mathrm{Col}\,(\Sigma_{cov} - \gamma\Sigma_{cr})$. Then, as $\mathrm{Rank}\,(\Sigma_{cov}) = \mathrm{Rank}\,(\Sigma_{cov} - \gamma\Sigma_{cr})$ does not necessarily imply "$(\Sigma_{cov} - \gamma\Sigma_{cr})$ is positive semi-stable" or "$\mathbf{Index}\,(\Sigma_{cov} - \gamma\Sigma_{cr}) \leq 1$". By Corollary 6.2, we know that when when $\mathrm{Rank}\,(\Sigma_{cov}) = \mathrm{Rank}\,(\Sigma_{cov} - \gamma\Sigma_{cr})$, TD is stable if and only if "$(\Sigma_{cov} - \gamma\Sigma_{cr})$ is positive semi-stable" and "$\mathbf{Index}\,(\Sigma_{cov} - \gamma\Sigma_{cr}) \leq 1$". $\qquad\square$

### E.11 Nonsingular linear system

#### E.11.1 Proof of Corollary E.15

*Proof.* Assuming that $\Phi$ is full column rank and rank invariance (Condition 4.1) holds, by Proposition 4.5 we know that

$$(\Sigma_{cov} - \gamma\Sigma_{cr})$$

is nonsingular if and only if $\Phi$ is full column rank and rank invariance (Condition 4.1) holds. Therefore, $\mathbf{Index}\,(\Sigma_{cov} - \gamma\Sigma_{cr}) = 0$ and $(\Sigma_{cov} - \gamma\Sigma_{cr})$ has no eigenvalue equal to 0. Consequently, $(\Sigma_{cov} - \gamma\Sigma_{cr})$ is positive semi-stable if and only if it is positive stable. Moreover, by Proposition 4.2, we know that $\mathrm{Rank}\,(\Sigma_{cov}) = \mathrm{Rank}\,(\Sigma_{cov} - \gamma\Sigma_{cr})$ implies $\theta_{\phi,r} \in \mathrm{Col}\,(\Sigma_{cov} - \gamma\Sigma_{cr})$. Finally, from Corollary 6.2, we know that when $\Phi$ is full column rank and $\mathrm{Rank}\,(\Sigma_{cov}) = \mathrm{Rank}\,(\Sigma_{cov} - \gamma\Sigma_{cr})$, TD is stable if and only if $(\Sigma_{cov} - \gamma\Sigma_{cr})$ is positive stable. $\qquad\square$

#### E.11.2 Proof of Corollary E.16

*Proof.* When each feature has nonpositive correlation, the matrix $\Phi\Phi^\top$ has nonpositive off-diagonal entries and is thus a Z-matrix. At the same time, it is clearly symmetric and positive semidefinite, meaning all of its nonzero eigenvalues are positive. This implies that it is also an M-matrix [5, Chapter 6, Theorem 4.6, E11]. From this property and Proposition E.9, it follows that $\mathbf{D}(I - \gamma\mathbf{P}_\pi)$ is a nonsingular M-matrix. Therefore, when $\Phi\Phi^\top\mathbf{D}(I - \gamma\mathbf{P}_\pi)$ is a Z-matrix, it is also an M-matrix

[5, Chapter 6, Page 159, 5.2], and hence positive semi-stable. Given that $\Sigma_{cov} - \gamma\Sigma_{cr}$ is nonsingular, Lemma E.18 implies:

$$\sigma\left(\Phi^\top \mathbf{D}(I - \gamma\mathbf{P}_\pi)\Phi\right) = \sigma\left(\Phi\Phi^\top \mathbf{D}(I - \gamma\mathbf{P}_\pi)\right) \setminus \{0\}.$$

Thus, $\Phi^\top \mathbf{D}(I - \gamma\mathbf{P}_\pi)\Phi$ is positive stable, and by Corollary E.15, TD is stable.

$\square$

### E.12 Over-parameterization

#### E.12.1 Proof of Corollary E.11

*Proof.* Assuming $\Phi$ is full row rank, by Proposition J.2 we know that target linear system is universal consistent so that $\theta_{\phi,r} \in \mathrm{Col}\left(\Sigma_{cov} - \gamma\Sigma_{cr}\right)$. Then, by Lemma E.17 we know that **Index** $\left(\Phi^\top \mathbf{D}(I - \gamma\mathbf{P}_\pi)\Phi\right) \leq 1$, and by Corollary 6.2 we can conclude that in such a setting, TD is stable if and only if $\left(\Phi^\top \mathbf{D}(I - \gamma\mathbf{P}_\pi)\Phi\right)$ is positive semi-stable. Additionally, by Lemma E.17 we see $\sigma\left(\Phi^\top \mathbf{D}(I - \gamma\mathbf{P}_\pi)\Phi\right) \setminus \{0\} = \sigma\left(\mathbf{D}(I - \gamma\mathbf{P}_\pi)\Phi\Phi^\top\right)$, as we know $\mathbf{D}(I - \gamma\mathbf{P}_\pi)\Phi\Phi^\top$ is non-singular matrix, so $\left(\Phi^\top \mathbf{D}(I - \gamma\mathbf{P}_\pi)\Phi\right)$ is positive semi-stable if and only if $\left(\mathbf{D}(I - \gamma\mathbf{P}_\pi)\Phi\Phi^\top\right)$ is positive stable. We can conclude that TD is stable if and only if $\left(\mathbf{D}(I - \gamma\mathbf{P}_\pi)\Phi\Phi^\top\right)$ is positive stable. $\square$

**Lemma E.17.** *If $\Phi$ is full row rank,*

$$\mathbf{Index}\left(\Phi^\top \mathbf{D}(I - \gamma\mathbf{P}_\pi)\Phi\right) \leq 1$$

*and*

$$\sigma\left(\Phi^\top \mathbf{D}(I - \gamma\mathbf{P}_\pi)\Phi\right) \setminus \{0\} = \sigma\left(\mathbf{D}(I - \gamma\mathbf{P}_\pi)\Phi\Phi^\top\right).$$

*Proof.* Given that $\Phi$ is full row rank, as we know $\mathbf{D}(I - \gamma\mathbf{P}_\pi)$ is full rank, so when $h > d$, $\left(\mathbf{D}(I - \gamma\mathbf{P}_\pi)\Phi\Phi^\top\right)$ is full rank, then by Lemma E.19 we know that: **Index** $\left(\Phi^\top \mathbf{D}(I - \gamma\mathbf{P}_\pi)\Phi\right) = 1$.

When $h = d$, $\Phi$ is a full rank square matrix, so $\Phi^\top \mathbf{D}(I - \gamma\mathbf{P}_\pi)\Phi$ is nonsingular matrix, and **Index** $\left(\Phi^\top \mathbf{D}(I - \gamma\mathbf{P}_\pi)\Phi\right) = 0$. We can conclude that given that $\Phi$ is full row rank,

$$\mathbf{Index}\left(\Phi^\top \mathbf{D}(I - \gamma\mathbf{P}_\pi)\Phi\right) \leq 1.$$

Next, $\Phi$ is full row rank, so $\Phi\Phi^\top$ is also full rank, therefore $\mathbf{D}(I - \gamma\mathbf{P}_\pi)\Phi\Phi^\top$ is a full rank matrix, and then by Lemma E.18, we know that:

$$\sigma\left(\Phi^\top \mathbf{D}(I - \gamma\mathbf{P}_\pi)\Phi\right) \setminus \{0\} = \sigma\left(\mathbf{D}(I - \gamma\mathbf{P}_\pi)\Phi\Phi^\top\right).$$

$\square$

**Lemma E.18.** *Given any matrix $A \in \mathbb{C}^{m \times n}$ and matrix $B \in \mathbb{C}^{n \times m}$, suppose $m \geq n$, then the matrices $AB$ and $BA$ share the same non-zero eigenvalues:*

$$\sigma\left(AB\right) \setminus \{0\} = \sigma\left(BA\right) \setminus \{0\},$$

*and every non-zero eigenvalue's algebraic multiplicity:*

$$\forall \lambda \in \sigma\left(AB\right) \setminus \{0\}, \mathrm{alg\,mult}_{\mathbf{AB}}(\lambda) = \mathrm{alg\,mult}_{\mathbf{BA}}(\lambda).$$

*Proof.* Given any matrix $A \in \mathbb{C}^{m \times n}$ and matrix $B \in \mathbb{C}^{n \times m}$, suppose $m \geq n$. From [26, Chapter Solution, Page 128, 7.1.19(b)], we know that it has:

$$\det\left(AB - \lambda I\right) = (-\lambda)^{m-n} \det\left(BA - \lambda I\right),$$

where $\det\left(AB - \lambda I\right)$ is characteristic polynomial of matrix $AB$ and $\det\left(BA - \lambda I\right)$ is characteristic polynomial of matrix $BA$. Therefore, they share the same nonzero eigenvalues and every nonzero eigenvalues' algebraic multiplicity. $\square$

**Lemma E.19.** *Given any matrix $A \in \mathbb{C}^{m \times n}$ and matrix $B \in \mathbb{C}^{n \times m}$, suppose $m > n$ and $A$ is full column rank and $B$ is full row rank, if $BA$ is nonsingular matrix, then:*

$$\mathbf{Index}\,(AB) = 1.$$

*Proof.* Given that $m > n$ and $A$ is full column rank and $B$ is full row rank, and $BA$ is nonsingular matrix. let's define Jordon form of $AB$ as

$$P^{-1}\,(AB)\,P = J = \left[ \begin{array}{cc} J_{\lambda \neq 0} & 0 \\ 0 & J_{\lambda = 0} \end{array} \right],$$

where $J_{\lambda \neq 0}$ is composed by all Jordan blocks of nonzero eigenvalues, and $J_{\lambda = 0}$ is composed by all Jordan blocks of eigenvalue 0. Next, we define Jordon form of $BA$ as:

$$\bar{P}^{-1}\,(BA)\,\bar{P} = \bar{J}_{n \times n}.$$

$\bar{J}$ is full rank matrix: $\mathrm{Rank}\,(\bar{J}) = n$. Since $BA$ is a nonsingular matrix, then by Lemma E.18 we know that $AB$ and $BA$ share the same non-zero eigenvalue and every non-zero eigenvalue's algebraic multiplicity, so

$$\sigma\,(AB) = \sigma\,(BA) \cup \{0\},$$

and

$$\forall \lambda \in \sigma\,(BA)\,, \mathrm{alg\,mult}_{\mathbf{AB}}(\lambda) = \mathrm{alg\,mult}_{\mathbf{BA}}(\lambda),$$

which means we have that $J_{\lambda \neq 0}$ is a nonsingular matrix whose size is equal to $\bar{J}_{n \times n}$, which is an $n \times n$ matrix, so $\mathrm{Rank}\,(J_{\lambda \neq 0}) = n$. Assume that eigenvalue 0 of matrix $\Phi^{\top} \mathbf{D}(I - \gamma \mathbf{P}_{\pi})\Phi$ is not semisimple, means that $\mathrm{Rank}\,(J_{\lambda = 0}) > 0$, then clearly $\mathrm{Rank}\,(J) > n$. In this case from Fact C.6 we know it violates the maximum rank $J$ can have, which is $n$, as $\mathrm{Rank}\,(A) = n$ and $\mathrm{Rank}\,(B) = n$, so it is impossible. Finally, we conclude that the eigenvalue 0 of matrix $AB$ is necessarily semisimple, so by Lemma E.24, we know that $\mathbf{Index}\,(AB) = 1$. $\qquad \square$

### E.12.2 Proof of Proposition E.10

**Proposition E.20** (Restatement of Proposition E.10). *When the state-action pairss features are orthogonal to each other, TD is table.*

*Proof.* When the state-action pairs' feature are orthogonal to each other, we know that the rows of $\Phi$ are orthogonal to each other, as well as linearly independent, so $\Phi$ is full row rank and $\Phi \Phi^{\top}$ is a positive definite diagonal matrix. Subsequently, by Proposition E.9 we know $\mathbf{D}(I - \gamma \mathbf{P}_{\pi})$ is a nonsingular M-matrix. Therefore, by Lemma E.12 we can see that $\mathbf{D}(I - \gamma \mathbf{P}_{\pi})\Phi \Phi^{\top}$ is also a nonsingular M-matrix, and then by the property of nonsingular M-matrix [5, Chapter 6, Page 135, G20], we know that $\mathbf{D}(I - \gamma \mathbf{P}_{\pi})\Phi \Phi^{\top}$ is positive stable. Hence, by Corollary E.11, TD is stable. $\quad \square$

### E.13 On-policy

#### E.13.1 Alignment with previous results

In the on-policy setting, it is well-known that if $\Phi$ has full column rank (linearly independent features (Condition 4.3)), then $\left[\Phi^{\top} \mathbf{D}(I - \gamma \mathbf{P}_{\pi})\Phi\right]$ is positive definite, which directly supports the proof of TD's convergence [40]. This result aligns with our off-policy findings in Corollary 6.2, as explained below:

First, as demonstrated in Proposition 4.7, the consistency condition is inherently satisfied in the on-policy setting. Second, because $\left[\Phi^{\top} \mathbf{D}(I - \gamma \mathbf{P}_{\pi})\Phi\right]$ is positive definite, all its eigenvalues have positive real parts (as shown in Property A.4), which ensures that it is positive stable. Additionally, since $\left[\Phi^{\top} \mathbf{D}(I - \gamma \mathbf{P}_{\pi})\Phi\right]$ is nonsingular, we have $\mathbf{Index}\,(\Phi^{\top} \mathbf{D}(I - \gamma \mathbf{P}_{\pi})\Phi) = 0$. Thus, both the positive semi-stability condition and the index condition are satisfied, so the necessary and sufficient conditions for TD being stable are fully met.

**Proposition E.21.** *In the on-policy setting ($\mu \mathbf{P}_{\pi} = \mu$), $\left[\Phi^{\top} \mathbf{D}(I - \gamma \mathbf{P}_{\pi})\Phi\right]$ is an RPN matrix.*

*Proof.* In the n-policy setting, as show in [40], $\left[\mathbf{D}(\gamma \mathbf{P}_{\pi} - I)\right]$ is negative definite, therefore, $\left[\mathbf{D}(I - \gamma \mathbf{P}_{\pi})\right]$ is positive definite. Hence, by Lemma A.6, we know that $\Phi^{\top} \mathbf{D}(I - \gamma \mathbf{P}_{\pi})\Phi$ is RPN matrix.

$\qquad \square$

### E.13.2 Proof of Theorem 6.4

**Theorem E.22** (Restatement of Theorem 6.4). *In the on-policy setting when $\Phi$ is not full column rank, TD is stable.*

*Proof.* First, as shown in Proposition E.21,

$$\left[\Phi^\top \mathbf{D}(I - \gamma \mathbf{P}_\pi)\Phi\right]$$

is an RPN matrix, and from Lemma E.23, we know that its eigenvalue $\lambda = 0 \in \sigma(A)$ is semisimple. Subsequently, by Lemma E.24, we can obtain that

$$\mathbf{Index}\left(\Phi^\top \mathbf{D}(I - \gamma \mathbf{P}_\pi)\Phi\right) = 1.$$

Second, because $\mathbf{D}(I - \gamma \mathbf{P}_\pi)$ is positive definite, by Lemma A.16, we know that

$$\mathrm{Ker}\left(\Phi^\top \mathbf{D}(I - \gamma \mathbf{P}_\pi)\Phi\right) = \mathrm{Ker}\left(\Phi\right).$$

Then, by Lemma C.1, we know that

$$\mathrm{Rank}\left(\Sigma_{cov}\right) = \mathrm{Rank}\left(\Sigma_{cov} - \gamma \Sigma_{cr}\right).$$

Moreover, from Proposition 4.2 we know that $\theta_{\phi,r} \in \mathrm{Col}\left(\Sigma_{cov} - \gamma \Sigma_{cr}\right)$. As

$$\Re\left(x^\mathrm{H}\mathbf{D}(I - \gamma \mathbf{P}_\pi)x\right) > 0 \text{ for all } x \in \mathbb{C}^h\backslash\{0\},$$

so

$$\Re\left(x^\mathrm{H}\Phi^\top\mathbf{D}(I - \gamma \mathbf{P}_\pi)\Phi x\right) > 0 \text{ for all } x \in \mathbb{C}^d\backslash \mathrm{Ker}\left(\Phi\right),$$

we know that for $\left[\Phi^\top \mathbf{D}(I - \gamma \mathbf{P}_\pi)\Phi\right]$, for eigenvector $v_\lambda \in \mathrm{Ker}\left(\Phi\right)$, the corresponding eigenvalue $\lambda = 0$, and for eigenvector $v_\lambda \notin \mathrm{Ker}\left(\Phi\right)$, the corresponding eigenvalue $\Re(\lambda) > 0$. Therefore, $\left[\Phi^\top \mathbf{D}(I - \gamma \mathbf{P}_\pi)\Phi\right]$ is positive semi-stable.

Finally, by Corollary 6.2, we know TD is stable. $\qquad\square$

**Lemma E.23.** *For any singular RPN matrix $A \in \mathbb{C}^{n\times n}$, its eigenvalue $\lambda = 0 \in \sigma(A)$ is semisimple.*

*Proof.* As $A$ is singular RNP matrix, $\lambda = 0 \in \sigma(A)$ and $\mathbf{Index}\,(A) = 1$ by the Property A.5 for singular RNP matrices. Hence, by Lemma E.24 we know that its eigenvalue $\lambda = 0$ is semisimple. $\quad\square$

**Lemma E.24.** *Given a singular matrix $A \in \mathbb{C}^{n\times n}$, its eigenvalue $\lambda = 0 \in \sigma(A)$ is semisimple if and only if $\mathbf{Index}\,(A) = 1$.*

*Proof.* Given a singular matrix $A \in \mathbb{C}^{n\times n}$, and $\lambda$ denoting its eigenvalue. from [26, Page 596, 7.8.4.] we know that $\mathrm{index}\,(\lambda) = 1$ if and only if $\lambda$ is a semisimple eigenvalue. and by definition of index of and eigenvalue:
$$\mathrm{index}\,(\lambda = 0) = \mathbf{Index}\,(A - 0I) = \mathbf{Index}\,(A),$$
so $\mathbf{Index}\,(A) = 1$ if and only if its eigenvalue $\lambda = 0$ is semisimple. $\qquad\square$

### E.14 Expected TD results in this paper can be easily adapted for stochastic TD and batch TD

**Stochastic TD** From the traditional ODE perspective, it has been shown that if expected TD converges to a fixed point, then stochastic TD, with decaying step sizes (as per the Robbins-Monro condition [32, 40] or stricter step size conditions), will also converge to a bounded region within the solution set of the fixed point [3, 19, 13, 40]. Additionally, if stochastic TD can converge, expected TD as a special case of stochastic TD must also converge. Therefore, the necessary and sufficient conditions for the convergence of expected TD can be easily extended to stochastic TD, forming necessary and sufficient conditions for convergence of stochastic TD to a bounded region of the fixed point's solution set. Thus, all our previous result in this section automatically extend to for convergence of stochastic TD to a bounded region of the fixed point's solution set. All the convergence condition results presented in Section 6 naturally hold as convergence condition results for convergence of stochastic TD to a bounded region of the fixed-point's solution set.

For instance, as demonstrated in Theorem 6.4, expected TD is guaranteed to converge in the on-policy setting of Tsitsiklis and Van Roy [40], even without assuming linearly independent features. This implies that stochastic TD with decaying step sizes, under the same on-policy setting and without assuming linearly independent features, converges to a bounded region of the fixed point's solution set. In other words, the linearly independent features assumption in Tsitsiklis and Van Roy [40] can be removed — a result that, to the best of our knowledge, has not been previously established.

**Batch TD**    By replacing $\Phi$, $D$, $\mathbf{P}_\pi$, $\Sigma_{cov}$, $\Sigma_{cr}$ and $\theta_{\phi,r}$ with their empirical counterparts $\widehat{\Phi}$, $\widehat{\mathbf{D}}$, $\widehat{\mathbf{P}_\pi}$, $\widehat{\Sigma}_{cov}$, $\widehat{\Sigma}_{cr}$ and $\widehat{\theta}_{\phi,r}$, respectively, we can extend the convergence results of expected TD to batch TD[18]. For example, Corollary 6.3, which identifies the specific learning rates that make expected TD converge, is particularly useful for batch TD. By replacing each matrix with its empirical counterpart, we can determine which learning rates will ensure batch TD convergence and which will not. This aligns with widely held intuitions in pratical use of batch TD: When a large learning rate doesn't work, trying a smaller one may help. If TD can converge, it must do so with sufficiently small learning rates. In summary, reducing the learning rate can improve stability.

# F    The convergence of PFQI

## F.1    Interpretation of convergence condition and fixed point for PFQI

In Theorem 7.1, the necessary and sufficient condition for PFQI convergence are established, comprising two primary conditions: $b_{\text{LSTD}} \in \text{Col}\left(A_{\text{LSTD}}\right)$, and the semiconvergence of $H_{\text{PFQI}} = I - M_{\text{PFQI}} A_{\text{LSTD}}$. As demonstrated in Section 4, the condition $b_{\text{LSTD}} \in \text{Col}\left(A_{\text{LSTD}}\right)$ ensures that the target linear system is consistent, which implies the existence of a fixed point for PFQI. The semiconvergence of $H_{\text{PFQI}}$ indicates that $H_{\text{PFQI}}$ converges on $\overline{\text{Ker}}\left(A_{\text{LSTD}}\right)$ and functions as the identity matrix on $\text{Ker}\left(A_{\text{LSTD}}\right)$ if $\text{Ker}\left(A_{\text{LSTD}}\right) \neq \{0\}$.

Since any vector can be decomposed into two components — one from $\text{Ker}\left(A_{\text{LSTD}}\right)$ and one from $\overline{\text{Ker}}\left(A_{\text{LSTD}}\right)$ — the above condition ensures that iterations converge to a fixed point for the component in $\overline{\text{Ker}}\left(A_{\text{LSTD}}\right)$ while remaining stable for the component in $\text{Ker}\left(A_{\text{LSTD}}\right)$, with no amplification. Given that $H_{\text{PFQI}} = I - M_{\text{PFQI}} A_{\text{LSTD}}$, if $\text{Ker}\left(A_{\text{LSTD}}\right) \neq \{0\}$, then $H_{\text{PFQI}}$ necessarily includes an eigenvalue equal to 1, necessitating measures to prevent amplification of this component through iterations.

The fixed point to which FQI converges is composed of two elements:

$$\left(M_{\text{PFQI}} A_{\text{LSTD}}\right)^{\text{D}} M_{\text{PFQI}} b_{\text{LSTD}} \text{ and } \left(I - (M_{\text{PFQI}} A_{\text{LSTD}})(M_{\text{PFQI}} A_{\text{LSTD}})^{\text{D}}\right) \theta_0.$$

The term $\left(I - (M_{\text{PFQI}} A_{\text{LSTD}})(M_{\text{PFQI}} A_{\text{LSTD}})^{\text{D}}\right) \theta_0$ represents any vector from $\text{Ker}\left(A_{\text{LSTD}}\right)$, because

$$\left[(M_{\text{PFQI}} A_{\text{LSTD}})(M_{\text{PFQI}} A_{\text{LSTD}})^{\text{D}}\right]$$

acts as a projector onto

$$\text{Col}\left((M_{\text{PFQI}} A_{\text{LSTD}})^{k}\right) \text{ along } \text{Ker}\left((M_{\text{PFQI}} A_{\text{LSTD}})^{k}\right),$$

while

$$\left(I - (M_{\text{PFQI}} A_{\text{LSTD}})(M_{\text{PFQI}} A_{\text{LSTD}})^{\text{D}}\right)$$

serves as the complementary projector onto

$$\text{Ker}\left((M_{\text{PFQI}} A_{\text{LSTD}})^{k}\right) \text{ along } \text{Col}\left((M_{\text{PFQI}} A_{\text{LSTD}})^{k}\right)$$

where $k = \mathbf{Index}\left(M_{\text{PFQI}} A_{\text{LSTD}}\right)$. Consequently,

$$\text{Col}\left(I - (M_{\text{PFQI}} A_{\text{LSTD}})(M_{\text{PFQI}} A_{\text{LSTD}})^{\text{D}}\right) = \text{Ker}\left((M_{\text{PFQI}} A_{\text{LSTD}})^{k}\right).$$

---

[18]While the extension to the on-policy setting is straightforward in principle, in practice when data are sampled from the policy to be evaluated, it is unlikely that $\widehat{\mu} \widehat{\mathbf{P}_\pi} = \widehat{\mu}$ will hold exactly.

Given that $H_{\text{PFQI}}$ is semiconvergent, it indicates that $\textbf{Index}\,(M_{\text{PFQI}}A_{\text{LSTD}}) \leq 1$ since $M_{\text{PFQI}}A_{\text{LSTD}} = I - H_{\text{FQI}}$. Then, we deduce that

$$\text{Col}\left(I - (M_{\text{PFQI}}A_{\text{LSTD}})(M_{\text{PFQI}}A_{\text{LSTD}})^{\text{D}}\right) = \text{Ker}\,(M_{\text{PFQI}}A_{\text{LSTD}})\,.$$

Since $M_{\text{PFQI}}$ is an invertible matrix, it follows that

$$\text{Ker}\,(M_{\text{PFQI}}A_{\text{LSTD}}) = \text{Ker}\,(A_{\text{LSTD}})\,.$$

Thus, $\left(I - (M_{\text{PFQI}}A_{\text{LSTD}})(M_{\text{PFQI}}A_{\text{LSTD}})^{\text{D}}\right)\theta_0$ can represent any vector in $\text{Ker}\,(A_{\text{LSTD}})$. Additionally, given that $\textbf{Index}\,(M_{\text{PFQI}}A_{\text{LSTD}}) \leq 1$, we obtain

$$(M_{\text{PFQI}}A_{\text{LSTD}})^{\text{D}}\,M_{\text{PFQI}}b_{\text{LSTD}} = (M_{\text{PFQI}}A_{\text{LSTD}})^{\#}\,M_{\text{PFQI}}b_{\text{LSTD}}\,.$$

In summary, we can conclude that any fixed point to which PFQI converges is the sum of the group inverse solution of target linear system, i.e., $(A_{\text{LSTD}})^{\#}\,b_{\text{LSTD}}$, and a vector from the nullspace of $A_{\text{LSTD}}$, i.e., $\text{Ker}\,(A_{\text{LSTD}})$.

### F.2  Proof of Theorem 7.1

**Theorem F.1** (Restatement of Theorem 7.1). *PFQI converges for any initial point $\theta_0$ if and only if*

$$\theta_{\phi,r} \in \text{Col}\,(\Sigma_{cov} - \gamma\Sigma_{cr})$$

*and*

$$I - \alpha\sum_{i=0}^{t-1}(I - \alpha\Sigma_{cov})^i\,(\Sigma_{cov} - \gamma\Sigma_{cr})\ \text{is semiconvergent.}$$

*It converges to*

$$\left(\sum_{i=0}^{t-1}(I - \alpha\Sigma_{cov})^i(\Sigma_{cov} - \gamma\Sigma_{cr})\right)^{\text{D}}\sum_{i=0}^{t-1}(I - \alpha\Sigma_{cov})^i\theta_{\phi,r} \tag{116}$$

$$+ \left(I - (\sum_{i=0}^{t-1}(I - \alpha\Sigma_{cov})^i(\Sigma_{cov} - \gamma\Sigma_{cr}))(\sum_{i=0}^{t-1}(I - \alpha\Sigma_{cov})^i(\Sigma_{cov} - \gamma\Sigma_{cr}))^{\text{D}}\right)\theta_0 \tag{117}$$

$$\in \Theta_{LSTD}. \tag{118}$$

*Proof.* From Proposition B.1 we know that PFQI is fundamentally a iterative method to solve the target linear system

$$(\Sigma_{cov} - \gamma\Sigma_{cr})\,\theta = \theta_{\phi,r}\,.$$

Therefore, by Berman and Plemmons [5, Pages 198, lemma 6.13] we know that this iterative method converges if and only if $(\Sigma_{cov} - \gamma\Sigma_{cr})\,\theta = \theta_{\phi,r}$ is consistent:

$$\theta_{\phi,r} \in \text{Col}\,(\Sigma_{cov} - \gamma\Sigma_{cr})$$

and

$$I - \alpha\sum_{i=0}^{t-1}(I - \alpha\Sigma_{cov})^i\,(\Sigma_{cov} - \gamma\Sigma_{cr})\ \text{is semiconvergent.}$$

It converges to

$$\left(\sum_{i=0}^{t-1}(I - \alpha\Sigma_{cov})^i(\Sigma_{cov} - \gamma\Sigma_{cr})\right)^{\text{D}}\sum_{i=0}^{t-1}(I - \alpha\Sigma_{cov})^i\theta_{\phi,r} \tag{119}$$

$$+ \left(I - (\sum_{i=0}^{t-1}(I - \alpha\Sigma_{cov})^i(\Sigma_{cov} - \gamma\Sigma_{cr}))(\sum_{i=0}^{t-1}(I - \alpha\Sigma_{cov})^i(\Sigma_{cov} - \gamma\Sigma_{cr}))^{\text{D}}\right)\theta_0 \tag{120}$$

$$\in \Theta_{LSTD}. \tag{121}$$

$\square$

## F.3  Linearly independent features

Proposition F.2 studies the convergence of PFQI, showing that linearly independent features does not really relax the convergence conditions compared to those without the assumption of linearly independent features. However, linearly independent features remains important for the preconditioner of PFQI: $M_{\text{PFQI}} = \alpha \sum_{i=0}^{t-1} (I - \alpha\Sigma_{cov})^i$, because it is upper bounded with increasing $t$, precisely as $\lim_{t\to\infty} \alpha \sum_{i=0}^{t-1} (I - \alpha\Sigma_{cov})^i = \Sigma_{cov}^{-1}$. Without linearly independent features, $M_{\text{PFQI}} = \alpha \sum_{i=0}^{t-1} (I - \alpha\Sigma_{cov})^i$ will diverge with increasing $t$ (for a detailed proof, see Appendix F.3.1), and consequently, $H_{\text{PFQI}} = I - M_{\text{PFQI}} A_{\text{LSTD}}$ may also diverge. This will cause divergence of the iteration except in some specific cases, like an over-parameterized representation, which we will show in Appendix J.3 where the divergent components can be canceled out. Therefore, we know that when the chosen features are not linearly independent, taking a large or increasing number of updates under each target value function will most likely not only fail to stabilize the convergence of PFQI, but will also make it more divergent. Thus, if the chosen features are a poor representation, the more updates PFQI takes toward the same target value function, the more divergent the iteration becomes. This provides a more nuanced understanding of the impact of slowly updated target networks, as commonly used in deep RL. While they are typically viewed as stabilizing the learning process, they can have the opposite effect if the provided or learned feature representation is not good.

**Proposition F.2.** *Let Condition 4.3 be satisfied, i.e., $\Phi$ is full column rank. Then PFQI converges for any initial point $\theta_0$ if and only if*

$$b_{LSTD} \in \text{Col}\,(A_{LSTD})$$

*and*

$$(I - M_{PFQI} A_{LSTD})\ \text{ is semiconvergent.}$$

*It converges to*

$$\left[ (M_{PFQI} A_{LSTD})^{\text{D}} M_{PFQI} b_{LSTD} + \left( I - (M_{PFQI} A_{LSTD})(M_{PFQI} A_{LSTD})^{\text{D}} \right) \theta_0 \right] \in \Theta_{LSTD}.$$

### F.3.1  When $\Phi$ is not full column rank, $M_{\text{PFQI}}$ diverges as $t$ increases

When $\Phi$ is not full column rank, $\Sigma_{cov} = \Phi^\top \mathbf{D}\Phi$ is a symmetric positive semidefinite matrix, and it can be diagonalized into:

$$\Sigma_{cov} = Q^{-1} \begin{bmatrix} 0 & 0 \\ 0 & K_{r\times r} \end{bmatrix} Q,$$

where $K_{r\times r}$ is a full rank diagonal matrix whose diagonal entries are all positive numbers, $r = \text{Rank}\,(\Sigma_{cov})$, and $Q$ is the matrix of eigenvectors. We will use $K$ to indicate $K_{r\times r}$ for the rest of the proof. Therefore, we know

$$M_{\text{PFQI}} = \alpha \sum_{i=0}^{t-1} (I - \alpha\Sigma_{cov})^i = Q^{-1} \begin{bmatrix} (\alpha t)I & 0 \\ 0 & \left( I - (I - \alpha K)^t \right) K^{-1} \end{bmatrix} Q.$$

Clearly, given a fixed $\alpha$, we can see that as $t \to \infty$, $[(\alpha t)I] \to \infty$ in the matrix above. Therefore, $M_{\text{PFQI}}$ will also diverge.

### F.3.2  Proof of Proposition F.2

**Proposition F.3** (Restatement of Proposition F.2)**.** *When $\Phi$ is full column rank (Condition 4.3 holds), PFQI converges for any initial point $\theta_0$ if and only if*

- $\theta_{\phi,r} \in \text{Col}\,(\Sigma_{cov} - \gamma\Sigma_{cr})$ *and*

- $\left[ I - \alpha \sum_{i=0}^{t-1} (I - \alpha\Sigma_{cov})^i (\Sigma_{cov} - \gamma\Sigma_{cr}) \right]$ *or* $\left[ \gamma\Sigma_{cov}^{-1}\Sigma_{cr} + (I - \alpha\Sigma_{cov})^t (I - \gamma\Sigma_{cov}^{-1}\Sigma_{cr}) \right]$ *is semiconvergent.*

*It converges to*

$$\left(\sum_{i=0}^{t-1}(I-\alpha\Sigma_{cov})^i(\Sigma_{cov}-\gamma\Sigma_{cr})\right)^{\mathrm{D}}\sum_{i=0}^{t-1}(I-\alpha\Sigma_{cov})^i\theta_{\phi,r} \tag{122}$$

$$+\left(I-\left(\sum_{i=0}^{t-1}(I-\alpha\Sigma_{cov})^i(\Sigma_{cov}-\gamma\Sigma_{cr})\right)\left[\sum_{i=0}^{t-1}(I-\alpha\Sigma_{cov})^i(\Sigma_{cov}-\gamma\Sigma_{cr})\right]^{\mathrm{D}}\right)\theta_0 \tag{123}$$

$$\in\Theta_{LSTD}. \tag{124}$$

*Proof.* As we show in Proposition B.1, when $\Phi$ is full column rank,

$$\left[I-\alpha\sum_{i=0}^{t-1}(I-\alpha\Sigma_{cov})^i(\Sigma_{cov}-\gamma\Sigma_{cr})\right]=\left[\gamma\Sigma_{cov}^{-1}\Sigma_{cr}+(I-\alpha\Sigma_{cov})^t(I-\gamma\Sigma_{cov}^{-1}\Sigma_{cr})\right].$$

Then, using Theorem 7.1 we know PFQI converges for any initial point $\theta_0$ if and only if

$$\theta_{\phi,r}\in\mathrm{Col}\left(\Sigma_{cov}-\gamma\Sigma_{cr}\right)$$

and

$$\left[\gamma\Sigma_{cov}^{-1}\Sigma_{cr}+(I-\alpha\Sigma_{cov})^t(I-\gamma\Sigma_{cov}^{-1}\Sigma_{cr})\right]\text{ is semiconvergent.}$$

It converges to

$$\left(\sum_{i=0}^{t-1}(I-\alpha\Sigma_{cov})^i(\Sigma_{cov}-\gamma\Sigma_{cr})\right)^{\mathrm{D}}\sum_{i=0}^{t-1}(I-\alpha\Sigma_{cov})^i\theta_{\phi,r} \tag{125}$$

$$+\left(I-\left(\sum_{i=0}^{t-1}(I-\alpha\Sigma_{cov})^i(\Sigma_{cov}-\gamma\Sigma_{cr})\right)\left[\sum_{i=0}^{t-1}(I-\alpha\Sigma_{cov})^i(\Sigma_{cov}-\gamma\Sigma_{cr})\right]^{\mathrm{D}}\right)\theta_0 \tag{126}$$

$$\in\Theta_{\mathrm{LSTD}}. \tag{127}$$

$\square$

## F.4 Rank invariance and nonsingularity

First, Proposition F.4 shows the necessary and sufficient conditions for convergence of PFQI under rank invariance (Condition 4.1). We see that while the consistency condition can be completely dropped, the other conditions cannot be relaxed, unlike FQI. Second, in Proposition F.4, we provide necessary and sufficient condition for convergence of PFQI under nonsingularity (Condition 4.4). We can see that in such a case, the fixed point is unique, and requires $H_{\mathrm{PFQI}}$ to be strictly convergent ($\rho\left(H_{\mathrm{PFQI}}\right)<1$) instead of being semiconvergent.

**Proposition F.4.** *When rank invariance (Condition 4.1) holds, PFQI converges for any initial point $\theta_0$ if and only if $H_{PFQI}=(I-M_{PFQI}A_{LSTD})$ is semiconvergent. It converges to*

$$\left[(M_{PFQI}A_{LSTD})^{\mathrm{D}}M_{PFQI}b_{LSTD}+\left(I-(M_{PFQI}A_{LSTD})(M_{PFQI}A_{LSTD})^{\mathrm{D}}\right)\theta_0\right]\in\Theta_{LSTD}.$$

**Corollary F.5.** *When $(\Sigma_{cov}-\gamma\Sigma_{cr})$ is nonsingular (Condition 4.4 holds) and $(I-\alpha\Sigma_{cov})$ is nonsingular, PFQI converges for any initial point $\theta_0$ if and only if $\rho\left(I-M_{PFQI}A_{LSTD}\right)<1$. It converges to*

$$\left[(\Sigma_{cov}-\gamma\Sigma_{cr})^{-1}\theta_{\phi,r}\right]\in\Theta_{LSTD}$$

## F.5 Rank invariance

### F.5.1 Proof of Proposition F.4

**Proposition F.6** (Restatement of Proposition F.4). *If $\mathrm{Rank}\left(\Sigma_{cov}\right)=\mathrm{Rank}\left(\Sigma_{cov}-\gamma\Sigma_{cr}\right)$ (Condition 4.1holds), then PFQI converges for any initial point $\theta_0$ if and only if*

$$\left[I-\alpha\sum_{i=0}^{t-1}(I-\alpha\Sigma_{cov})^i(\Sigma_{cov}-\gamma\Sigma_{cr})\right]\text{ is semiconvergent.}$$

*It converges to*

$$\left(\sum_{i=0}^{t-1}(I - \alpha\Sigma_{cov})^i(\Sigma_{cov} - \gamma\Sigma_{cr})\right)^{\mathrm{D}} \sum_{i=0}^{t-1}(I - \alpha\Sigma_{cov})^i\theta_{\phi,r} \tag{128}$$

$$+ \left(I - \left(\sum_{i=0}^{t-1}(I - \alpha\Sigma_{cov})^i(\Sigma_{cov} - \gamma\Sigma_{cr})\right)\left[\sum_{i=0}^{t-1}(I - \alpha\Sigma_{cov})^i(\Sigma_{cov} - \gamma\Sigma_{cr})\right]^{\mathrm{D}}\right)\theta_0 \tag{129}$$

$$\in \Theta_{LSTD}. \tag{130}$$

*Proof.* When $\mathrm{Rank}\,(\Sigma_{cov}) = \mathrm{Rank}\,(\Sigma_{cov} - \gamma\Sigma_{cr})$, from Proposition 4.2 we know that it implies $\theta_{\phi,r} \in \mathrm{Col}\,(\Sigma_{cov} - \gamma\Sigma_{cr})$.

Next, Since $\mathrm{Rank}\,(\Sigma_{cov}) = \mathrm{Rank}\,(\Sigma_{cov} - \gamma\Sigma_{cr})$ does not necessarily imply

$$\left[I - \alpha\sum_{i=0}^{t-1}(I - \alpha\Sigma_{cov})^i\,(\Sigma_{cov} - \gamma\Sigma_{cr})\right] \text{ being semiconvergent,}$$

by Theorem 7.1, we know that when $\mathrm{Rank}\,(\Sigma_{cov}) = \mathrm{Rank}\,(\Sigma_{cov} - \gamma\Sigma_{cr})$, PFQI converges for any initial point $\theta_0$ if and only if

$$\left[I - \alpha\sum_{i=0}^{t-1}(I - \alpha\Sigma_{cov})^i\,(\Sigma_{cov} - \gamma\Sigma_{cr})\right] \text{ is semiconvergent.}$$

It converges to

$$\left(\sum_{i=0}^{t-1}(I - \alpha\Sigma_{cov})^i(\Sigma_{cov} - \gamma\Sigma_{cr})\right)^{\mathrm{D}} \sum_{i=0}^{t-1}(I - \alpha\Sigma_{cov})^i\theta_{\phi,r} \tag{131}$$

$$+ \left(I - \left(\sum_{i=0}^{t-1}(I - \alpha\Sigma_{cov})^i(\Sigma_{cov} - \gamma\Sigma_{cr})\right)\left[\sum_{i=0}^{t-1}(I - \alpha\Sigma_{cov})^i(\Sigma_{cov} - \gamma\Sigma_{cr})\right]^{\mathrm{D}}\right)\theta_0 \tag{132}$$

$$\in \Theta_{LSTD}. \tag{133}$$

$$\square$$

### F.6 Nonsingular linear system

#### F.6.1 Proof of Corollary F.5

**Corollary F.7** (Restatement of Corollary F.5). *When $(\Sigma_{cov} - \gamma\Sigma_{cr})$ is nonsingular (Condition 4.4 holds) and $(I - \alpha\Sigma_{cov})$ is nonsingular, PFQI converges for any initial point $\theta_0$ if and only if*

$$\rho\left(I - \alpha\sum_{i=0}^{t-1}(I - \alpha\Sigma_{cov})^i\,(\Sigma_{cov} - \gamma\Sigma_{cr})\right) < 1.$$

*It converges to* $\left[(\Sigma_{cov} - \gamma\Sigma_{cr})^{-1}\,\theta_{\phi,r}\right] \in \Theta_{LSTD}$.

*Proof.* Given that $(\Sigma_{cov} - \gamma\Sigma_{cr})$ is nonsingular and $(I - \alpha\Sigma_{cov})$ is nonsingular, by Lemma B.2 we know that $\alpha\sum_{i=0}^{t-1}(I - \alpha\Sigma_{cov})^i$ is full rank. Therefore,

$$\alpha\sum_{i=0}^{t-1}(I - \alpha\Sigma_{cov})^i\,(\Sigma_{cov} - \gamma\Sigma_{cr}) \text{ is full rank,}$$

which means it has no eigenvalue equal to 0. Therefore, by Lemma A.1 we know that $I - \alpha\sum_{i=0}^{t-1}(I - \alpha\Sigma_{cov})^i\,(\Sigma_{cov} - \gamma\Sigma_{cr})$ has no eigenvalue equal to 1.

Subsequently, $I - \alpha \sum_{i=0}^{t-1}(I - \alpha\Sigma_{cov})^i (\Sigma_{cov} - \gamma\Sigma_{cr})$ is semiconvergent if and only if

$$\rho\left(I - \alpha\sum_{i=0}^{t-1}(I - \alpha\Sigma_{cov})^i (\Sigma_{cov} - \gamma\Sigma_{cr})\right) < 1.$$

By Proposition 4.2 we know that $\mathrm{Rank}\,(\Sigma_{cov}) = \mathrm{Rank}\,(\Sigma_{cov} - \gamma\Sigma_{cr})$ implies

$$\theta_{\phi,r} \in \mathrm{Col}\,(\Sigma_{cov} - \gamma\Sigma_{cr}).$$

Next, using Theorem 7.1, we can conclude that when $(\Sigma_{cov} - \gamma\Sigma_{cr})$ is nonsingular (Condition 4.4 holds) and $(I - \alpha\Sigma_{cov})$ is also nonsingular, then PFQI converges for any initial point $\theta_0$ if and only if

$$\rho\left(I - \alpha\sum_{i=0}^{t-1}(I - \alpha\Sigma_{cov})^i (\Sigma_{cov} - \gamma\Sigma_{cr})\right) < 1.$$

Additionally, as $\alpha\sum_{i=0}^{t-1}(I - \alpha\Sigma_{cov})^i (\Sigma_{cov} - \gamma\Sigma_{cr})$ is full rank,

$$\left(\sum_{i=0}^{t-1}(I - \alpha\Sigma_{cov})^i (\Sigma_{cov} - \gamma\Sigma_{cr})\right)^{\mathrm{D}} = \left(\sum_{i=0}^{t-1}(I - \alpha\Sigma_{cov})^i (\Sigma_{cov} - \gamma\Sigma_{cr})\right)^{-1}.$$

Hence, we know that converges to

$$\left(\sum_{i=0}^{t-1}(I - \alpha\Sigma_{cov})^i(\Sigma_{cov} - \gamma\Sigma_{cr})\right)^{\mathrm{D}} \sum_{i=0}^{t-1}(I - \alpha\Sigma_{cov})^i\theta_{\phi,r} \tag{134}$$

$$+ \left(I - \left(\sum_{i=0}^{t-1}(I - \alpha\Sigma_{cov})^i(\Sigma_{cov} - \gamma\Sigma_{cr})\right)\left[\sum_{i=0}^{t-1}(I - \alpha\Sigma_{cov})^i(\Sigma_{cov} - \gamma\Sigma_{cr})\right]^{\mathrm{D}}\right)\theta_0 \tag{135}$$

$$= \left(\sum_{i=0}^{t-1}(I - \alpha\Sigma_{cov})^i (\Sigma_{cov} - \gamma\Sigma_{cr})\right)^{-1} \sum_{i=0}^{t-1}(I - \alpha\Sigma_{cov})^i\theta_{\phi,r} \tag{136}$$

$$= (\Sigma_{cov} - \gamma\Sigma_{cr})^{-1}\left(\sum_{i=0}^{t-1}(I - \alpha\Sigma_{cov})^i\right)^{-1} \sum_{i=0}^{t-1}(I - \alpha\Sigma_{cov})^i\theta_{\phi,r} \tag{137}$$

$$= (\Sigma_{cov} - \gamma\Sigma_{cr})^{-1}\theta_{\phi,r} \tag{138}$$

$$\in \Theta_{\mathrm{LSTD}}. \tag{139}$$

$\square$

# G   PFQI as transition between TD and FQI

## G.1   Relationship between PFQI and TD convergence

### G.1.1   Proof of Theorem 8.1

**Theorem G.1** (Restatement of Theorem 8.1). *If TD is stable, then for any finite $t \in \mathbb{N}$ there exists $\epsilon_t \in \mathbb{R}^+$ that for any $\alpha \in (0, \epsilon_t)$ PFQI converges.*

*Proof.* Assuming TD is stable, then by Corollary 6.2 we know that

- $\theta_{\phi,r} \in \mathrm{Col}\,(\Sigma_{cov} - \gamma\Sigma_{cr})$,

- $(\Sigma_{cov} - \gamma\Sigma_{cr})$ is positive semi-stable, and

- **Index** $(\Sigma_{cov} - \gamma\Sigma_{cr}) \leq 1$.

From Theorem 7.1 we know that for any $t \in \mathbb{Z}^+$, if PFQI converges from any initial point $\theta_0$ if and only if

$$\left(I - \alpha \sum_{i=0}^{t-1}(I - \alpha\Sigma_{cov})^i \left(\Sigma_{cov} - \gamma\Sigma_{cr}\right)\right) \text{ semiconvergent} \tag{140}$$

and

$$\theta_{\phi,r} \in \text{Col}\left(\Sigma_{cov} - \gamma\Sigma_{cr}\right).$$

From Lemma E.5 and Lemma E.6, we know that Equation (140) holds when

$$\sum_{i=0}^{t-1}(I - \alpha\Sigma_{cov})^i(\Sigma_{cov} - \gamma\Sigma_{cr})$$

is positive stable or positive semi-stable, where $\lambda = 0 \in \sigma\left(\sum_{i=0}^{t-1}(I - \alpha\Sigma_{cov})^i(\Sigma_{cov} - \gamma\Sigma_{cr})\right)$ is semisimple, and $\alpha \in (0, \epsilon)$ where

$$\epsilon = \min_{\lambda \in \sigma\left(\sum_{i=0}^{t-1}(I - \alpha\Sigma_{cov})^i(\Sigma_{cov} - \gamma\Sigma_{cr})\right)\backslash 0} \frac{\Re(\lambda)}{|\lambda|^2}.$$

Next, from Lemma G.2 we know

$$\sum_{i=0}^{t-1}(I - \alpha\Sigma_{cov})^i \left(\Sigma_{cov} - \gamma\Sigma_{cr}\right) = t\left(\Sigma_{cov} - \gamma\Sigma_{cr}\right) \tag{141}$$

$$- \alpha\left(\sum_{i=2}^{t}\binom{t}{i}(\alpha)^{i-2}(-\Sigma_{cov})^{i-1}\right)\left(\Sigma_{cov} - \gamma\Sigma_{cr}\right). \tag{142}$$

For a fixed, finite $t \in \mathbb{Z}^+$, define an operator

$$T_t(\alpha) = A + \alpha E,$$

where $A = t\left(\Sigma_{cov} - \gamma\Sigma_{cr}\right)$ and $E = \left(\sum_{i=2}^{t}\binom{t}{i}(\alpha)^{i-2}(-\Sigma_{cov})^{i-1}\right)\left(\Sigma_{cov} - \gamma\Sigma_{cr}\right)$, so clearly,

$$T_t(\alpha) = \sum_{i=0}^{t-1}(I - \alpha\Sigma_{cov})^i \left(\Sigma_{cov} - \gamma\Sigma_{cr}\right).$$

From Meyer [26, Page 425, 5.12.4.], we know for any sufficiently small perturbation when the $\ell_2$-norm of perturbation is smaller that the smallest nonzero singular value of unperturbed operator, then the perturbed operator must have greater or equal rank than unperturbed operator. Therefore, for any sufficiently small $\alpha$ such that $\|\alpha E\|_2$ is smaller than the smallest nonzero singular value of $A$, $\text{Rank}\left(T_t(\alpha)\right) \geq \text{Rank}\left(A\right)$. Obviously $\alpha E \in \text{Row}\left(A\right)$ so $\text{Rank}\left(A + \alpha E\right) \leq \text{Rank}\left(A\right)$. Therefore, for any sufficiently small $\alpha$, $\text{Rank}\left(T_t(\alpha)\right) = \text{Rank}\left(A\right)$, so

$$\text{geo mult}_{\mathbf{T_t}(\alpha)}(0) = \text{geo mult}_{\mathbf{A}}(0) = \dim\left(\text{Ker}\left(A\right)\right).$$

It is easy to see that $\lim_{\alpha \to 0} T_t(\alpha) = T_t(0) = A$, so $T_t(\alpha)$ is continuous at the point $\alpha = 0$. By the theorem of continuity of eigenvalues[22, Theorem 5.1], we know that if the operator $T_t(\alpha)$ is continuous at $\alpha = 0$, then the eigenvalues of $T(x)$ also vary continuously near $\alpha = 0$. This means small changes in $\alpha$ will lead to small changes in the eigenvalues of $T_t(\alpha)$. Therefore, if $T_t(0)$ is positive semi-stable, there must exist small enough $\epsilon' \in \mathbb{R}^+$ that for any $\alpha \in (0, \epsilon')$, $T_t(\alpha)$ is positive semi-stable, and the sum of the algebraic multiplicity of every nonzero eigenvalue for $T_t(\alpha)$ is the same as for $T_t(0)$ (no nonzero eigenvalue of $T_t(0)$ changes to 0 by perturbations $(\alpha E)$), which implies $\text{alg mult}_{\mathbf{T_t}(\mathbf{0})}(0) = \text{alg mult}_{\mathbf{T_t}(\alpha)}(0)$. Then, when $\lambda = 0 \in \sigma\left(A\right)$ is semisimple, it means $\text{alg mult}_{\mathbf{T_t}(\mathbf{0})}(0) = \text{geo mult}_{\mathbf{T_t}(\mathbf{0})}(0)$. Since we already know $\text{alg mult}_{\mathbf{T_t}(\mathbf{0})}(0) = \text{alg mult}_{\mathbf{T_t}(\alpha)}(0)$ and $\text{geo mult}_{\mathbf{T_t}(\mathbf{0})}(0) = \text{geo mult}_{\mathbf{T_t}(\alpha)}(0)$, then $\text{alg mult}_{\mathbf{T_t}(\alpha)}(0) = \text{geo mult}_{\mathbf{T_t}(\alpha)}(0)$, $\lambda = 0 \in \sigma\left(\sum_{i=0}^{t-1}(I - \alpha\Sigma_{cov})^i(\Sigma_{cov} - \gamma\Sigma_{cr})\right)$ is semisimple. Thus, if $\alpha \in \min(\epsilon, \epsilon')$, the PFQI convergence condition satisfies.

Finally, we can conclude that if when $\left(\Sigma_{cov} - \gamma\Sigma_{cr}\right)$ is positive semi-stable and $\lambda = 0 \in \sigma\left(A\right)$ is semisimple, for any finite $t \in \mathbb{N}$, there must exist a $\epsilon \in \mathbb{R}^+$ that for any $\alpha \in (0, \epsilon)$, PFQI converges from any initial point $\theta_0$.

$\square$

**Lemma G.2.**

$$\sum_{i=0}^{t-1}(I - \alpha\Sigma_{cov})^i\,(\Sigma_{cov} - \gamma\Sigma_{cr}) = t\,(\Sigma_{cov} - \gamma\Sigma_{cr}) \tag{143}$$

$$- \alpha\left(\sum_{i=2}^{t}\binom{t}{i}(\alpha)^{i-2}(-\Sigma_{cov})^{i-1}\right)(\Sigma_{cov} - \gamma\Sigma_{cr}) \tag{144}$$

*Proof.* As $\Sigma_{cov}$ is symmetric positive semidefinite matrix, it can be diagonalized into:

$$\Sigma_{cov} = Q^{-1}\begin{bmatrix} 0 & 0 \\ 0 & K_{r\times r}\end{bmatrix}Q,$$

where $K_{r\times r}$ is full rank diagonal matrix whose diagonal entries are all positive numbers, and $r = \mathrm{Rank}\,(\Sigma_{cov})$. Thus, it's easy to pick a $\alpha$ that $(I - \alpha K_{r\times r})$ nonsingular, so we will assume $(I - \alpha K_{r\times r})$ as nonsingular matrix for rest of proof. We will also use $K$ to indicate $K_{r\times r}$ for rest of proof. Therefore, we know

$$\alpha\sum_{i=0}^{t-1}(I - \alpha\Sigma_{cov})^i = Q^{-1}\begin{bmatrix} (\alpha t)I & 0 \\ 0 & \left(\alpha\sum_{i=0}^{t-1}(I - \alpha K)^i\right)\end{bmatrix}Q \tag{145}$$

$$= Q^{-1}\begin{bmatrix} (\alpha t)I & 0 \\ 0 & \left(I - (I - \alpha K)^t\right)K^{-1}\end{bmatrix}Q \tag{146}$$

$$= Q^{-1}\begin{bmatrix} (\alpha t)I & 0 \\ 0 & \left(I - \sum_{i=0}^{t}\binom{t}{i}(\alpha)^i(-K)^i\right)K^{-1}\end{bmatrix}Q \tag{147}$$

$$= Q^{-1}\begin{bmatrix} (\alpha t)I & 0 \\ 0 & \left(-\sum_{i=1}^{t}\binom{t}{i}(\alpha)^i(-K)^i\right)K^{-1}\end{bmatrix}Q \tag{148}$$

$$= Q^{-1}\begin{bmatrix} (\alpha t)I & 0 \\ 0 & \left((\alpha t)I - \sum_{i=2}^{t}\binom{t}{i}(\alpha)^i(-K)^{i-1}\right)\end{bmatrix}Q \tag{149}$$

$$= (\alpha t)I - Q^{-1}\begin{bmatrix} 0 & 0 \\ 0 & \left(\sum_{i=2}^{t}\binom{t}{i}(\alpha)^i(-K)^{i-1}\right)\end{bmatrix}Q \tag{150}$$

$$= (\alpha t)I - \left(\sum_{i=2}^{t}\binom{t}{i}(\alpha)^i(-\Sigma_{cov})^{i-1}\right) \tag{151}$$

$$= (\alpha t)I - \left(\alpha^2\sum_{i=2}^{t}\binom{t}{i}(\alpha)^{i-2}(-\Sigma_{cov})^{i-1}\right). \tag{152}$$

Moreover,

$$\sum_{i=0}^{t-1}(I - \alpha\Sigma_{cov})^i\,(\Sigma_{cov} - \gamma\Sigma_{cr}) = t\,(\Sigma_{cov} - \gamma\Sigma_{cr}) \tag{153}$$

$$- \left(\alpha\sum_{i=2}^{t}\binom{t}{i}(\alpha)^{i-2}(-\Sigma_{cov})^{i-1}\right)(\Sigma_{cov} - \gamma\Sigma_{cr}). \tag{154}$$

$\square$

**Lemma G.3.** *Given a matrix:* $A \in \mathbb{R}^{n\times m}$, *if* $B \in \mathbb{R}^{n\times m}$ *and* $\mathrm{Col}\,(B) \subseteq \mathrm{Col}\,(A)$, *then* $\mathrm{Rank}\,(A + B) \le \mathrm{Rank}\,(A)$.

*Proof.* Assuming $\mathrm{Col}\,(B) \subseteq \mathrm{Col}\,(A)$, then we know there exists a matrix $C \in \mathbb{R}^{m\times m}$ such that $B = AC$. Therefore, $A + B = A(I + C)$, and by Fact C.6, we know that $\mathrm{Rank}\,(A + B) \le \min\,(\mathrm{Rank}\,(A), \mathrm{Rank}\,(I + C)) \le \mathrm{Rank}\,(A)$. $\square$

## G.2 Relationship Between PFQI and FQI Convergence

**Proposition G.4** (Restatement of Proposition 8.2). *For a full column rank matrix $\Phi$ and any learning rate $\alpha \in \left(0, \frac{2}{\lambda_{max}(\Sigma_{cov})}\right)$, if there exists an integer $T \in \mathbb{Z}^+$ such that PFQI converges for all $t \geq T$ from any initial point $\theta_0$, then FQI converges from any initial point $\theta_0$.*

*Proof.* From Lemma G.5 we know that when $\Phi$ is full column rank, $H_{\text{PFQI}} = I - \alpha \sum_{i=0}^{t-1}(I - \alpha\Sigma_{cov})^i (\Sigma_{cov} - \gamma\Sigma_{cr})$ can be also expressed as

$$H_{\text{PFQI}} = \left(\gamma\Sigma_{cov}^{-1}\Sigma_{cr} + (I - \alpha\Sigma_{cov})^t(I - \gamma\Sigma_{cov}^{-1}\Sigma_{cr})\right)$$

and the PFQI update equation can be written as:

$$\theta_{k+1} = \left(\gamma\Sigma_{cov}^{-1}\Sigma_{cr} + (I - \alpha\Sigma_{cov})^t(I - \gamma\Sigma_{cov}^{-1}\Sigma_{cr})\right)\theta_k \\ + (I - (I - \alpha\Sigma_{cov})^t)\Sigma_{cov}^{-1}\theta_{\phi,r}.$$

From Theorem 5.1, we know that PFQI converges from any initial point $\theta_0$ if and only if $\theta_{\phi,r} \in \text{Col}(\Sigma_{cov} - \gamma\Sigma_{cr})$ and $H_{\text{PFQI}}$ is semiconvergent.

Next, when $\alpha$ is not sufficiently small, its value can be easily adjusted so that $\alpha\sum_{i=0}^{t-1}(I - \alpha\Sigma_{cov})^i(\Sigma_{cov} - \gamma\Sigma_{cr})$ has no eigenvalue equal to 1. By Lemma A.1, this implies that $I - \alpha\sum_{i=0}^{t-1}(I - \alpha\Sigma_{cov})^i(\Sigma_{cov} - \gamma\Sigma_{cr})$ has no eigenvalue equal to 0, and thus it is nonsingular. Therefore, assuming $I - \alpha\sum_{i=0}^{t-1}(I - \alpha\Sigma_{cov})^i(\Sigma_{cov} - \gamma\Sigma_{cr})$ to be nonsingular in such cases does not lose generality.

When $\alpha$ is sufficiently small, the entries of $\alpha\sum_{i=0}^{t-1}(I - \alpha\Sigma_{cov})^i(\Sigma_{cov} - \gamma\Sigma_{cr})$ are also sufficiently small. From [26, Chapter 4, Page 216], we know that the rank of a matrix perturbed by a sufficiently small perturbation can only increase or remain the same, so $I - \alpha\sum_{i=0}^{t-1}(I - \alpha\Sigma_{cov})^i(\Sigma_{cov} - \gamma\Sigma_{cr})$ is nonsingular since $I$ is nonsingular.

Overall, we can see that $I - \alpha\sum_{i=0}^{t-1}(I - \alpha\Sigma_{cov})^i(\Sigma_{cov} - \gamma\Sigma_{cr})$ is a nonsingular matrix, which has no eigenvalue equal to 0, independent of $t$.

Therefore, when there exists an integer $T \in \mathbb{Z}^+$ such that for all $t \geq T$, $\theta_{\phi,r} \in \text{Col}(\Sigma_{cov} - \gamma\Sigma_{cr})$ holds and $H_{\text{PFQI}}$ is semiconvergent, by theorem of continuity of eigenvalues[22, Theorem 5.1] we know that:

$$\lim_{t\to\infty}\left(\gamma\Sigma_{cov}^{-1}\Sigma_{cr} + (I - \alpha\Sigma_{cov})^t(I - \gamma\Sigma_{cov}^{-1}\Sigma_{cr})\right) = \gamma\Sigma_{cov}^{-1}\Sigma_{cr} \text{ is semiconvergent.}$$

Then, by Theorem 5.1, we know that FQI converges for any initial point $\theta_0$.

$\square$

**Lemma G.5.** *When $\Phi$ is full column rank, the PFQI update can also be written as:*

$$\theta_{k+1} = \left(\gamma\Sigma_{cov}^{-1}\Sigma_{cr} + (I - \alpha\Sigma_{cov})^t(I - \gamma\Sigma_{cov}^{-1}\Sigma_{cr})\right)\theta_k \\ + (I - (I - \alpha\Sigma_{cov})^t)\Sigma_{cov}^{-1}\theta_{\phi,r}. \tag{155}$$

*Proof.* As we know that when $\Phi$ is full column rank, $\Sigma_{cov} = \Phi^\top D\Phi$ is full rank. Therefore, by Fact G.6 we know that

$$\alpha\sum_{i=0}^{t-1}(I - \alpha\Sigma_{cov})^i = \left(I - (I - \alpha\Sigma_{cov})^t\right)\Sigma_{cov}^{-1}.$$

Then, we plug this into the PFQI update:

$$\theta_{k+1} = \left[I - \alpha\sum_{i=0}^{t-1}(I - \alpha\Sigma_{cov})^i(\Sigma_{cov} - \gamma\Sigma_{cr})\right]\theta_k + \alpha\sum_{i=0}^{t-1}(I - \alpha\Sigma_{cov})^i\theta_{\phi,r} \tag{156}$$

$$= \left[I - \left(I - (I - \alpha\Sigma_{cov})^t\right)\Sigma_{cov}^{-1}(\Sigma_{cov} - \gamma\Sigma_{cr})\right]\theta_{k+1} + \left(I - (I - \alpha\Sigma_{cov})^t\right)\Sigma_{cov}^{-1}\theta_{\phi,r} \tag{157}$$

$$= \left[\gamma\Sigma_{cov}^{-1}\Sigma_{cr} + (I - \alpha\Sigma_{cov})^t(I - \gamma\Sigma_{cov}^{-1}\Sigma_{cr})\right]\theta_k + \left(I - (I - \alpha\Sigma_{cov})^t\right)\Sigma_{cov}^{-1}\theta_{\phi,r}. \tag{158}$$

$\square$

**Fact G.6.** For a square matrix $T$ and a positive integer $n$, the geometric series of matrices is defined as:

$$S_n := \sum_{k=0}^{n-1} T^k. \tag{159}$$

Assuming that $I - T$ is invertible (where $I$ is the identity matrix of the same dimension as $T$), the sum of the geometric series can be expressed as

$$S_n = (I - T^n)(I - T)^{-1} = (I - T)^{-1}(I - T^n). \tag{160}$$

This is implied by Lemma G.8.

**Lemma G.7.** *Given three square matrices $A, B, C \in \mathbb{R}^{n \times n}$, if $A$ commutes with $B$ and $C$ then, $A$ also commutes with $B + C$.*

*Proof.* If $A$ commutes with $B$ and $C$, this means $AB = BA$ and $AC = CA$. Therefore $A(B+C) = AB + AC = BA + CA = (B + C)A$. $\square$

**Lemma G.8.** *Given a square matrix $A \in \mathbb{C}^{n \times n}$, $(I - A^i)$ and $(I - A)^{-1}$ commute for any $i \in \mathbb{N}$.*

*Proof.* For any $t \geq 1$:
$$\sum_{i=0}^{t-1} A^i (I - A) = I - A^t,$$

so $\sum_{i=0}^{t-1} A^i = (I - A^t)(I - A)^{-1}$. Next, we also have:

$$(I - A) \sum_{i=0}^{t-1} A^i = I - A^t,$$

so $\sum_{i=0}^{t-1} A^i = (I - A)^{-1}(I - A^t)$. Therefore, we know:

$$(I - A^t)(I - A)^{-1} = (I - A)^{-1}(I - A^t).$$

Thus, $(I - A^t)$ and $(I - A)$ commute. $\square$

**Theorem G.9** (Restatement of Theorem 8.3). *When the target linear system is nonsingular (satisfying Condition 4.4), the following statements are equivalent:*

1. *FQI converges from any initial point $\theta_0$.*

2. *For any learning rate $\alpha \in \left(0, \frac{2}{\lambda_{max}(\Sigma_{cov})}\right)$, there exists an integer $T \in \mathbb{Z}^+$ such that for $t \geq T$, PFQI converges from any initial point $\theta_0$.*

*Proof.* First, since Proposition 8.2 has proven that under linearly independent features (Condition 4.3), Item 2 implies Item 1, and Condition 4.4 implies Condition 4.3, therefore, under Condition 4.4, Item 2 implies Item 1. Second, from Corollary D.6 we know that FQI converges from any initial point $\theta_0$ if and only if $\rho\left(\gamma \Sigma_{cov}^{-1} \Sigma_{cr}\right) < 1$. Next, for any learning rate $\alpha \in \left(0, \frac{2}{\lambda_{max}(\Sigma_{cov})}\right)$, $\rho\left(I - \alpha \Sigma_{cov}\right) < 1$, so

$$\lim_{t \to \infty} ((I - \alpha \Sigma_{cov})^t (I - \gamma \Sigma_{cov}^{-1} \Sigma_{cr})) = 0.$$

Therefore, by the theorem of continuity of eigenvalues[22, Theorem 5.1] we know that if $\rho\left(\gamma \Sigma_{cov}^{-1} \Sigma_{cr}\right) < 1$, then there must exist an integer $T \in \mathbb{Z}^+$ such that for all $t \geq T$:

$$\rho\left(\gamma \Sigma_{cov}^{-1} \Sigma_{cr} + (I - \alpha \Sigma_{cov})^t (I - \gamma \Sigma_{cov}^{-1} \Sigma_{cr})\right) < 1.$$

In this case, by Corollary F.5, we know PFQI converges from any initial point $\theta_0$. Therefore, Item 1 implies Item 2.

The proof is complete.

$\square$

### G.3 Convergence of TD and FQI: no mutual implication

**TD converges while FQI diverges**  Consider a system with $|\mathcal{S} \times \mathcal{A}| = 3$, $d = 2$, and $\gamma = 0.8$, where the feature matrix $\Phi$, the state-action distribution $\mathbf{D}$, and the transition dynamics $\mathbf{P}_\pi$ are defined as follows:

$$\Phi = \begin{pmatrix} 0.1 & 0.1 \\ 0.8 & 0.2 \\ 0.8 & 0.4 \end{pmatrix}, \quad \mathbf{D} = \begin{pmatrix} 0.7 & 0 & 0 \\ 0 & 0.1 & 0 \\ 0 & 0 & 0.2 \end{pmatrix}, \quad \mathbf{P}_\pi = \begin{pmatrix} 0 & 1 & 0 \\ 0.5 & 0 & 0.5 \\ 0.7 & 0.2 & 0.1 \end{pmatrix}.$$

In this system, the matrix $\Phi^\top \mathbf{D}(I - \gamma \mathbf{P}_\pi)\Phi$ has two distinct, positive eigenvalues, $0.09385551$ and $0.01006449$, indicating that it is nonsingular and positive stable. Therefore, by Corollary E.15, TD is stable. On the other hand, $\gamma \left(\Phi^\top \mathbf{D}\Phi\right)^{-1} \Phi^\top \mathbf{D}\Phi \approx 1.011068 > 1$, and from Corollary D.6, this implies that FQI diverges.

**FQI converges while TD diverges**  Now, consider a different system, again with $|\mathcal{S} \times \mathcal{A}| = 3$, $d = 2$, and $\gamma = 0.8$, where the feature matrix $\Phi$, the state-action distribution $\mathbf{D}$, and the transition dynamics $\mathbf{P}_\pi$ are defined as follows:

$$\Phi = \begin{pmatrix} 0.1 & 0.2 \\ 0.6 & 0.3 \\ 0.7 & 1.0 \end{pmatrix}, \quad \mathbf{D} = \begin{pmatrix} 0.2 & 0 & 0 \\ 0 & 0.7 & 0 \\ 0 & 0 & 0.1 \end{pmatrix}, \quad P = \begin{pmatrix} 0.1 & 0.3 & 0.6 \\ 0.1 & 0.2 & 0.7 \\ 0.1 & 0.1 & 0.8 \end{pmatrix}.$$

In this case, the matrix $\Phi^\top \mathbf{D}(I - \gamma \mathbf{P}_\pi)\Phi$ has two complex eigenvalues, $-0.00056 + 0.02484586i$ and $-0.00056 - 0.02484586i$, which shows that it is nonsingular but not positive semi-stable. Therefore, by Corollary E.15, TD diverges. Meanwhile, $\gamma \left(\Phi^\top \mathbf{D}\Phi\right)^{-1} \Phi^\top \mathbf{D}\Phi \approx 0.94628 < 1$, and from Corollary D.6, we know that FQI converges.

## H  TD and FQI in a Z-matrix system

In the previous section, we showed that the convergence of TD and FQI do not necessarily imply each other, even when the target linear system is nonsingular. A natural question arises: Under what conditions does the convergence of one algorithm imply the convergence of the other? In this section, we investigate the conditions under which such mutual implications hold.

**Assumption H.1.**  [Z-matrix System]

$$(1)\, (\Sigma_{cov} - \gamma \Sigma_{cr}) \text{ is a Z-matrix} \quad (2)\Sigma_{cov}^{-1} \geqq 0 \quad (3)\Sigma_{cov}^{-1}\Sigma_{cr} \geqq 0 \tag{161}$$

First, we will introduce Assumption H.1, which essentially requires preserving certain properties from the system's dynamics: $\mathbf{D}(I - \gamma \mathbf{P}_\pi)$ and its components, $\mathbf{D}$ and $\mathbf{P}_\pi$. Assumption H.1 is composed of two parts: First, $A_{\text{LSTD}}(= \Sigma_{cov} - \gamma \Sigma_{cr})$ is a Z-matrix; second, $\Sigma_{cov}^{-1} \geqq 0$ and $\Sigma_{cov}^{-1}\Sigma_{cr} \geqq 0$, which means that $\Sigma_{cov}$ and $\Sigma_{cr}$ form a weak regular splitting of $(\Sigma_{cov} - \gamma \Sigma_{cr})$. Given these matrices' decomposed forms:

$$\Sigma_{cov} - \gamma \Sigma_{cr} = \Phi^\top \mathbf{D}(I - \gamma \mathbf{P}_\pi)\Phi, \quad \Sigma_{cov} = \Phi^\top \mathbf{D}\Phi, \quad \Sigma_{cr} = \Phi^\top \mathbf{D}\mathbf{P}_\pi \Phi,$$

examining the components between $\Phi^\top$ and $\Phi$ in each matrix reveals something interesting: First, $\mathbf{D}(I - \gamma \mathbf{P}_\pi)$ from $\left(\Phi^\top \mathbf{D}(I - \gamma \mathbf{P}_\pi)\Phi\right)$ is a Z-matrix (proven in Proposition E.9), and second, $\mathbf{D}$ and $(\gamma \mathbf{D}\mathbf{P}_\pi)$ form a weak regular splitting of $[\mathbf{D}(I - \gamma \mathbf{P}_\pi)]$. Essentially, Assumption H.1 requires that these properties be preserved when the matrices are used as coefficient matrices in the matrix quadratic form where $\Phi$ is the variable matrix.

**Theorem H.2.**  *Under Assumption H.1 and rank invariance (Condition 4.1), the following statements are equivalent:*

1. *TD is stable.*

2. *FQI converges for any initial point $\theta_0$.*

Theorem H.2 shows that when Assumption H.1 and rank invariance (Condition 4.1) are satisfied, the convergence of either TD or FQI implies the convergence of the other. The intuition behind this

equivalence in convergence is that when Assumption H.1 and rank invariance (Condition 4.1) hold, the target linear system is a nonsingular Z-matrix system, and the matrix splitting scheme FQI uses to formulate its preconditioner and iterative components is both a weak regular splitting and a proper splitting. In such cases, from the convergence of either TD or FQI, we can deduce that target linear system is a nonsingular M-matrix system (where $A_{\text{LSTD}}$ is nonsingular M-matrix), which is naturally positive stable (TD is stable) and whose every weak regular splitting is convergent (FQI converges). Overall, from above we see that under the Z-matrix System(Assumption H.1) and rank invariance (Condition 4.1), the convergence of TD and FQI imply each other:

$$\text{TD is stable} \Leftrightarrow \text{FQI converges}$$

## H.1 Feature correlation reversal

First, let us denote each column of the feature matrix $\Phi$ as $\varphi_i$, where $i$ represents the index of that feature. For a feature matrix with $d$ features, the columns are: $\varphi_1, \varphi_2, \varphi_3, ..., \varphi_d$. Each $\varphi_i$ represents the $i$-th feature across all state-action pair. We call $\varphi_i$ the *feature basis vector*, which is distinct from the feature vector $\phi(s, a)$ that forms a row of $\Phi$.

Assumption H.4 presents an interesting scenario where the transition dynamics ($\mathbf{P}_\pi$) can reverse the correlation between different feature basis vectors, and importantly, it satisfies the Z-matrix System(Assumption H.1). More specifically: First, $\Sigma_{cov} = \Phi^\top \mathbf{D} \Phi$ being a nonsingular Z-matrix means that the feature basis vectors are linearly independent (i.e., $\Phi$ is full column rank). Moreover, after these vectors are reweighted by the sampling distribution, any reweighted feature basis vector has nonpositive correlation with any other original (unreweighted) feature basis vector, i.e., $\forall i \neq j, \varphi_i^\top \mathbf{D} \varphi_j \leq 0$. Second, $\Sigma_{cr} = \Phi^\top \mathbf{D} \mathbf{P}_\pi \Phi \geqq 0$ means that $\mathbf{P}_\pi$ can reverse these nonpositive correlations to nonnegative correlations, i.e., $\forall i \neq j, \varphi_i^\top \mathbf{D} \mathbf{P}_\pi \varphi_j \geq 0$. Under this scenario, as shown in Proposition H.3, Assumption H.1 is satisfied, and consequently, all previously established results apply to this case.

**Proposition H.3.** *If Assumption H.4 holds, then Assumption H.1 also holds.*

**Assumption H.4.** [Feature Correlation Reversal]

$$(1)\Sigma_{cov} \text{ is nonsingular Z-matrix} \quad (2)\Sigma_{cr} \geqq 0 \tag{162}$$

## H.2 Proof of Theorem H.2

*Proof.* Under Assumption H.1 and rank invariance (Condition 4.1), $(\Sigma_{cov} - \gamma\Sigma_{cr})$ is a Z-matrix, $\Sigma_{cov}^{-1} \geqq 1$ and $\Sigma_{cov}^{-1}\Sigma_{cr} \geqq 0$. Then by definition, $\Sigma_{cov}$ and $\gamma\Sigma_{cr}$ form a weak regular splitting of $(\Sigma_{cov} - \gamma\Sigma_{cr})$, and by Proposition 4.5, $(\Sigma_{cov} - \gamma\Sigma_{cr})$ is a nonsingular matrix.

TD is stable$\Rightarrow$FQI converges: When TD is stable, by Corollary E.15 we know that $(\Sigma_{cov} - \gamma\Sigma_{cr})$ is positive semi-stable. Since $(\Sigma_{cov} - \gamma\Sigma_{cr})$ is also a Z-matrix, by [5, Chapter 6, Theorem 2.3, G20] we know that $(\Sigma_{cov} - \gamma\Sigma_{cr})$ is a nonsingular M-matrix. Therefore, since $\Sigma_{cov}$ and $\gamma\Sigma_{cr}$ form a weak regular splitting of $(\Sigma_{cov} - \gamma\Sigma_{cr})$, by the property of nonsingular M-matrix[5, Chapter 6, Theorem 2.3, O47], every weak regular splitting is convergent, so $\rho\left(\gamma\Sigma_{cov}^{-1}\Sigma_{cr}\right) < 1$. Then, by Corollary D.6, we know that FQI converges for any initial point $\theta_0$.

FQI converges$\Rightarrow$TD is stable: Assume FQI converges. By Corollary D.6 we know that $\rho(\gamma\Sigma_{cov}^{-1}\Sigma_{cr}) < 1$. As $\Sigma_{cov}$ and $\gamma\Sigma_{cr}$ form a weak regular splitting of $(\Sigma_{cov} - \gamma\Sigma_{cr})$ and $(\Sigma_{cov} - \gamma\Sigma_{cr})$ is Z-matrix, by [5, Chapter 5, Theorem 2.3, N46], $(\Sigma_{cov} - \gamma\Sigma_{cr})$ is a nonsingular M-matrix. By the property of nonsingular M-matrix[5, Chapter 6, Theorem 2.3, G20], $(\Sigma_{cov} - \gamma\Sigma_{cr})$ is positive stable. Then, by Corollary D.6, we know TD is stable.

The proof is complete.

$\square$

## H.3 Proof of Proposition H.3

*Proof.* When Assumption H.4 holds, $\Sigma_{cov}$ is a nonsingular Z-matrix, and $\Sigma_{cr} \geqq 0$. Since $\Sigma_{cov}$ is also symmetric positive definite, by Berman and Plemmons [5, Chapter 6, Page 156, 4.15], we know that $\Sigma_{cov}$ is a nonsingular M-matrix. Moreover, by the property of nonsingular M-matrix[5, Chapter 6, Theorem 2.3, N38], we know that $\Sigma_{cov}^{-1} \geqq 0$. Together with $\Sigma_{cr} \geqq 0$, this implies:

First, $(\Sigma_{cov} - \gamma\Sigma_{cr})$ has nonpositive off-diagonal entries, which means $(\Sigma_{cov} - \gamma\Sigma_{cr})$ is Z-matrix. Second, $\Sigma_{cov}^{-1}\Sigma_{cr} \geqq 0$. Therefore, Assumption H.1 is satisfied. $\qquad\square$

# I   Corrections to previous results

**Section 2.2 of Ghosh and Bellemare [17]**   The paper claims that in the off-policy setting and assuming linearly independent features when TD has a fixed point, that fixed point is unique, citing Lagoudakis and Parr [23]. This result is used throughout their paper. However, Lagoudakis and Parr [23] does not actually provide such a result, and this claim does not necessarily hold. More specifically, as we show in Section 4, the fixed point is unique if and only if both linearly independent features and rank invariance hold, where rank invariance is a stricter condition than target linear system being consistent (which is equivalent to the existence of a fixed point). Therefore, when TD has a fixed point (target linear system is consistent) and linearly independent features holds, the fixed point is not necessarily unique since the target linear system being consistent does not imply rank invariance. It is aslo worth mentioning that in the on-policy setting with linearly independent features, when TD has a fixed point, that fixed point is unique, as we demonstrate in Section 4.1.

**Proposition 3.1 of Ghosh and Bellemare [17]**   It is a only sufficient but not necessary condition. Specifically, the proposition states that, assuming $\Phi$ is full column rank, TD is stable if and only if $\left(\Phi^\top \mathbf{D}(I - \gamma\mathbf{P}_\pi)\Phi\right)$ is positive stable. As interpreted in this paper, while positive stability of $\left(\Phi^\top \mathbf{D}(I - \gamma\mathbf{P}_\pi)\Phi\right)$ is indeed a sufficient condition, it is not strictly necessary.

In Proposition E.13, we establish that, under the assumption that $\Phi$ is full column rank, TD is stable if and only if the following three conditions are satisfied:

1. The system is consistent, i.e., $\left(\Phi^\top \mathbf{D}R\right) \in \mathrm{Col}\left(\Phi^\top \mathbf{D}(I - \gamma\mathbf{P}_\pi)\Phi\right)$.
2. $\left[\Phi^\top \mathbf{D}(I - \gamma\mathbf{P}_\pi)\Phi\right]$ is positive semi-stable.
3. $\mathbf{Index}\left(\Phi^\top \mathbf{D}(I - \gamma\mathbf{P}_\pi)\Phi\right) \leq 1$.

If $\left(\Phi^\top \mathbf{D}(I - \gamma\mathbf{P}_\pi)\Phi\right)$ is positive stable, then $\left[\Phi^\top \mathbf{D}(I - \gamma\mathbf{P}_\pi)\Phi\right]$ is necessarily positive semi-stable and nonsingular. As shown in Section 4, any nonsingular linear system must be consistent; hence, the nonsingularity of $\left(\Phi^\top \mathbf{D}(I - \gamma\mathbf{P}_\pi)\Phi\right)$ ensures that $\left(\Phi^\top \mathbf{D}R\right) \in \mathrm{Col}\left(\Phi^\top \mathbf{D}(I - \gamma\mathbf{P}_\pi)\Phi\right)$ holds. By definition, this also implies $\mathbf{Index}\left(\Phi^\top \mathbf{D}(I - \gamma\mathbf{P}_\pi)\Phi\right) = 0$, satisfying the condition $\mathbf{Index}\left(\Phi^\top \mathbf{D}(I - \gamma\mathbf{P}_\pi)\Phi\right) \leq 1$. Therefore, the positive stability of $\left(\Phi^\top \mathbf{D}(I - \gamma\mathbf{P}_\pi)\Phi\right)$ guarantees TD stability.

However, the three conditions in Proposition E.13 reveal that TD can still be stable when $\left(\Phi^\top \mathbf{D}(I - \gamma\mathbf{P}_\pi)\Phi\right)$ is singular and not strictly positive stable. Therefore, while positive stability of $\left(\Phi^\top \mathbf{D}(I - \gamma\mathbf{P}_\pi)\Phi\right)$ is a sufficient condition for TD stability, it is not a necessary one.

**Corollary 2 of Asadi et al. [1]**   It is a only sufficient but not necessary condition. In the context of our paper, their Corollary 2 states that, given $\Phi$ has full column rank, FQI ("Value Function Optimization with Exact Updates" in their paper) converges for any initial point if and only if $\rho\left(\gamma\Sigma_{cov}^{-1}\Sigma_{cr}\right) < 1$. In Proposition D.3, we demonstrate that, given $\Phi$ has full column rank, FQI converges for any initial point if and only if the following two conditions are met: (1) the target linear system must be consistent, i.e., $\theta_{\phi,r} \in \mathrm{Col}\left(\Sigma_{cov} - \gamma\Sigma_{cr}\right)$, and (2) $\left(\gamma\Sigma_{cov}^{-1}\Sigma_{cr}\right)$ is semiconvergent. When $\rho\left(\gamma\Sigma_{cov}^{-1}\Sigma_{cr}\right) < 1$, it implies that $\left(\gamma\Sigma_{cov}^{-1}\Sigma_{cr}\right)$ is semiconvergent and that $\left(I - \gamma\Sigma_{cov}^{-1}\Sigma_{cr}\right)$ is nonsingular, as it has no eigenvalue equal to 1 (see Lemma A.1). Since $\Sigma_{cov}$ is full rank, it follows that $\Sigma_{cov}(I - \gamma\Sigma_{cov}^{-1}\Sigma_{cr}) = \Sigma_{cov} - \gamma\Sigma_{cr}$ is also full rank, ensuring the consistency of the system, i.e., $\theta_{\phi,r} \in \mathrm{Col}\left(\Sigma_{cov} - \gamma\Sigma_{cr}\right)$. Therefore, $\rho\left(\gamma\Sigma_{cov}^{-1}\Sigma_{cr}\right) < 1$ is indeed a sufficient condition for convergence. However, as we show, $\left(\gamma\Sigma_{cov}^{-1}\Sigma_{cr}\right)$ being semiconvergent, according to Definition A.7, does not necessarily imply that $\rho\left(\gamma\Sigma_{cov}^{-1}\Sigma_{cr}\right) < 1$. Thus, while $\rho\left(\gamma\Sigma_{cov}^{-1}\Sigma_{cr}\right) < 1$ is a sufficient condition for FQI convergence, it is not a necessary condition.

**Theorem 2 and Theorem 3 of Xiao et al. [44]**   In Theorem 2, Xiao et al. [44] study the convergence of Temporal Difference (TD) learning with over-parameterized linear approximation, assuming that the state's feature representations are linearly independent. The paper proposes a condition claimed

to be both necessary and sufficient for the convergence of TD. However, the proposed condition is flawed and does not hold as either sufficient or necessary due to errors in the proof. Specifically, between equations (51) and (53), it is claimed that for a non-symmetric matrix, $\|\boldsymbol{W}\| < 1$ implies: "Given $\|\boldsymbol{W}\| < 1/\gamma$, all eigenvalues of $\boldsymbol{I}_k - \gamma\boldsymbol{W}$ are positive." This claim is incorrect, as we can only guarantee that the eigenvalues of $\boldsymbol{I}_k - \gamma\boldsymbol{W}$ have positive real parts, not that they are strictly positive.

Additionally, the matrix $\eta\left(\boldsymbol{I}_k - \gamma\boldsymbol{W}\right)\boldsymbol{M}\boldsymbol{M}^\top\boldsymbol{D}_k$ is not generally symmetric positive definite, as its eigenvalues can be negative or have an imaginary part. Consequently, the condition $\left\|\eta\left(\boldsymbol{I}_k - \gamma\boldsymbol{W}\right)\boldsymbol{M}\boldsymbol{M}^\top\boldsymbol{D}_k\right\| < 1$ does not necessarily imply that the matrix power series $\sum_{i=0}^{t}\left(\boldsymbol{I}_k - \eta\left(\boldsymbol{I}_k - \gamma\boldsymbol{W}\right)\boldsymbol{M}\boldsymbol{M}^\top\boldsymbol{D}_k\right)^i$ converges, and vice versa.

In Theorem 3, Xiao et al. [44] also attempts to analyze the convergence of Fitted Value Iteration (FVI) in the same setting, providing a condition claimed to be both necessary and sufficient. However, the paper does not provide a proof for it being a necessary condition, and as we demonstrate, while the condition is sufficient, it is not necessary for convergence.

**Proposition 3.1 of Che et al. [11]**   In Proposition 3.1, the paper claims that the convergence of TD in their overparameterized setting ($d > k$) can be guaranteed under two conditions. One of them is $\rho\left(I - \eta M^\top D_k(M - \gamma N)\right) < 1$, where $M \in \mathbb{R}^{k \times d}$ and $N \in \mathbb{R}^{k \times d}$. Since $d > k$, we know that $M^\top D_k(M - \gamma N)$ is a singular matrix. Then, by Lemma A.1, we know that $\left(I - \eta M^\top D_k(M - \gamma N)\right)$ must have an eigenvalue equal to 1, which contradicts the condition $\rho\left(I - \eta M^\top D_k(M - \gamma N)\right) < 1$. Therefore, this condition can never hold. [19]

# J   Over-parameterized setting

## J.1   Consistency and nonsingularity in the over-parameterized setting

**Consistency**   Condition J.1 describes an over-parameterized setting in which the number of features is greater than or equal to the number of distinct state-action pairs ($h \le d$), and each state-action pair is represented by a different, linearly independent feature vector (row in $\Phi$). It is completely different from linearly independent features, which means full column rank of $\Phi$. Condition J.1 implies rank invariance. Therefore, it also implies the target linear system is universally consistent. In this case, the existence of a fixed point is guaranteed for these iterative algorithms that solve the target linear system. However, in the over-parameterized setting rank invariance does not necessary hold without Condition J.1.

**Nonsingularity**   For the nonsingularity of the target linear system under the over-parameterized setting, when $h = d$, it can still be guaranteed if linearly independent features (Condition 4.3) holds. however, in the case of $h < d$, the linearly independent features (Condition 4.3) condition can never be satisfied, and thus nonsingularity—and consequently the uniqueness of the solution—is impossible.

**Condition J.1** (Linearly Independent State-Action Feature Vectors).  $\Phi$ is full row rank.

**Proposition J.2.** *If $\Phi$ has full row rank (satisfying Condition 4.3), then rank invariance (Condition 4.1) holds and the target linear system is universally consistent.*

### J.1.1   Proof of Proposition J.2

*Proof.* Since $\Phi$ is full row rank, we know that $\mathrm{Rank}\left(\Phi\right) = h$ and
$$\mathrm{Col}\left(\Phi\right) = \mathbb{R}^h$$
therefore, $\mathrm{Col}\left(\mathbf{P}_\pi\Phi\right) \subseteq \mathrm{Col}\left(\Phi\right)$. By Lemma J.3 we know that $\mathrm{Col}\left(\mathbf{P}_\pi\Phi\right) \subseteq \mathrm{Col}\left(\Phi\right)$ implies
$$\mathrm{Rank}\left(\Sigma_{cov}\right) = \mathrm{Rank}\left(\Sigma_{cov} - \gamma\Sigma_{cr}\right).$$

Hence, the proof is complete.   □

---

[19]We expect that this will be corrected in the arXiv version of the paper. (Personal communication with Che et al., October 2025)

**Lemma J.3.** *If* $\mathrm{Col}\left(\mathbf{P}_\pi \Phi\right) \subseteq \mathrm{Col}\left(\Phi\right)$, *then*

$$\mathrm{Rank}\left(\Sigma_{cov}\right) = \mathrm{Rank}\left(\Sigma_{cov} - \gamma \Sigma_{cr}\right).$$

*However, the converse does not necessarily hold.*

*Proof.* First, assuming $\mathrm{Col}\left(\mathbf{P}_\pi \Phi\right) \subseteq \mathrm{Col}\left(\Phi\right)$, by Lemma J.4, we know that

$$\mathrm{Col}\left(\Phi\right) = \mathrm{Col}\left(\Phi - \gamma \mathbf{P}_\pi \Phi\right)$$

holds. Then by Lemma J.5, we know that

$$\mathrm{Col}\left(\mathbf{D}^{\frac{1}{2}}\Phi\right) = \mathrm{Col}\left(\mathbf{D}^{\frac{1}{2}}(I - \gamma \mathbf{P}_\pi)\Phi\right).$$

Subsequently, by Lemma C.1 we know that $\mathrm{Rank}\left(\Phi^\top \mathbf{D}(I - \gamma \mathbf{P}_\pi)\Phi\right) = \mathrm{Rank}\left(\Phi\right)$ if and only if

$$\mathrm{Ker}\left(\Phi^\top \mathbf{D}^{\frac{1}{2}}\right) \cap \mathrm{Col}\left(\mathbf{D}^{\frac{1}{2}}(I - \gamma \mathbf{P}_\pi)\Phi\right).$$

Next, since we have

$$\mathrm{Col}\left(\mathbf{D}^{\frac{1}{2}}(I - \gamma \mathbf{P}_\pi)\Phi\right) = \mathrm{Col}\left(\mathbf{D}^{\frac{1}{2}}\Phi\right) = \mathrm{Row}\left(\Phi^\top \mathbf{D}^{\frac{1}{2}}\right) \perp \mathrm{Ker}\left(\Phi^\top \mathbf{D}^{\frac{1}{2}}\right),$$

we know $\mathrm{Ker}\left(\Phi^\top \mathbf{D}^{\frac{1}{2}}\right) \cap \mathrm{Col}\left(\mathbf{D}^{\frac{1}{2}}(I - \gamma \mathbf{P}_\pi)\Phi\right) = \{0\}$, therefore,

$$\mathrm{Rank}\left(\Phi^\top \mathbf{D}(I - \gamma \mathbf{P}_\pi)\Phi\right) = \mathrm{Rank}\left(\Phi\right).$$

Second, we will show that $\mathrm{Rank}\left(\Sigma_{cov}\right) = \mathrm{Rank}\left(\Sigma_{cov} - \gamma \Sigma_{cr}\right)$ does not necessarily imply $\mathrm{Col}\left(\mathbf{P}_\pi \Phi\right) \subseteq \mathrm{Col}\left(\Phi\right)$ by demonstrating that

$$\mathrm{Ker}\left(\Phi^\top \mathbf{D}^{\frac{1}{2}}\right) \cap \mathrm{Col}\left(\mathbf{D}^{\frac{1}{2}}(I - \gamma \mathbf{P}_\pi)\Phi\right) = \{0\}$$

does not necessarily imply $\mathrm{Col}\left(\mathbf{P}_\pi \Phi\right) \subseteq \mathrm{Col}\left(\Phi\right)$. This follows from Lemma C.1, which establishes the equivalence between $\mathrm{Ker}\left(\Phi^\top \mathbf{D}^{\frac{1}{2}}\right) \cap \mathrm{Col}\left(\mathbf{D}^{\frac{1}{2}}(I - \gamma \mathbf{P}_\pi)\Phi\right) = \{0\}$ and $\mathrm{Rank}\left(\Sigma_{cov}\right) = \mathrm{Rank}\left(\Sigma_{cov} - \gamma \Sigma_{cr}\right)$.

Assuming that $\mathrm{Ker}\left(\Phi^\top \mathbf{D}^{\frac{1}{2}}\right) \cap \mathrm{Col}\left(\mathbf{D}^{\frac{1}{2}}(I - \gamma \mathbf{P}_\pi)\Phi\right) = \{0\}$ does imply $\mathrm{Col}\left(\mathbf{P}_\pi \Phi\right) \subseteq \mathrm{Col}\left(\Phi\right)$. When $\Phi^\top \mathbf{D}^{\frac{1}{2}}$ doesn't have full column rank:

$$\mathrm{Ker}\left(\Phi^\top \mathbf{D}^{\frac{1}{2}}\right) \neq \{0\}.$$

From Lemma J.4, we know $\mathrm{Ker}\left(\Phi^\top \mathbf{D}^{\frac{1}{2}}\right) \cap \mathrm{Col}\left(\mathbf{D}^{\frac{1}{2}}(I - \gamma \mathbf{P}_\pi)\Phi\right) = \{0\}$ implies

$$\mathrm{Col}\left(\Phi\right) = \mathrm{Col}\left(\Phi - \gamma \mathbf{P}_\pi \Phi\right),$$

which is equal to $\mathrm{Col}\left(\mathbf{D}^{\frac{1}{2}}\Phi\right) = \mathrm{Col}\left(\mathbf{D}^{\frac{1}{2}}(I - \gamma \mathbf{P}_\pi)\Phi\right)$ by Lemma J.5.

Since

$$\mathrm{Row}\left(\Phi^\top \mathbf{D}^{\frac{1}{2}}\right) = \mathrm{Col}\left(\mathbf{D}^{\frac{1}{2}}\Phi\right),$$

we deduce that $\mathrm{Ker}\left(\Phi^\top \mathbf{D}^{\frac{1}{2}}\right) \cap \mathrm{Col}\left(\mathbf{D}^{\frac{1}{2}}(I - \gamma \mathbf{P}_\pi)\Phi\right) = \{0\}$ if and only if

$$\mathrm{Row}\left(\Phi^\top \mathbf{D}^{\frac{1}{2}}\right) = \mathrm{Col}\left(\mathbf{D}^{\frac{1}{2}}(I - \gamma \mathbf{P}_\pi)\Phi\right),$$

which means among all subspaces whose dimension is equal to $\dim\left(\mathrm{Row}\left(\Phi^\top \mathbf{D}^{\frac{1}{2}}\right)\right)$,

$$\mathrm{Row}\left(\Phi^\top \mathbf{D}^{\frac{1}{2}}\right)$$

is only subspace for which

$$\mathrm{Ker}\left(\Phi^\top \mathbf{D}^{\frac{1}{2}}\right) \cap \mathrm{Row}\left(\Phi^\top \mathbf{D}^{\frac{1}{2}}\right) = \{0\}.$$

However, as $\mathrm{Ker}\left(\Phi^\top \mathbf{D}^{\frac{1}{2}}\right) \neq \{0\}$, we know this is impossible as it is contradicted by Lemma J.6. Therefore, we conclude that

$$\mathrm{Ker}\left(\Phi^\top \mathbf{D}^{\frac{1}{2}}\right) \cap \mathrm{Col}\left(\mathbf{D}^{\frac{1}{2}}(I - \gamma \mathbf{P}_\pi)\Phi\right) = \{0\}$$

does not necessarily imply $\mathrm{Col}\left(\mathbf{P}_\pi \Phi\right) \subseteq \mathrm{Col}\left(\Phi\right)$.

$\square$

**Lemma J.4.** $\mathrm{Col}\left(\Phi\right) = \mathrm{Col}\left(\Phi - \gamma \mathbf{P}_\pi \Phi\right)$ *if and only if* $\mathrm{Col}\left(\mathbf{P}_\pi \Phi\right) \subseteq \mathrm{Col}\left(\Phi\right)$.

*Proof.* First, assuming $\mathrm{Col}\left(\Phi\right) = \mathrm{Col}\left(\Phi - \gamma \mathbf{P}_\pi \Phi\right)$, we know that there must exist a matrix $C \in \mathbb{R}^{h \times d}$ such that

$$\Phi C = (I - \gamma \mathbf{P}_\pi)\Phi$$

which is equal to $\gamma \mathbf{P}_\pi \Phi = \Phi(I - C)$. Therefore, the following must hold :

$$\mathrm{Col}\left(\gamma \mathbf{P}_\pi \Phi\right) \subseteq \mathrm{Col}\left(\Phi\right).$$

Next, assuming $\mathrm{Col}\left(\gamma \mathbf{P}_\pi \Phi\right) \subseteq \mathrm{Col}\left(\Phi\right)$, then we know that there must exist a matrix $\bar{C} \in \mathbb{R}^{h \times d}$ such that $\Phi \bar{C} = \gamma \mathbf{P}_\pi \Phi$, and therefore,

$$(I - \gamma \mathbf{P}_\pi)\Phi = \Phi - \Phi \bar{C} = \Phi(I - \bar{C}), \tag{163}$$

which implies $\mathrm{Col}\left((I - \gamma \mathbf{P}_\pi)\Phi\right) \subseteq \mathrm{Col}\left(\Phi\right)$. Subsequently, as $(I - \gamma \mathbf{P}_\pi)$ is full rank and

$$\mathrm{Rank}\left((I - \gamma \mathbf{P}_\pi)\Phi\right) = \mathrm{Rank}\left(\Phi\right),$$

we can get:

$$\mathrm{Col}\left((I - \gamma \mathbf{P}_\pi)\Phi\right) = \mathrm{Col}\left(\Phi\right).$$

From above we know that

$$\mathrm{Col}\left(\Phi\right) = \mathrm{Col}\left(\Phi - \gamma \mathbf{P}_\pi \Phi\right) \Leftrightarrow \mathrm{Col}\left(\gamma \mathbf{P}_\pi \Phi\right) \subseteq \mathrm{Col}\left(\Phi\right).$$

Then, as $\mathrm{Col}\left(\gamma \mathbf{P}_\pi \Phi\right) = \mathrm{Col}\left(\mathbf{P}_\pi \Phi\right)$, we have

$$\mathrm{Col}\left(\Phi\right) = \mathrm{Col}\left(\Phi - \gamma \mathbf{P}_\pi \Phi\right) \Leftrightarrow \mathrm{Col}\left(\mathbf{P}_\pi \Phi\right) \subseteq \mathrm{Col}\left(\Phi\right).$$

$\square$

**Lemma J.5.** *Given two matrices $A \in \mathbb{R}^{n \times m}$ and $B \in \mathbb{R}^{n \times m}$ and a full rank matrix $X \in \mathbb{R}^{n \times n}$, if*

$$\mathrm{Col}\left(XA\right) = \mathrm{Col}\left(XB\right),$$

*then*

$$\mathrm{Col}\left(A\right) = \mathrm{Col}\left(B\right),$$

*and vice versa.*

*Proof.* If $\mathrm{Col}\left(XA\right) = \mathrm{Col}\left(XB\right)$, then there must exist two matrices $V, W \in \mathbb{R}^{m \times m}$ such that

$$XAV = XB, \quad XBW = XA.$$

Since $X$ is invertible, naturally, we have:

$$AV = B, \quad BW = A,$$

which implies respectively: $\mathrm{Col}\left(A\right) \subseteq \mathrm{Col}\left(B\right)$ and $\mathrm{Col}\left(A\right) \supseteq \mathrm{Col}\left(B\right)$. Therefore, we can conclude that $\mathrm{Col}\left(A\right) = \mathrm{Col}\left(B\right)$.

Next, Assuming $\mathrm{Col}\left(A\right) = \mathrm{Col}\left(B\right)$, then there must exist two matrices $\bar{V}, \bar{W} \in \mathbb{R}^{m \times m}$ such that

$$A\bar{V} = B, \quad B\bar{W} = A$$

then for any full rank matrix $X \in \mathbb{R}^{n \times n}$

$$XA\bar{V} = XB, \quad XB\bar{W} = XA$$

which implies respectively: $\mathrm{Col}\left(XA\right) \subseteq \mathrm{Col}\left(XB\right)$ and $\mathrm{Col}\left(XA\right) \supseteq \mathrm{Col}\left(XB\right)$. therefore, we can conclude that $\mathrm{Col}\left(XA\right) = \mathrm{Col}\left(XB\right)$.

Finally, the proof is complete.

$\square$

**Lemma J.6.** *Given any matrix $A \in \mathbb{R}^{n \times m}$ that $\mathrm{Ker}\,(A) \neq \{0\}$, there must exist subspace $W$ that $\dim(W) = \mathrm{Rank}\,(A)$, $W \neq \mathrm{Row}\,(A)$ and $\mathrm{Ker}\,(A) \cap W = \{0\}$.*

*Proof.* Assuming $\mathrm{Rank}\,(A) = r$ and $\mathrm{Row}\,(A) = \{v_1, \cdots, v_r\}$ where $v_1 \cdots v_r$ are $r$ linearly independent vectors which are the basis of $\mathrm{Row}\,(A)$. Since $\mathrm{Ker}\,(A) \neq \{0\}$, we define a nonzero vector $u \in \mathrm{Ker}\,(A)$, and subspace

$$W = \{(v_1 + u), \cdots, (v_r + u)\}.$$

Since $\forall i \in \{1, \cdots, r\}, u \perp v_i$, we know that vectors $(v_1 + u), \cdots, (v_r + u)$ are also linearly independent and

$$\{(v_1 + u), \cdots, (v_r + u)\} \cap \mathrm{Ker}\,(A) = \{0\}$$

, so $\dim(W) = \dim(A)$. Subsequently, we know

$$W \neq \{v_1, \cdots, v_r\} = \mathrm{Row}\,(A),$$

e.g. $v_1 \in \mathrm{Row}\,(A)$ and $v_1 \notin W$. Hence, the proof is complete. $\qquad\square$

## J.2 Over-parameterized FQI

**Linearly independent state-action representation**  In the over-parameterized setting ($h \leq d$), when each distinct state-action pair is represented by linearly independent features vectors (Condition J.1), from Proposition J.2 we know that the target linear system is universally consistent. Furthermore, we can prove that $\rho\left(\gamma\Sigma_{cov}^{\dagger}\Sigma_{cr}\right) < 1$ (see Appendix J.2.1 for details). Consequently, by Corollary 5.3, FQI is guaranteed to converge from any initial point in this setting. And in such setting, the FQI update equation can be simplified as: $\theta_{k+1} = \gamma\Phi^{\dagger}\mathbf{P}_{\pi}\Phi\theta_k + \Phi^{\dagger}R$ (detailed derivation in Lemma A.19)

**Linearly dependent state-action representation**  However, if we relax the assumption of a linearly independent state-action feature representation (Condition J.1) in the same over-parameterized setting ($h \leq d$), the previous conclusion no longer necessarily holds. In this case, FQI is not guaranteed to retain the favorable properties established above for the case of linearly independent state-action feature representation. Consequently, its convergence is *not* necessarily guaranteed, but all results (e.g., Theorem 5.1) that did not assume any specific parameterization remain valid.

### J.2.1  Why is $\rho\left(\gamma\Sigma_{cov}^{\dagger}\Sigma_{cr}\right) < 1$

First, as we know when $\Phi$ is full row rank, $\gamma\Sigma_{cov}^{\dagger}\Sigma_{cr} = \gamma\Phi^{\dagger}\mathbf{P}_{\pi}\Phi$, and by Lemma E.18, we know that $\sigma\left(\gamma\Phi^{\dagger}\mathbf{P}_{\pi}\Phi\right) \backslash \{0\} = \sigma\left(\gamma\Phi\Phi^{\dagger}\mathbf{P}_{\pi}\right) \backslash \{0\}$. Additionally, $\gamma\Phi\Phi^{\dagger}\mathbf{P}_{\pi} = \gamma\mathbf{P}_{\pi}$ as $\Phi$ is full row rank, and $\rho\left(\gamma\mathbf{P}_{\pi}\right) < 1$. Therefore, $\rho\left(\gamma\Sigma_{cov}^{\dagger}\Sigma_{cr}\right) = \rho\left(\gamma\Phi^{\dagger}\mathbf{P}_{\pi}\Phi\right) = \rho\left(\gamma\mathbf{P}_{\pi}\right) < 1$.

## J.3  Over-parameterized PFQI

**Over-parameterized PFQI with linearly independent state action feature vectors**  Corollary J.7 reveals the necessary and sufficient condition for the convergence of PFQI when each state-action pair can be represented by a distinct linearly independent features vector (Condition J.1 is satisfied). In this setting, its preconditioner $M_{\mathrm{PFQI}} = \alpha\sum_{i=0}^{t-1}(I - \alpha\Sigma_{cov})^i$ is not upper bounded as $t$ increases, indicating that $M_{\mathrm{PFQI}}$ will diverge with increasing $t$. However, $M_{\mathrm{PFQI}}A_{\mathrm{LSTD}}$ remains upper bounded as $t$ increases. This is because the divergence in $M_{\mathrm{PFQI}}$ is caused by the redundancy of features rather than the lack of features, and the divergent components in $M_{\mathrm{PFQI}}$ that grow with $t$ are effectively canceled out when $M_{\mathrm{PFQI}}$ is multiplied by $A_{\mathrm{LSTD}}$. For more mathematical details on this process, please see Appendix J.3.1. Leveraging this result, in Proposition J.8, we prove that under this setting, if updates are performed for a sufficiently large number of iterations toward each target value, the convergence of PFQI is guaranteed. Che et al. [11, Proposition 3.3] previously proved the same result as Proposition J.8 using a different proof path. It is worth noting, however, that this proposition does not guarantee PFQI's convergence in all practical batch settings, even for sufficiently large $t$. A detailed explanation is provided in our batch setting section(Appendix K.3).

**Corollary J.7.** *When $\Phi$ is full row rank (Condition J.1 is satisfied) and $\sigma\left(\alpha\Sigma_{cov}\right) \cap \{1, 2\} = \emptyset$, PFQI converges for any initial point $\theta_0$ if and only if $\rho\left(H_{PFQI}\right) = 1$ where the $\lambda = 1$ is only eigenvalue on*

*the unit circle. It converges to*

$$\left(\sum_{i=0}^{t-1}(I-\alpha\Sigma_{cov})^i(\Sigma_{cov}-\gamma\Sigma_{cr})\right)^{\#}\sum_{i=0}^{t-1}(I-\alpha\Sigma_{cov})^i\theta_{\phi,r} \tag{164}$$

$$+\left(I-\left(\sum_{i=0}^{t-1}(I-\alpha\Sigma_{cov})^i(\Sigma_{cov}-\gamma\Sigma_{cr})\right)\left[\sum_{i=0}^{t-1}(I-\alpha\Sigma_{cov})^i(\Sigma_{cov}-\gamma\Sigma_{cr})\right]^{\#}\right)\theta_0 \tag{165}$$

$$\in\Theta_{LSTD}. \tag{166}$$

**Proposition J.8.** *When $\Phi$ is full row rank and $d > h$, for any learning rate $\alpha \in \left(0, \frac{2}{\rho(\Sigma_{cov})}\right)$, there must exist big enough finite $T$ such that for any $t > T$, Partial FQI converges for any initial point $\theta_0$.*

**Over-parameterized PFQI without linearly independent state-action feature vectors** In this over-parameterized setting, our previous results that assumed $\Phi$ to be full row rank no longer apply. However, all results (e.g., Theorem 5.1) that do not rely on any specific parameterization remain valid.

### J.3.1 Why the divergent part in $M_{\textbf{PFQI}}$ can be canceled out when $\Phi$ is full row rank

As we know from Appendix F.3.1, when $\Phi$ is not full column rank, $M_{\text{PFQI}} = \alpha\sum_{i=0}^{t-1}(I-\alpha\Sigma_{cov})^i$ will diverge as $t$ increases. However, when $\Phi$ is full row rank (which also includes the case where $\Phi$ is not full column rank), $(M_{\text{PFQI}}A_{\text{LSTD}})$ becomes:

$$\left(\alpha\sum_{i=0}^{t-1}(I-\alpha\Sigma_{cov})^i(\Sigma_{cov}-\gamma\Sigma_{cr})\right) = \alpha\sum_{i=0}^{t-1}\left(I-\alpha\Phi^\top\mathbf{D}\Phi\right)^i\Phi^\top\mathbf{D}(I-\gamma\mathbf{P}_\pi)\Phi \tag{167}$$

$$= \alpha\Phi^\top\sum_{i=0}^{t-1}\left(I-\alpha\mathbf{D}\Phi\Phi^\top\right)^i\mathbf{D}(I-\gamma\mathbf{P}_\pi)\Phi. \tag{168}$$

In Equation (167), $\left(I-\Phi^\top\mathbf{D}\Phi\right)$ is a singular positive semidefinite matrix. From Appendix F.3.1, we know that $\text{Ker}\left(\Phi^\top\mathbf{D}\Phi\right) \neq \{0\}$, so in $I-\alpha\Phi^\top\mathbf{D}\Phi$, there are components that cannot be reduced by adjusting $\alpha$ (see the mathematical derivation in Appendix F.3.1). These components will accumulate as $t$ increases, causing $M_{\text{PFQI}}$ to diverge. However, when $\Phi$ is full row rank and $M_{\text{PFQI}}$ is multiplied with $A_{\text{LSTD}}$, $\Phi^\top\mathbf{D}\Phi$ can be transformed as $\left(\mathbf{D}\Phi\Phi^\top\right)$ as shown in Equation (168), which is a nonsingular matrix. Thus, $\text{Ker}\left(\mathbf{D}\Phi\Phi^\top\right) = \{0\}$, meaning that by adjusting $\alpha$ we can always control $\rho\left(I-\alpha\mathbf{D}\Phi\Phi^\top\right) < 1$. This also indicates that the previously divergent components are canceled out by $A_{\text{LSTD}}$.

### J.3.2 Proof of Corollary J.7

*Proof.* When $h > d$ and $\Phi$ is full row rank, we know that $\Sigma_{cov}$ and $(\Sigma_{cov} - \gamma\Sigma_{cr})$ are singular matrices and the PFQI update is:

$$\theta_{k+1} = \left(I-\alpha\sum_{i=0}^{t-1}(I-\alpha\Sigma_{cov})^i(\Sigma_{cov}-\gamma\Sigma_{cr})\right)\theta_k + \alpha\sum_{i=0}^{t-1}(I-\alpha\Sigma_{cov})^i\theta_{\phi,r}.$$

From Proposition J.2, we know the target linear system is universal consistent, then by Theorem 7.1 we know that PFQI converges for any initial point $\theta_0$ if and only if

$$\left(I-\alpha\sum_{i=0}^{t-1}(I-\alpha\Sigma_{cov})^i(\Sigma_{cov}-\gamma\Sigma_{cr})\right)$$

is semiconvergent. Since

$$\left(\alpha\sum_{i=0}^{t-1}(I-\alpha\Sigma_{cov})^i(\Sigma_{cov}-\gamma\Sigma_{cr})\right)$$

is singular matrix, by Lemma A.1 we know that

$$\left(I - \alpha \sum_{i=0}^{t-1}(I - \alpha\Sigma_{cov})^i \left(\Sigma_{cov} - \gamma\Sigma_{cr}\right)\right)$$

must have eigenvalue equal to 1. Therefore, by definition of semiconvergent matrix in Definition A.7, we know that PFQI converges for any initial point $\theta_0$ if and only if

$$\rho\left(I - \alpha \sum_{i=0}^{t-1}(I - \alpha\Sigma_{cov})^i \left(\Sigma_{cov} - \gamma\Sigma_{cr}\right)\right) = 1,$$

where the $\lambda = 1$ is only eigenvalue on the unit circle and is semisimple. Next, from Lemma J.9, we know $\mathbf{Index}\left(\alpha \sum_{i=0}^{t-1}(I - \alpha\Sigma_{cov})^i \left(\Sigma_{cov} - \gamma\Sigma_{cr}\right)\right) = 1$, so we have

$$\left(\alpha \sum_{i=0}^{t-1}(I - \alpha\Sigma_{cov})^i \left(\Sigma_{cov} - \gamma\Sigma_{cr}\right)\right)^{\mathrm{D}} = \left(\alpha \sum_{i=0}^{t-1}(I - \alpha\Sigma_{cov})^i \left(\Sigma_{cov} - \gamma\Sigma_{cr}\right)\right)^{\#}.$$

Then, by Lemma E.24 and Lemma A.1 we can get:

$$\lambda = 1 \in \sigma\left(I - \alpha \sum_{i=0}^{t-1}(I - \alpha\Sigma_{cov})^i \left(\Sigma_{cov} - \gamma\Sigma_{cr}\right)\right) \text{ is semisimple.}$$

Therefore, we can conclude that when $h > d$ and $\Phi$ is full row rank and $\sigma\left(\alpha\Sigma_{cov}\right) \cap \{1, 2\} = \emptyset$, PFQI converges for any initial point $\theta_0$ if and only if

$$\rho\left(I - \alpha \sum_{i=0}^{t-1}(I - \alpha\Sigma_{cov})^i \left(\Sigma_{cov} - \gamma\Sigma_{cr}\right)\right) = 1,$$

where the $\lambda = 1$ is only eigenvalue on the unit circle. By Theorem 7.1, it converges to

$$\left(\sum_{i=0}^{t-1}(I - \alpha\Sigma_{cov})^i(\Sigma_{cov} - \gamma\Sigma_{cr})\right)^{\#} \sum_{i=0}^{t-1}(I - \alpha\Sigma_{cov})^i\theta_{\phi,r} \tag{169}$$

$$+ \left(I - \left(\sum_{i=0}^{t-1}(I - \alpha\Sigma_{cov})^i(\Sigma_{cov} - \gamma\Sigma_{cr})\right)\left[\sum_{i=0}^{t-1}(I - \alpha\Sigma_{cov})^i(\Sigma_{cov} - \gamma\Sigma_{cr})\right]^{\#}\right)\theta_0 \tag{170}$$

$$\in \Theta_{\mathrm{LSTD}}. \tag{171}$$

$\square$

**Lemma J.9.** *When $h > d$, $\Phi$ is full row rank and $\sigma\left(\alpha\Sigma_{cov}\right) \cap \{1, 2\} = \emptyset$ and $\Phi$ is full row rank, then*

$$\mathbf{Index}\left(\alpha \sum_{i=0}^{t-1}(I - \alpha\Sigma_{cov})^i \left(\Sigma_{cov} - \gamma\Sigma_{cr}\right)\right) = 1.$$

*Proof.* First, we have

$$\left(\alpha \sum_{i=0}^{t-1}(I - \alpha\Sigma_{cov})^i \left(\Sigma_{cov} - \gamma\Sigma_{cr}\right)\right) = \alpha \sum_{i=0}^{t-1}\left(I - \alpha\Phi^\top\mathbf{D}\Phi\right)^i \Phi^\top\mathbf{D}(I - \gamma\mathbf{P}_\pi)\Phi \tag{172}$$

$$= \alpha\Phi^\top \sum_{i=0}^{t-1}\left(I - \alpha\mathbf{D}\Phi\Phi^\top\right)^i \mathbf{D}(I - \gamma\mathbf{P}_\pi)\Phi. \tag{173}$$

As we know that $\alpha\Phi^\top\mathbf{D}\Phi$ is a singular matrix, $\alpha\mathbf{D}\Phi\Phi^\top$ is a nonsingular matrix and $\sigma\left(\alpha\Phi^\top\mathbf{D}\Phi\right) \cap \{1, 2\} = \emptyset$. By Lemma E.18 we can obtain that

$$\sigma\left(\alpha\Phi^\top\mathbf{D}\Phi\right) \setminus \{0\} = \sigma\left(\alpha\mathbf{D}\Phi\Phi^\top\right),$$

which implies $\sigma\left(\alpha\mathbf{D}\Phi\Phi^\top\right)\cap\{1,2\}=\emptyset$. By Lemma J.10, we know

$$\sum_{i=0}^{t-1}(I-\alpha\mathbf{D}\Phi\Phi^\top)^i$$

is a full rank matrix, and subsequently,

$$\left(\sum_{i=0}^{t-1}(I-\alpha\mathbf{D}\Phi\Phi^\top)^i\mathbf{D}(I-\gamma\mathbf{P}_\pi)\right)$$

is a full rank matrix. Together with $\Phi^\top$ being a full column rank matrix, we know that

$$\left(\sum_{i=0}^{t-1}(I-\alpha\mathbf{D}\Phi\Phi^\top)^i\mathbf{D}(I-\gamma\mathbf{P}_\pi)\Phi\Phi^\top\right)$$

is a nonsingular matrix. Therefore, by Lemma E.19, we know that:

$$\mathbf{Index}\left(\alpha\Phi^\top\sum_{i=0}^{t-1}\left(I-\alpha\mathbf{D}\Phi\Phi^\top\right)^i\mathbf{D}(I-\gamma\mathbf{P}_\pi)\Phi\right)=1.$$

Hence, $\mathbf{Index}\left(\alpha\sum_{i=0}^{t-1}(I-\alpha\Sigma_{cov})^i\left(\Sigma_{cov}-\gamma\Sigma_{cr}\right)\right)=1.$ $\qquad\square$

**Lemma J.10.** *Given a nonsingular matrix* $A\in\mathbb{R}^{n\times n}$, *if* $\sigma\left(A\right)\cap\{1,2\}=\emptyset$, $\sum_{i=0}^t\left(I-A\right)^i$ *is nonsingular for any positive integer* $t$.

*Proof.* Given a nonsingular matrix $A\in\mathbb{R}^{n\times n}$, assuming $\sigma\left(A\right)\cap\{1,2\}=\emptyset$, by Lemma A.1 we know $\sigma\left(I-A\right)\cap\{0,1,2\}=\emptyset$. Next, we define the Jordan form of $A$ as

$$QAQ^{-1}=J,$$

where $J$ is full rank upper triangular matrix with nonzero diagonal entries. By Lemma A.1, we know the Jordan form of full rank matrix $(I-A)$ is:

$$Q\left(I-A\right)Q^{-1}=(I-J),$$

where $(I-J)$ is also a full rank upper triangular matrix with no diagonal entries equal to 0, 1 and -1. Therefore, $\forall i\in\mathbb{N},(I-J)^i$ is an full rank upper triangular matrix with no diagonal entries equal to 0 and 1, so $\forall i\in\mathbb{N},\left(I-(I-J)^i\right)$ is nonsingular. Moreover, by Fact G.6 we know that:

$$\sum_{i=0}^t\left(I-A\right)^i=Q\sum_{i=0}^t\left(I-J\right)^iQ^{-1}=Q\left(I-(I-J)^{t+1}\right)J^{-1}Q^{-1},$$

Since $Q,\left(I-(I-J)^{t+1}\right),J$ are all nonsingular, $\sum_{i=0}^t\left(I-A\right)^i$ is nonsingular. $\qquad\square$

### J.3.3 Proof of Proposition J.8

**Proposition J.11** (Restatement of Proposition J.8). *When* $\Phi$ *is full row rank and* $d>h$, *for any learning rate* $\alpha\in\left(0,\frac{2}{\rho(\Sigma_{cov})}\right)$, *there must exist a big enough finite* $T$ *for any* $t>T$, *such that PFQI converges for any initial point* $\theta_0$.

*Proof.*

$$\left(\alpha\sum_{i=0}^{t-1}(I-\alpha\Sigma_{cov})^i\left(\Sigma_{cov}-\gamma\Sigma_{cr}\right)\right)\tag{174}$$

$$=\alpha\sum_{i=0}^{t-1}\left(I-\alpha\Phi^\top\mathbf{D}\Phi\right)^i\Phi^\top\mathbf{D}(I-\gamma\mathbf{P}_\pi)\Phi\tag{175}$$

$$=\alpha\Phi^\top\sum_{i=0}^{t-1}\left(I-\alpha\mathbf{D}\Phi\Phi^\top\right)^i\mathbf{D}(I-\gamma\mathbf{P}_\pi)\Phi\tag{176}$$

$$=\Phi^\top\left(I-(I-\alpha\mathbf{D}\Phi\Phi^\top)^t\right)(\mathbf{D}\Phi\Phi^\top)^{-1}\mathbf{D}(I-\gamma\mathbf{P}_\pi)\Phi\tag{177}$$

$$=\Phi^\top\left(I-(I-\alpha\mathbf{D}\Phi\Phi^\top)^t\right)(\Phi\Phi^\top)^{-1}(I-\gamma\mathbf{P}_\pi)\Phi.\tag{178}$$

By Lemma E.18 we know that:

$$\sigma \left( \Phi^\top \left( I - (I - \alpha \mathbf{D} \Phi \Phi^\top)^t \right) (\Phi \Phi^\top)^{-1} (I - \gamma \mathbf{P}_\pi) \Phi \right) \backslash \{0\} \tag{179}$$

$$= \sigma \left( \Phi \Phi^\top \left( I - (I - \alpha \mathbf{D} \Phi \Phi^\top)^t \right) (\Phi \Phi^\top)^{-1} (I - \gamma \mathbf{P}_\pi) \right), \tag{180}$$

then by Lemma A.1 we know that

$$\sigma \left( I - \Phi^\top \left( I - (I - \alpha \mathbf{D} \Phi \Phi^\top)^t \right) (\Phi \Phi^\top)^{-1} (I - \gamma \mathbf{P}_\pi) \Phi \right) \tag{181}$$

$$= \sigma \left( I - \Phi \Phi^\top \left( I - (I - \alpha \mathbf{D} \Phi \Phi^\top)^t \right) (\Phi \Phi^\top)^{-1} (I - \gamma \mathbf{P}_\pi) \right) \cup \{1\}, \tag{182}$$

and we get that

$$I - \Phi \Phi^\top \left( I - (I - \alpha \mathbf{D} \Phi \Phi^\top)^t \right) (\Phi \Phi^\top)^{-1} (I - \gamma \mathbf{P}_\pi) \tag{183}$$

$$= \gamma \mathbf{P}_\pi + \Phi \Phi^\top (I - \alpha \mathbf{D} \Phi \Phi^\top)^t (\Phi \Phi^\top)^{-1} (I - \gamma \mathbf{P}_\pi). \tag{184}$$

Since $\rho \left( I - \alpha \mathbf{D} \Phi \Phi^\top \right) < 1$, $\lim_{t \to \infty} \left( I - \alpha \mathbf{D} \Phi \Phi^\top \right)^t = 0$, then

$$\lim_{t \to \infty} \left[ \gamma \mathbf{P}_\pi + \Phi \Phi^\top (I - \alpha \mathbf{D} \Phi \Phi^\top)^t (\Phi \Phi^\top)^{-1} (I - \gamma \mathbf{P}_\pi) \right] = \gamma \mathbf{P}_\pi.$$

As we know that $\rho \left( \gamma \mathbf{P}_\pi \right) < 1$, then by the theorem of continuity of eigenvalues[22, Theorem 5.1], we can know that there must be finite positive integer $T$ that for any $t > T$,

$$\rho \left( \gamma \mathbf{P}_\pi + \Phi \Phi^\top (I - \alpha \mathbf{D} \Phi \Phi^\top)^t (\Phi \Phi^\top)^{-1} (I - \gamma \mathbf{P}_\pi) \right) < 1.$$

In that case, we know that

$$\forall \lambda \neq 1 \in \sigma \left( I - \Phi^\top \left( I - (I - \alpha \mathbf{D} \Phi \Phi^\top)^t \right) (\Phi \Phi^\top)^{-1} (I - \gamma \mathbf{P}_\pi) \Phi \right), |\lambda| < 1.$$

Therefore, $\rho \left( I - \Phi^\top \left( I - (I - \alpha \mathbf{D} \Phi \Phi^\top)^t \right) (\Phi \Phi^\top)^{-1} (I - \gamma \mathbf{P}_\pi) \Phi \right) = 1$, and eigenvalue $\lambda = 1$ is only eigenvalue in the unit circle. By Lemma J.9 and Lemma E.24, we know that $\lambda = 1$ is also semisimple. By Definition A.7, we know that

$$\left( I - \Phi^\top \left( I - (I - \alpha \mathbf{D} \Phi \Phi^\top)^t \right) (\Phi \Phi^\top)^{-1} (I - \gamma \mathbf{P}_\pi) \Phi \right) \text{ is semiconvergent.}$$

Additionally, Proposition J.2 shows that $\theta_{\phi,r} \in \text{Col} \left( \Sigma_{cov} - \gamma \Sigma_{cr} \right)$ naturally holds when $\Phi$ is full row rank. By Theorem 7.1, we know that PFQI converges for any initial point $\theta_0$. $\qquad\square$

## J.4 Over-parameterized TD

The results on TD in over-parameterized setting are presented in Appendix E.6.

# K Batch case

Offline policy evaluation is a special but realistic case of the policy evaluation task, where sampling from the environment is not possible. Instead, a collected batch dataset $\{(s_i, a_i, r_i (s_i, a_i), s_i')\}_{i=1}^{\bar{n}}$, comprising $\bar{n}$ samples, is provided. Therefore, this is also referred to as a **batch setting**. In this dataset, we define $(s_i, a_i)$ as the *initial state-action*, sampled from some arbitrary distribution $\mathcal{D}$. The reward is represented as $r_i (s_i, a_i) = R (s_i, a_i)$, and the next state is sampled from the transition model, $s_i' \sim P (\cdot \mid s_i, a_i)$. Since the next action is sampled according to $\pi$, $a_i' \sim \pi (s_i')$, we can express the dataset as $\{(s_i, a_i, r_i (s_i, a_i), s_i', a_i')\}_{i=1}^n$ for clarity of presentation. We refer to $(s_i', a_i')$ as the *next state-action*. Here, the sample number $n \geq \bar{n}$, since usually multiple actions at a single state have a nonzero probability of being sampled.

Let $m$ denote the total number of distinct state-action pairs that appear either as initial state-action pairs or as next state-action pairs in the dataset. Let $n(s, a) = \sum_{i=1}^n \mathbb{I}[s_i = s, a_i = a]$ represent the number of times the state-action pair $(s, a)$ appears as the initial state-action pair in the dataset. For a state-action pair $(s, a)$ that appears as an initial state-action pair, we define $\widehat{\mu}(s, a) = n(s, a)/n$. For state-action pairs $(s, a)$ that appear only as next state-action pairs and not as initial state-action pairs, we set $\widehat{\mu}(s, a) = 0$. Thus, $\widehat{\mu} \in \mathbb{R}^m$ is the vector of empirical sample distributions for all state-action pairs in the dataset. Next, $\widehat{\Phi} \in \mathbb{R}^{m \times d}$ is the *empirical feature matrix*, where each row corresponds to a feature vector $\phi(s, a)$ for a state-action pair $(s, a)$ in the dataset.

The empirical counterparts of the covariance matrix $\Sigma_{cov}$, cross-variance matrix $\Sigma_{cr}$, and feature-reward vector $\theta_{\phi,r}$, as defined are given by:

$$\widehat{\Sigma}_{cov} := \frac{1}{n}\sum_{i=1}^{n}\phi\left(s_i,a_i\right)\phi\left(s_i,a_i\right)^{\top} = \widehat{\Phi}^{\top}\widehat{\mathbf{D}}\widehat{\Phi},$$

$$\widehat{\Sigma}_{cr} := \frac{1}{n}\sum_{i=1}^{n}\phi\left(s_i,a_i\right)\phi\left(s_i',a_i'\right)^{\top} = \widehat{\Phi}^{\top}\widehat{\mathbf{D}}\widehat{\mathbf{P}_{\pi}}\widehat{\Phi}, \tag{185}$$

$$\widehat{\theta}_{\phi,r} := \frac{1}{n}\sum_{i=1}^{n}\phi\left(s_i,a_i\right)r\left(s_i,a_i\right) = \widehat{\Phi}^{\top}\widehat{\mathbf{D}}\widehat{R}.$$

Here, we define the *empirical distribution matrix* $\widehat{\mathbf{D}} = \mathrm{diag}\left(\widehat{\mu}\right)$ as a diagonal matrix whose diagonal entries correspond to the empirical distribution of the state-action pairs. Similarly, $\widehat{R} \in \mathbb{R}^m$ is the vector of rewards for all state-action pairs in the dataset.[20] The empirical transition matrix between state-action pairs, $\widehat{\mathbf{P}_{\pi}} \in \mathbb{R}^{|\mathcal{S}\times\mathcal{A}|\times|\mathcal{S}\times\mathcal{A}|}$, is defined as:

$$\widehat{\mathbf{P}_{\pi}}\left(s',a' \mid s,a\right) = \frac{\sum_{i=1}^{n}\mathbb{I}\left[s_i = s,\, a_i = a,\, s_i' = s',\, a_i' = a'\right]}{n(s,a)}$$

for state-action pairs $(s,a)$ that appear as initial state-action pairs, and $\widehat{\mathbf{P}_{\pi}}\left(s',a' \mid s,a\right) = 0$ for state-action pairs that only appear as next state-action pairs but not as initial state-action pairs. As a result, $\widehat{\mathbf{P}_{\pi}}$ is a sub-stochastic matrix.

It is worth noting that for state-action pairs that appear in the dataset only as next state-action pairs but not as initial state-action pairs, we do not remove their corresponding entries from $\widehat{\Phi}$ when defining $\widehat{\Sigma}_{cov}$, $\widehat{\Sigma}_{cr}$, and $\widehat{\theta}_{\phi,r}$ in Equation (185). Including these state-action pairs does not affect generality, as their interactions with other components are effectively canceled out. For example, in $\widehat{\Sigma}_{cov} = \widehat{\Phi}^{\top}\widehat{\mathbf{D}}\widehat{\Phi}$, their feature vectors in $\widehat{\Phi}$ are nullified by $\widehat{\mathbf{D}}$, since their observed sampling probabilities are zero. However, retaining these entries facilitates analysis. For instance, it ensures that we can model the empirical transition matrix $\widehat{\mathbf{P}_{\pi}}$ as a sub-stochastic square matrix, which has desirable properties, such as $\rho\left(\widehat{\mathbf{P}_{\pi}}\right) \leq 1$, rather than as a rectangular matrix.

**FQI in the batch setting** Given the datset $\{(s_i,a_i,r_i\left(s_i,a_i\right),s_i',a_i')\}_{i=1}^{n}$, with linear function approximation, at every iteration, the update of FQI involves iterative solving a least squares regression problem. The update equation is:

$$\theta_{k+1} = \arg\min_{\theta}\sum_{i=1}^{n}\left(\phi\left(s_i,a_i\right)^{\top}\theta - r\left(s_i,a_i\right) - \gamma\phi\left(s_i',a_i'\right)^{\top}\widehat{\theta}_t\right)^2 \tag{186}$$

$$= \gamma\widehat{\Sigma}_{cov}^{\dagger}\widehat{\Sigma}_{cr}\theta_k + \widehat{\Sigma}_{cov}^{\dagger}\widehat{\theta}_{\phi,r}. \tag{187}$$

**Batch TD** In the batch setting / offline policy evaluation setting, TD uses the entire dataset instead of stochastic samples to update:

$$\theta_{k+1} = \theta_k - \alpha\cdot\frac{1}{n}\sum_{i=1}^{n}\left[\nabla_{\theta_k}Q_{\theta_k}(s,a)\left(Q_{\theta_k}(s,a) - \gamma Q_{\theta_k}(s',a') - r(s,a)\right)\right] \tag{188}$$

$$= \theta_k - \alpha\cdot\frac{1}{n}\sum_{i=1}^{n}\left[\phi(s,a)\left(\phi(s,a)^{\top}\theta_k - \gamma\phi(s',a')^{\top}\theta_k - r(s,a)\right)\right] \tag{189}$$

$$= \theta_k - \alpha\left[\left(\widehat{\Sigma}_{cov} - \widehat{\Sigma}_{cr}\right)\theta_k - \widehat{\theta}_{\phi,r}\right]. \tag{190}$$

---

[20]For state-action pairs whose rewards are not observed, we set their rewards to 0.

### K.1  Extension of FQI convergence results to the batch setting

By replacing $\Phi$, $\mathbf{D}$, $\mathbf{P}_\pi$, $\Sigma_{cov}$, $\Sigma_{cr}$, and $\theta_{\phi,r}$ with their empirical counterparts $\widehat{\Phi}$, $\widehat{\mathbf{D}}$, $\widehat{\mathbf{P}_\pi}$, $\widehat{\Sigma}_{cov}$, $\widehat{\Sigma}_{cr}$, and $\widehat{\theta}_{\phi,r}$ respectively, we can extend the convergence results of expected FQI to Batch FQI. However, the conclusion in Appendix J.2 holds only when $\mathbf{D}$ is a full-rank matrix, but $\widehat{\mathbf{D}}$ is not necessarily full rank, and FQI in the batch setting does not necessarily converge unless $\widehat{\mathbf{D}}$ is a full-rank matrix (even in the over-parameterized setting where $\widehat{\Phi}$ has full row rank). Nevertheless, the batch version of Theorem 5.1 still implies necessary and sufficient condition convergence conditions under these circumstances.

### K.2  Extension of TD convergence results to the batch setting

By replacing $\Phi$, $D$, $\mathbf{P}_\pi$, $\Sigma_{cov}$, $\Sigma_{cr}$ and $\theta_{\phi,r}$ with their empirical counterparts $\widehat{\Phi}$, $\widehat{\mathbf{D}}$, $\widehat{\mathbf{P}_\pi}$, $\widehat{\Sigma}_{cov}$, $\widehat{\Sigma}_{cr}$ and $\widehat{\theta}_{\phi,r}$, respectively, we can extend the convergence results of expected TD to Batch TD[21]. For example, Corollary 6.3, which identifies the specific learning rates that make expected TD converge, is particularly useful for Batch TD. By replacing each matrix with its empirical counterpart, we can determine which learning rates will ensure Batch TD convergence and which will not.

### K.3  Extension of PFQI results to the batch setting

By replacing $\Phi$, $D$, $\mathbf{P}_\pi$, $\Sigma_{cov}$, $\Sigma_{cr}$ and $\theta_{\phi,r}$ with their empirical counterparts $\widehat{\Phi}$, $\widehat{\mathbf{D}}$, $\widehat{\mathbf{P}_\pi}$, $\widehat{\Sigma}_{cov}$, $\widehat{\Sigma}_{cr}$ and $\widehat{\theta}_{\phi,r}$, respectively, we can extend the convergence results of expected PFQI to PFQI in the batch setting, with one exception: Proposition J.8, which relies upon $\mathbf{D}$ being a nonsingular matrix, while $\widehat{\mathbf{D}}$ is not necessarily nonsingular anymore. However, if $\widehat{\mathbf{D}}$ is nonsingular, then Proposition J.8 can apply.

---

[21]While the extension to the on-policy setting is straightforward in principle, in practice when data are sampled from the policy to be evaluated, it is unlikely that $\widehat{\mu}\widehat{\mathbf{P}_\pi} = \widehat{\mu}$ will hold exactly.

