# OpenReview forum: "A Unifying View of Linear Function Approximation in Off-Policy RL Through Matrix Splitting and Preconditioning"
_NeurIPS.cc/2025/Conference — NeurIPS 2025 spotlight_

### Official Review · Reviewer_991y · 2025-07-01

**Clarity:** 3
**Significance:** 3
**Originality:** 3
**Rating:** 5
**Confidence:** 4

**Summary:**

This work analyzes the theoretical properties of policy evaluation algorithms in both the on- and off-policy settings. It characterizes the solutions of (expected) temporal difference update (TD), fitted Q iteration (FQI) and partial FQI (PFQI) using precise conditions (iff). The analysis is based on rank invariance (Condition 4.1). The paper then shows the convergence/stability properties of these methods based on rank invariance (Sec.5-7).

**Questions:**

For the cases of TD convergence versus FQI convergence, Appendix G.3 shows two counterexamples that one does not imply the other. However, it is not clear how these examples came to light or how they were specifically constructed. Further insights would be helpful. In addition, how likely are we going to see such cases in practice?

**Ethical Concerns:**

["NO or VERY MINOR ethics concerns only"]

**Final Justification:**

The paper provides a different angle for analyzing temporal difference learning using rank invariance, which is valuable for the RL community. As a result, I tend to accept it.

**Limitations:**

Yes

**Quality:**

3

**Strengths And Weaknesses:**

Strengths

1. The theoretical analysis focuses on a key condition of rank invariance (Condition 4.1), which is complementary to the feature independence condition widely used in the literature. It unifies three algorithms’ updates from a pre-conditioning point of view of solving linear systems. It also theoretically characterizes the singularity and consistency of the LSTD system using the rank invariant condition.

2. This new condition also offers new insights into the sufficient and necessary conditions for the convergence of the three algorithms with proper step sizes.

Weaknesses

1. A main concern is that the content in this work is very dense as a conference paper, and it seems that it would benefit from being a journal paper instead. This causes some problems such as notations not being clearly explained in the main text (e.g., $\sigma$ and $\mathfrak{R}$ in Corollary 6.3), and it has to constantly refer to the respective appendix sections for the complete discussion, which hurts the reading flow very much.

2. Some writing can be more precise. For example, L147: "Its solution is the same as the original linear system" but this should depend on the choice of $M$ (at least not iff).

Minor comments

- The capitalization of section titles is inconsistent throughout the paper.
- The space before the left brackets is inconsistent throughout the paper.
- Matrix bolding is inconsistent throughout the paper (e.g., see Proposition 4.2). It seems that not bolding any matrices would be easier as there are many $A, \Phi, \Sigma$, etc.
- $\theta_{\phi,r}$ not sure why we need to emphasize $\phi$ here. Throughout the paper, it seems that all that matters is to show that this depends on $r$.
- L106 misses transpose
- L124 can refer to the equation below it instead of (23) in the appendix
- L166 can refer to (8) instead of (67)
- L238 of of, L240 $\rho$ should be explained. Also, it seems that both $\rho$ and $\lambda$ are used in the paper for the same meaning.
- Eq.(31) is empty, and it makes more sense to switch (30) and (29) since (30) is (28) in matrix form.
- L736 uses $\theta_t$, (33) uses $\theta_k$ and (37) uses $\theta_{k+1}$. Please make sure the indices are consistent (maybe $k+1$ is the most clear). Also in the main text (L137).

---

> ### Author Rebuttal · Authors · 2025-07-31
>
> Thank you very much for your thoughtful feedback.
>
> ## Regarding the comment on weaknesses
>
> We understand your concern about the density of the paper and agree that the depth of the content may be better suited for a journal format.
>
> Due to space constraints, we placed several definitions and discussions in the appendix, which may have affected the reading flow. We appreciate you highlighting this and will revise the paper to bring the most essential notations and explanations into the main text to improve clarity and accessibility.
>
> ***Regarding the comment on writing precision:***
>
> Thank you for pointing this out. You mentioned that the statement on line 147—"Its solution is the same as the original linear system"—may depend on the choice of $M$. We appreciate the opportunity to clarify this.
>
> In this case, as stated just prior to that sentence, the preconditioner matrix $M$ is assumed to be a nonsingular matrix. Under this assumption, the preconditioned linear system $MAx = Mb$ indeed has the same solution set as the original system $Ax = b$. Therefore, in this case the choice of $M$ does not affect the solution set. We carefully chose the language to reflect this exact condition, and we believe the statement is precise in this context.
>
> That said, we truly appreciate your close reading and will consider clarifying this further to avoid any possible ambiguity for readers.
>
>
> ## To answer your question
>
> You asked about how likely such cases are in practice, and how these examples came to light. It's difficult to say how much a real world problem has in common with a 3-state, 2-feature MDP, but we can say that we emphasized linearly independent features because it is widely perceived to be a benign case where things are often well behaved. A practitioner aiming for a good set of features for linear approximation will often aim for linear independence as a starting point. So, we believe this is a reasonable assumption. From that starting point,  it's simply a matter of finding two pairs of features that satisfy the convergence conditions of one algorithm, but not the other. Since that was not difficult do intentionally, we don't expect it would be very hard to do accidentally in practice.
>
> ***Thank you once again for your insightful review and for helping us strengthen the paper.***

---

> > ### Comment · Reviewer_991y · 2025-08-08
> >
> > I thank the authors for the clarification. I have no further questions.

---

### Official Review · Reviewer_Fdq8 · 2025-07-02

**Clarity:** 3
**Significance:** 2
**Originality:** 2
**Rating:** 4
**Confidence:** 5

**Summary:**

The paper provides an analysis of the relationship between temporal different (TD), partially fitted Q iteration (PFQI) and fitted Q iteration (FQI) in the linear function approximation setting. The authors formulate the problem as solving a preconditioned linear system, with TD PFQI and FQI all  and uses this to analyse the stability and convergence

**Questions:**

Before the paper is published ANYWHERE, the authors need to do a better job of CAREFULLY ACKNOWLEDGING related worked instead of presenting their theorems as corrections, clarifications or novelties and claiming existing work is misleading.

Even with this essential correction, I'm not sure the paper contributes enough. A true unified view would have to account for nonlinearity, stochasticity and non-finite MDPs, as these have been tackled before and better represent the reality of where TD algorithms are actually used. I can't raise my score until this is the case.

Another approach would be to re-write the paper as a review and correction paper. Here previous results and conclusions from existing papers could be carefully collated (acknowledging contribution and similarities between works, corrected if necessary) and expanded upon for certain cases (like in the linear, expected finite case here). I'm not sure this would be a significant enough contribution on its own, but would leave that to the AC to decide.

**Ethical Concerns:**

["NO or VERY MINOR ethics concerns only"]

**Final Justification:**

I think my more major concerns have been addressed by the authors' rebuttal, however I do still believe the linear setting and the contribution  is somewhat limited given the results of prior work. My updated score reflects this.

**Limitations:**

The authors present many analyses for which identical or similar results have been found before without proper acknowledgement and in a much more restrictive setting. This significantly limits the paper's contribution and claims of a unified framework for analysing off policy RL.

**Quality:**

3

**Strengths And Weaknesses:**

STRENGTHS:

The paper is mathematically rigorous and well written. It clearly introduces notation and carefully analyses each algorithm systematically. One of the better parts of the paper is the correction of previous mistakes in SOME of the related work. A main contribution is the formulation of the problem as a preconditioned linear system. The paper then goes through analysing each of TD, PFQI and FQI, providing sufficient and necessary conditions for convergence of each. The discussion around each theoretical result is intuitive and offers nice insights. The paper also generalises analysis for when the Hessian is singular by analysing the Moore-Penrose pseudo-inverse.

WEAKNESSES:

This paper seems very similar to [1] in terms of its goals and I'm not sure it offers a significant enough contribution beyond this.

NOT AS GENERAL AS OTHER WORKS: The paper is limited to linear function approximation for finite state-action MDPs in the expected TD setting, whereas papers like [1] study general nonlinear smooth stochastic TD methods in general state-action MDPs. This is especially important in TD and related methods where nonlinearity is not only commonplace, but is a major source of instability itself, and so not accounting for this weakens the power of the analysis.

MANY RESULTS ARE RE-FORMULATIONS OF KNOWN RESULTS:

Many results can be recovered from [1] by noting that the following matrices are equivalent under the authors' notation for the special case of a linear function approximator: $ A_\textrm{LSTD} = J_\textrm{TD}(\omega,\omega^\star )$,  $ \Sigma_\textrm{cov} = H(\omega,\omega^\star;\omega_l )$  and $\gamma \Sigma_\textrm{cr} = J_\delta(\omega,\omega^\star;\omega_l )$.

In particular:

The update equation on line 140 is the same as Lemma 3 in [1]

Eq. 6 is simply the TD fixed point equation, which has been characterised many times before.

Analysis of FQI

Theorem  5.1 draws the same conclusions as Theorem 1 from [1] where convergence of FQI is governed by the convergence to zero of $\prod_{i=0}^{l-1} H(\omega_{i+1}^\star,\omega^\star;\omega_i^\star )^{-1} J_\delta(\omega_i,\omega^\star;\omega^\star ) $, which reduces down EXACTLY to the convergence of  $(H_\textrm{FQI})^l=(\gamma \Sigma_\textrm{cov}^{-1}\Sigma_\textrm{cr})^l $ in the linear case, thereby recovering the same result. The only slight generalisation in the authors' work is the use of the Moore-Penrose Pseudo inverse, however beyond this the result is not as general as [1] as it does not allow for nonlinear function approximation and general state-action spaces. Indeed, this motivated Assumption 5 in [1], which guarantees stability of FQI using the max  2-norm of  $H(\omega_{i+1}^\star,\omega^\star;\omega_i^\star )^{-1} J_\delta(\omega_i,\omega^\star;\omega^\star )$.

Analysis of TD

In theorem 6.1, the conclusion that $H_\textrm{TD}$ must be semiconvergent has been characterised in many ways before [1] [2]. When considering the conditions needed for the spectral norm to be less than 1 (as the authors identify), this recovers the famous deadly triad of TD [1].

Analysis of PFQI

The analysis of PFQI and connection to TD and FQI relies on the preconditioning formulation in Eq. 8. This has been characterised in [1] by the condition function in Eq. 7 $C(\alpha,k)$. The reason for the condition function in [1] is that under the more general nonlinear analysis, an analytic matrix equation does not exist, and so it acts as an upper bound in terms of max eigenvalues to account for the nonlinearity, but plays the same role in analysis and leads the the same conclusions:

Theorem 7.1 draws the conclusion that convergence relies on $H_\textrm{PFQI}$ being semi-convergent. In the more general nonlinear  stochastic setting of [1], the same conclusions can be drawn from Theorem 3 if $C(\alpha,k)<1$, showing convergence to a ball due to stochasticity. $C(\alpha,k)<1$ is a slightly more strict assumption, however this is needed due to the upper bounding because of nonlinearity, but implies $H_\textrm{PFQI}$ being semi-convergent in the linear case.

Results in Theorem 8.1 have been shown in [1] under an aysmptotic analysis: the authors of [1] show that convergence of PFQI stems from convergence of TD under the diminishing stepsize regime in Theorem 2, implying there exist a sequence of stepsizes such that PFQI always converges if TD converges.

In Theorem 8.3, the learning rate conditions for convergence given by the authors are the same as those in [1]: observe that the condition function $C(\alpha,k)$ converges for the same learning rate interval $(0,\frac{2}{ \lambda^\star_H})$ giving convergence of PFQI from FQI (Property 3 of [1]). These results are completely analogous to the convergence of the precondition function converging, albeit for a nonlinear analysis. The proof that convergence of FQI implies convergence of PFQI for large enough $t$ recovers the same result of Theorem 4 of [1] for large enough $k$ using the preconditioning matrix instead of the condition function.

Finally, the apparently novel conclusion that convergence of PFQI implies convergence of FQI, even when TD does not converge, is discussed extensively in Section 5.4 of [1] including a graphical representation of this property in Fig .1

STRAWMANNING OF RELATED WORK:

In my opinion, it is not acceptable that the authors claim novelty for the many of the main the results of the paper (Theorem 5.1, 6.1, 7.1 8.1 and 8.3) because, as I have shown above, they have been proved before. What is worse though is that the authors seem to suggest that ALL previous work is misleading or incorrect:

Line 32: `The traditional perspective fails to fully capture the convergence connections between these algorithms and may lead to incorrect conclusions. For example, one might erroneously conclude that TD convergence necessarily implies FQI convergence.'

I don't think any rational person would draw this conclusion from [1] as the connections that authors have made in this paper are also made clearly there.

NOT A SIGNIFICANT ENOUGH CONTRIBUTION:

In light of this, the current paper and introduced unified view does not offer a substantial contribution. The contributions are a slight clarification of necessary and sufficient conditions, a compiling of existing results and a minor technical relaxation of some assumptions (i.e. no need for linearly independent features in the on policy case and relaxation of invertible Hessian) at the expense of introducing additional assumptions that are too restrictive to apply for most practical methods - in particular finite MDPs, linear function approximation and analysis only in the path mean case.

LITTLE PRACTICAL BENEFIT:

In my opinion, the purported benefits of the introduced unified view and analysis have little practical benefit. Relaxing the assumption of an invertible Hessian makes little difference when a arbitrarily small amount of regularisation can easily satisfy the assumption. As mentioned before, the introduced stricter assumptions DO affect the ability to apply the analysis to real world algorithms.

Minor weaknesses (These don't count towards my score)

Typos:

Line 7: solving same linear system, Line 18: help(for batch TD), Line 139: Target Network.When, Line 184: (Condition 4.1). it can Line 194: features(Condition 4.3)as, Line 196 features(Condition 4.3), Line 198 features(Condition 4.3) Line 236:  invariance(Condition 4.1), Line 238: he impact of of rank invariance, Line 242: the target linear system the solution of the FQI linear system Line 262: The fixed point it converge to Line 327: features(Condition 4.3)do Line 327: invariance(Condition 4.1),

Titles seems to use a mixture of title case and non title case, sometimes in the same title

[1] Fellows et al. Why Target Networks Stabilise Temporal Difference Methods, ICML 2023. https://arxiv.org/pdf/2302.12537v3

[2] Tsitsiklis and Van Roy. An Analysis of Temporal-Difference Learning with Function Approximation, https://www.mit.edu/~jnt/Papers/J063-97-bvr-td.pdf

---

> ### Author Rebuttal · Authors · 2025-07-31
>
> We say how our work differs from previous work throughout the paper. If there are places where we have failed, we are happy to make changes. Previous work gave important insights into some questions we address, especially sufficient convergence conditions. Focusing on linear case, we provide a fuller characterization - both necessary and sufficient conditions. Such a characterization for what is essentially the entire suite of core RL algorithms has never been provided before, even in the linear case.
>
>
> Our results *do not apply only in the expected TD setting*. Our TD results extend to  stochastic and batch settings (line 129 and appendix E.14). Results about PFQI and FQI are easily adapted to the batch setting (line 1802, line 1815). The expected matrix form is simply for cleaner exposition.
>
> [1] explores the connection between TD and PFQI, and between PFQI and FQI, to understand better why target networks stabilize PFQI convergence for a general class of approximators.
> Our paper shares some similar goals, but is very different. We provide stronger, more varied, and more general results in the area of linear approximation. We also provide many additional results beyond those considered in [1] - see our response to reviewer KHfd.
>
> The class of nonlinear approximators considered in [1] doesn't include all linear approximators - not even the case of linearly dependent features, which is thoroughly studied in our paper. This case is important: It captures what is happening in the final layer of most deep nets, where feature vectors often aren't linearly independent. Linear versions of assumptions in [1] are quite restrictive - much milder sufficient conditions were discovered  years ago - explained later.
>
> [1] only provides SUFFICIENT condition for TD, PFQI and FQI under some assumptions, and establishes convergence connections between TD and PFQI, and between PFQI and FQI, based on these  conditions. We provide NECESSARY and SUFFICIENT conditions for convergence for each algorithm. We remove all standard assumptions under linear function approximation, such as the commonly assumed linear independence of features, and provide NECESSARY and SUFFICIENT conditions for TD, FQI and PFQI without this. We reveal the convergence connection between TD and PFQI,  between PFQI and FQI, AND BETWEEN TD and FQI (there is no convergence connection analysis between TD and FQI in [1]).
> It is only with necessary and sufficient conditions that you can rigorously say one algorithm's convergence implies the other (e.g. thm. 8.1.), and fully characterize when and why it doesn't (e.g. line 369).
>
> **Response to RE-FORMULATIONS OF KNOWN RESULTS**
> Contrary to the review, we never claim contribution for discovering or characterizing the TD fixed point equation nor the PFQI update equation. We cite the relevant sources.
>
> **Response to Analysis of FQI**
> Thm. 1 from [1] implies a connection between conditions for convergence in the linear case and $\left(H_{\mathrm{FQI}}\right)^l$. Even in the  linear case, that theorem would not fully characterize convergence of FQI as we did, since it would only provide a sufficient condition. Second, from the necessary and sufficient conditions we provide in our thm. 5.1, we see that the condition of thm. 1 from [1] is not entirely sufficient as it omits the required consistency condition.
>
> You mention slight generalization for using pseudo inverse, but this ignores why it's there. It is because we studied the case where the covariance matrix is not invertible or features are not linearly independent, leading to new insights, e.g. Prop. 3, App. B2, and it is a key component in understanding the influence of that setting on convergence of FQI (sec 5.1).
>
> **Response to Analysis of TD**
>
> [2] is noted for its on-policy TD result. Our paper is in the off-policy setting, a much more general case.
> In the on-policy setting our paper is the first show (line 299) that the common assumption of linearly independent features from [2] can be dropped.
>
> [1] only provides a sufficient condition for TD convergence for a class a nonlinear approximator. Its contribution is mostly for the nonlinear part, not the sharpness of the condition. In the linear case, it's just a sufficient condition already covered in [34].  [34] proves that eigenvalues having negative real parts is sufficient, but [1] is stricter: strictly negative eigenvalues. Both are covered by cor. 6.2. which provides NECESSARY and sufficient conditions for TD, and cor. 6.3 identifies all possible learning rates that make TD converge.
>
> Your comment about our use of the spectral norm is puzzling because we never use any spectral norm.
>
> **Response to Analysis of  PFQI**
> Eq. 7 in [1] uses Jacobian analysis to explore connections between TD and PFQI, and between FQI and PFQI. Converting to the linear case, it's not equal to preconditioner analysis. The Jacobian  matrices of TD and FQI in eq. 7 are completely different from the preconditioner of TD and FQI. In Eq. 7, TD's Jacobian matrix
> $\bar{J}_{T D}^{\star}$
> is
>
> $I-\alpha\left(\Sigma_{c o v}-\gamma \Sigma_{c r}\right)$
> . Our preconditioner is just $\alpha$. Also, thm. 3 in [1] only shows that assumptions 1, 2, and conditions on $C(\alpha, k)$ are sufficient for convergence of PFQI. We show the preconditioner of TD, FQI and PFQI largely determine the convergence properties of each algorithm, providing necessary and sufficient conditions for convergence of each.
>
> As you say, $C(\alpha, k)<1$ is a stricter condition than $H_{PFQI}$ being semiconvergent, and even $H_{PFQI}$ being semiconvergent alone is not enough as necessary and sufficient condition for PFQI; it also needs a consistency condition for a complete necessary and sufficient condition for FQI. Also, there is a gap between sufficient condition and a necessary and sufficient condition; the latter is mandatory to establish the complete convergence connections between different algorithms.
>
> Difference between Thm. 8.1 and thm. 2 in [1]: thm. 2 provides SUFFICIENT conditions (assumptions 1-4) for convergence with diminishing step sizes for PFQI, and these SUFFICIENT conditions also include SUFFICIENT conditions for TD to converge. Therefore, this established a connection between sufficient convergence conditions of TD and PFQI, but it does not imply PFQI always converges if TD converges.
> On the other hand, our thm. 8.1 shows that there exist constant step sizes for which TD convergence NECESSARILY implies PFQI convergence. That's a much stronger result because it covers every possible case where TD converges, where thm. 2 only holds when specific TD sufficient condition holds. One of these conditions (assumption 4) is a strong condition for linear approximation - less strict conditions were provided in [34] from 2002.
>
> For your comment on Thm. 8.3, the learning rates conditions are not the same. Our $\frac{2}{\lambda_{\max }\left(\Sigma_{\operatorname{cov}}\right)}$ is only dependent on covariance matrix, however, $\frac{2}{\lambda_H^{\star}}$
> where
>
> $$
> \lambda_H^{\star} := \sup_{\omega, \omega', \omega''} \operatorname{argsup}_{\lambda' \in \lambda\left(\bar{H}(\omega, \omega'; \omega'')\right)}  \left|1 - \alpha_l \lambda'\right|
> $$
>
>  from [1] are obviously also dependent on learning rate $\alpha_l$.
>
> Thm. 4 of [1] doesn't cover the result that convergence of FQI implies convergence of PFQI for large enough $t$, as [1] only provides sufficient conditions for FQI and PFQI convergence, then establishes connection between two sufficient conditions. That's not enough to say, in general, that convergence of FQI necessarily implies convergence of PFQI for large enough $t$.
>
> We never claim that convergence of PFQI implies convergence of FQI, even when TD does not converge as novel conclusion in this paper. In fact, "convergence of PFQI implies convergence of FQI" is not necessarily true, specifically when features are not linearly independent, as shown in Sec 3 and App B.2 that PFQI and FQI are not solving the same linear system, therefore, the convergence of PFQI for large enough $t$ can not even guarantee the existence of fixed point for FQI, nor convergence. As mentioned before [1] only establishes connection between sufficient conditions for PFQI and FQI. Therefore, it is not a consequence of [1] either.
>
> Response on concern about line 32 "strawmanning": We do not refer to [1] here. Nevertheless, we believe we do correctly represent an intuition that is widely held in the community.
>
> ***Response on NOT A SIGNIFICANT ENOUGH CONTRIBUTION***
> We hope we have clarified many ways our paper is different from [1], going well beyond what can be inferred by specializing the results from [1] to the linear case, and answering many open questions not addressed by [1] or any previous authors. We also hope we have clarified that our results do not apply only to expected TD, and explained that such a complete characterization of core algorithms has never before been provided for any function approximation class.
>
> ***Response on LITTLE PRACTICAL BENEFIT***
> As our paper is the only paper to provide  a complete characterization of the convergence behavior of linear function approximation for all major algorithms in the expected, batch, and stochastic cases, it should be the definitive resource for anybody using linear value function approximation. The idea that the possibility of regularization makes understanding the behavior of unregularized systems of little practical benefit seems dismissive. At minimum, it's helpful to know when and why regularization may be needed.
> We demonstrate a huge gap between what can be said about the linear case and what is known about the non-linear case, where few results on necessary conditions exist. As is often the case, we expect that resolving fundamental, unresolved questions in the linear case will eventually help answer similar questions in the non-linear case, perhaps starting with the final, linear layer in many deep networks.

---

> > ### Comment · Reviewer_Fdq8 · 2025-08-05
> >
> > I think my more major concerns with the paper have been addressed by the authors and I have raised my score accordingly

---

> ### Comment · Area_Chair_3jZs · 2025-08-01
>
> Thank you for the review. Would the referee please take a look at the rebuttal to assess the merits of the authors' arguments? Notably on the value of necessary and sufficient versus only sufficient conditions, and the fact that the class of approximators considered is not a special case of those in [1], and in addition has value when the features are not linearly independent; but also their other arguments on the preconditioner etc.

---

### Official Review · Reviewer_KHfd · 2025-07-02

**Clarity:** 3
**Significance:** 2
**Originality:** 2
**Rating:** 4
**Confidence:** 2

**Summary:**

This paper offers a unified perspective on reinforcement-learning algorithms, Temporal Difference (TD), Fitted Q-Iteration (FQI), and Partially Fitted Q-Iteration (PFQI), by viewing them as matrix-splitting schemes equipped with preconditioners. In the off-policy evaluation setting, the authors derive a single iterative update based on the state–action covariance and cross-covariance matrices, with TD, FQI, and PFQI emerging as special cases. Within this framework they introduce rank invariance, a condition that is both necessary and sufficient for the consistency of the target linear system. Rank invariance ensures existence and uniqueness of the solution and, in turn, leads to characterization of convergence conditions for the three algorithms.

**Questions:**

Because linear problems have been widely studied in many contexts, I’m somewhat concerned about the novelty of this paper related to rank invariance condition. In Appendix Section I, the authors discuss the convergence conditions used in earlier work. As I understand it, the linear-independence condition has already been addressed in prior studies, whereas the rank-invariance condition has not been clearly articulated. Is this interpretation correct? If not, could you clarify?

**Ethical Concerns:**

["NO or VERY MINOR ethics concerns only"]

**Final Justification:**

I am satisfied with the response and have no unresolved issues. As noted in my original review, since the paper’s strengths slightly outweigh its weaknesses, I will maintain my score.

**Limitations:**

yes

**Paper Formatting Concerns:**

I don't have any formatting concerns.

**Quality:**

3

**Strengths And Weaknesses:**

Strengths:
The paper’s main contribution is to clarify the role of rank invariance, a necessary and sufficient condition for the underlying linear system, and to distinguish it from simple linear independence condition. Leveraging this property, this work rigorously analyzes TD, FQI, and PFQI, yielding results that are both necessary and meaningful to the reinforcement learning theory literature.

Weaknesses:
The theoretical guarantees do not extend to policy control settings (nonlinear problems), and the paper does not verify its claims with empirical experiments.

---

> ### Author Rebuttal · Authors · 2025-07-31
>
> Thank you very much for your thoughtful question.
>
> First, we would be happy to clarify that Appendix Section I is intended to highlight and correct certain inaccuracies in the existing literature. For example, Proposition 3.1 in [17] is presented as offering necessary and sufficient conditions for TD convergence under the assumption of linearly independent features. However, as we carefully demonstrate in our paper, this result in fact provides only a **sufficient** condition. We present the correct necessary and sufficient conditions for this setting.
>
> More broadly, one of the central novel contributions of our work is to **establish necessary and sufficient conditions for the convergence of TD, FQI, and PFQI without making any assumptions about the features used** (e.g., linear independence) under linear function approximation. We further show how such commonly assumed feature conditions influence convergence behavior in each case.
>
> ---
>
> ### Response regarding the linear independence assumption and rank invariance
>
> We truly appreciate your interest in this point. The assumption of **linearly independent features** is indeed a widely adopted simplification in the analysis of reinforcement learning algorithms such as TD, FQI, and PFQI under linear function approximation. For example, many of the earlier works discussed in **Appendix I** rely on this assumption to facilitate convergence analysis.
> We would also like to emphasize that **rank invariance** is, to the best of our knowledge, a completely new and previously unrecognized condition in the literature.
>
> In prior works—such as Section 2.2 of [17]—it has been commonly stated that the **uniqueness of the TD fixed point in the off-policy setting** can be guaranteed solely by assuming **linear independence of the features**. However, through our analysis in this paper, we identify **rank invariance** as a distinct and essential condition, separate from linear independence.
>
> Most importantly, we rigorously prove in **Proposition 4.5** that the fixed point of TD is **unique if and only if both linear independence and rank invariance hold**. This result corrects a commonly accepted assumption and provides a stronger, more precise characterization of the uniqueness condition.
>
> We are confident that rank invariance has not appeared in prior literature and constitutes a genuinely novel contribution. Moreover, as we discuss in **Appendix C.2**, it is a **mild condition that widely holds in practice**. Our results show that it plays a surprisingly central role in:
> - Ensuring the **existence of fixed points** (Proposition 4.2), and
> - Influencing the **convergence behavior of TD, PFQI, and FQI**, as elaborated in the respective sections of the paper.
>
> ---
>
> In addition to the strengths you kindly pointed out, we would like to take this opportunity to **clearly itemize several key contributions** of our paper. We hope this provides a more accessible overview—especially in light of Reviewer 991y’s comment that the content may feel too dense for a conference paper.
> - **Unifying Perspective on TD, PFQI, and FQI:**
>   - Shows that TD, Partial Fitted Q-Iteration (PFQI), and Fitted Q-Iteration (FQI) are a single iterative methods solving the same linear system (the LSTD system), differing only in their choice of preconditioners and matrix splittings.
>
> - **Matrix Preconditioning Interpretation:**
>   - Interprets TD as using a constant preconditioner, FQI as using a data-feature adaptive preconditioner (inverse feature covariance), and PFQI as interpolating between them.
>   - Demonstrates that increasing the number of updates under a fixed target value function corresponds to transitioning from a constant to an adaptive preconditioner.
>
> - **Clarification of Convergence Conditions:**
>   - Provides necessary and sufficient convergence conditions for TD, PFQI, and FQI under linear function approximation, without making any assumptions about the chosen features
>   - Identifies limitations of commonly used assumptions in prior literature.
>
> - **Introduction of the *Rank Invariance* Condition:**
>   - Proposes a new condition---rank invariance---that is necessary and sufficient for universal consistency of the target linear system (i.e., solvability for any reward function). It is also necessary and sufficient to ensure that every solution of the target linear system is a desired solution if it exists.
>   - Highlights the distinction between rank invariance and linear independence.
>   - Shows that when rank invariance holds, FQI is an iterative algorithm that uses the proper splitting scheme to formulate its iterative components and preconditioner.
>   - Demonstrates that rank invariance widely exists in practice and is guaranteed to hold in the on-policy setting.
>
> - **Conditions for Uniqueness of Solutions:**
>   - Proves that the target linear system has a unique solution if and only if both rank invariance and linear independence hold.
>
> - **Encoder-Decoder View of TD:**
>   - Introduces an encoder-decoder interpretation of TD's convergence condition, clarifying why TD may diverge even under overparameterization.
>
> - **Analysis of On-Policy and Off-Policy Settings:**
>   - Shows that rank invariance always holds in the on-policy setting, guaranteeing the existence of a fixed point.
>   - Extends on-policy TD convergence guarantees to cases without linearly independent features, showing that this common assumption can be dropped.
>
> - **New Insights into Learning Rates:**
>   - Demonstrates that when a learning rate ensures TD convergence, then all smaller learning rates will also ensure convergence.
>   - Provides guidance where to choose learning rate to make batch TD converge when it can converge
>
> - **Convergence Relationships Among TD, PFQI, and FQI:**
>   - Provides formal implications (and counterexamples) regarding convergence dependencies between TD, PFQI, and FQI.
>   - For example, if TD converges, PFQI with a small learning rate also converges. Moreover, despite the common perception that FQI is more stable than TD, TD convergence does not guarantee FQI convergence—our analysis explains why.
>
> - **Relevance to Deep RL and Nonlinear Function Approximation:**
>   - Extends insights to neural networks by noting that final linear layers justify applying results beyond the assumption of feature linear independence.
>
>
>
> ---
>
> ### Response to the comment on the Weaknesses
> We agree that it would be nice to extend insights from the current work to policy control settings (nonlinear problems). As noted by the Reviewer 991y, the paper is already quite dense and rich with results, many of which appear in the appendix because of space limitations. We hope the reviewer would agree that extensions to policy control settings are best deferred for future studies.
>
> With respect to the reviewer’s comment on the absence of empirical experiments to support our claims,
> We agree that experiments can play an important role in papers, especially when there are gaps between theory and practice. In such cases, experiments can reassure that theory still informs practice, and/or suggest new directions for inquiry. However, not all theoretical contributions require experiments. Some of our theoretical work, for example, is motivated by known empirical results. Other results provide reassurance about the conditions under which algorithms are guaranteed to perform well (necessary and sufficient conditions). This is an important thing to know, yet it is impossible to verify experimentally since these are statements about all possible sets of models and features. Still other results show that the good behavior of one algorithm does not necessarily imply the good behavior of others. This is validated in Appendix G.3, where we provide a specific MDP and set of features.
>
> **Thank you again for your thoughtful and constructive feedback—it has been invaluable in improving our work**

---

> ### Comment · Area_Chair_3jZs · 2025-08-05
>
> Dear referee,
>
> Could you please take a look at the author's rebuttal and update your review/score if and as needed?
>
> Thank you!

---

> ### Comment · Reviewer_KHfd · 2025-08-06
>
> We thank the author for the response and have no further questions. We maintain the original score.

---

### Official Review · Reviewer_iyii · 2025-07-03

**Clarity:** 3
**Significance:** 3
**Originality:** 3
**Rating:** 5
**Confidence:** 4

**Summary:**

This paper studies the off-policy policy evaluation (OPE) problem with linear function approximation. The authors study the TD, FQI, and Partial FQI (PFQI) algorithms and establish that they can be viewed as solving the same "target" linear system but with varying choices of splitting schemes and preconditioners. The authors study this "target" linear system and establish necessary and sufficient conditions for the existence and uniqueness of a solution. The authors study the convergence of TD, FQI, and PFQI and establish necessary and sufficient conditions for each of these, with connections to each other, and elucidate why convergence of TD and FQI do not necessarily imply convergence of each other.

**Questions:**

The convergence guarantees can be clarified a bit: e.g. how do the solutions $(A_{FQI})^D b_{FQI} + (I - A_{FQI}(A_{FQI})^D)\theta_0$ and  $A_{LSTD}^Db_{LSTD} + (I - (A_{LSTD})(A_{LSTD})^D)\theta_0$ of Theorem 6.1 relate to the desired solution (e.g. assuming realizability, how does it relate to $\theta^\star$?)

**Ethical Concerns:**

["NO or VERY MINOR ethics concerns only"]

**Limitations:**

Yes

**Paper Formatting Concerns:**

Several equations in the appendix overflow into the margin

**Quality:**

3

**Strengths And Weaknesses:**

This is a very solid paper which in my view should clearly be accepted to the conference. The results are not extremely technical in hindsight (all the proofs boil down to linear algebra) but impactful for understanding these fundamental algorithms and providing definitive convergence conditions and relations between them. Despite their simplicity, many of these results have been previously documented as far as I am aware.

I have some minor complaints:
- Typos: there are many typos and spacing issues throughout (especially in the appendix but even in the main text). Please review.
- Clarity: There are some undefined terminology and notation especially when it comes to linear algebra. E.g. in Section 5 and 6: what is a semiconvergent matrix and what is the notation (A)^D? What is a proper splitting, and what are the implications of it (why should I care about Lemma 5.2)? What is the drazin (group) inverse? Bringing some of these concepts which are necessary for interpreting the results out of the appendix and into the main text would improve interpretability.
- Missing discussion of related work: the condition that $\rho(\gamma \Sigma_{cov}^\dagger \Sigma_{cr}) < 1$ is necessary and sufficient for FQI convergence has been noted in Perdomo et al. (reference [30] of this paper), and the condition that nonsingularity of $A_{LSTD}$ is necessary and sufficient for OPE has been noted in Mou, Pananjady, Wainwright (https://arxiv.org/abs/2012.05299), Amortila, Jiang, Xie (https://arxiv.org/abs/2011.01075) and Amortila, Jiang, Szepesvari (https://arxiv.org/abs/2307.13332).

---

> ### Author Rebuttal · Authors · 2025-07-31
>
> **Thank you so much for taking the time to thoroughly review our paper and for recognizing its contributions—we truly appreciate your thoughtful feedback.**
>
> **In response to your minor concerns:**
>
> **Regarding the typos:**
> Thank you for pointing this out. We will carefully review the manuscript and make sure to correct any such issues.
>
> **Regarding clarity:**
> The terminology and notation are indeed defined in the paper; however, due to space constraints, we placed all definitions related to linear algebra in Appendix A.1, as noted in Section 2 (Preliminaries) of the main text. That said, we very much appreciate this comment. It’s a valuable suggestion, and we agree that moving some of the most frequently used or important definitions into the main body could improve accessibility and clarity for readers.
>
> **Regarding the missing discussion of related work you kindly pointed out:**
> Thank you very much for bringing these papers to our attention.
> Perdomo et al. (reference [30] in our paper) indeed provides a valuable finite-sample analysis. However, in their Section 3 on FQI preliminaries, they state that
>
> $$
> \theta_T = \sum_{k=0}^T \left(\gamma \Sigma_{\mathrm{cov}}^{-1} \Sigma_{\mathrm{cr}}\right)^k \Sigma_{\mathrm{cov}}^{-1} \theta_{\phi, r}
> $$
>
> converges to the true value function parameter $\theta_\gamma^{\star}$ only if $\rho(\gamma \Sigma_{\mathrm{cov}}^{-1} \Sigma_{\mathrm{cr}}) < 1$, which suggests that this condition is necessary. We would like to gently point out that this is not strictly correct.
>
> As shown in our Theorem 5.1 and Proposition D.3, the power series can still converge when $\gamma \Sigma_{\mathrm{cov}}^{-1} \Sigma_{\mathrm{cr}}$ is *semiconvergent* (definition provided in line 544), which includes cases where $\rho(\gamma \Sigma_{\mathrm{cov}}^{-1} \Sigma_{\mathrm{cr}}) = 1$.
>
> We also note that in the broader literature, it is a common misunderstanding to treat $\rho(A) < 1$ as a necessary and sufficient condition for the convergence of power series of the form
>
> $$
> X_{t+1} = \sum_{i=0}^{t} A^i B + A^{t+1} X_0
> $$
>
> derived from the iterative update equation $X_{t+1} = A X_t + B$. As we discuss in Appendix I, many prior works have made this same assumption. One of the contributions of our paper is to help clarify and correct such misconceptions in the literature.
>
> **Regarding your mention of three papers stating that $A_{\text{LSTD}}$ being nonsingular is necessary and sufficient for OPE:**
> We greatly appreciate your comment, though we were unable to locate this specific claim in the referenced works. Moreover, this statement does not generally hold. For instance, as we show in Appendix G.3, there are concrete examples where $A_{\text{LSTD}}$ is nonsingular, yet both TD and FQI fail to converge.
>
> If you wouldn’t mind pointing us to the specific statements or sections where this claim appears—or sharing a bit more about what you had in mind regarding $A_{\text{LSTD}}$ being a necessary and sufficient condition for OPE—we would be sincerely grateful. We would be more than happy to revisit those works with your guidance.
>
> **To answer your question:**
> Thank you for the thoughtful and insightful question! We address this point thoroughly in our paper. The two solution sets you mentioned are formally defined as $\Theta_{\text{FQI}}$ (line 160) and $\Theta_{\text{LSTD}}$ (line 174), with the desired solution set denoted as $\Theta_\pi$ (line 215).
>
> In Proposition 3.1, we prove that $\Theta_{\text{LSTD}} \supseteq \Theta_{\text{FQI}}$ always holds, and we provide the necessary and sufficient condition under which $\Theta_{\text{LSTD}} = \Theta_{\text{FQI}}$. Later, in Proposition 4.8, we show that $\Theta_{\text{LSTD}} \supseteq \Theta_\pi$ always holds, and we also give necessary and sufficient conditions under which every solution of the target linear system is indeed a desired solution—that is, when $\Theta_{\text{LSTD}} = \Theta_\pi$.
>
> We hope this fully addresses your question, and we’d be happy to clarify further if needed!
>
> ***We are grateful for your insights and thank you again for your valuable comments.***

---

> > ### Comment · Reviewer_iyii · 2025-08-07
> >
> > Dear authors,
> >
> > Thanks for your clarifications.
> >
> > Regarding $A_{LSTD}$ being necessary and sufficient. One can examine, for example, Theorem 3 of the Perdomo et al. paper, which establishes that all "linear estimators" will fail when $A$ is singular. Proposition 1 of the Amortila, Jiang, Xie paper also suffices: in their construction, $A_{LSTD}=0$ and no estimator can recover the correct value function, even asymptotically. This is extended in the Amortila, Jiang, and Szepesvari paper to the misspecified case. This covers the necessity. Regarding the sufficiency, one can simply run the "plug-in" LSTD estimator (no need to bother with TD or FQI), whose finite-sample performance is analyzed in Theorem 2 of the Perdomo et al. paper.
> >
> > Otherwise, I have no further questions at this time and maintain my positive score.

---

> ### Author Response · Authors · 2025-08-08
>
> Thank you so much for clarification. We definitely want to cite previous work appropriately.
>
> We'd like to bring to your attention that theorem 3 from Perdomo et al. does not strictly prove all "linear estimators" will fail when $A_{LSTD}$ is singular. The informal version of theorem 3 at Page 5 differs from what the formal version on page 15 says. In the formal version and proof, it only proves the necessity of $A_{LSTD}$ being nonsingular under the linearly independent features assumption (covariance matrix is invertible). Please note that this is assumed in the second bullet in the statement of the theorem on page 15.
> However, in our proposition 4.8 we prove the TRUE necessary and sufficient conditions without the linearly independent features assumption. Their result is implied by ours, but our result is not implied by theirs because it is more general.
>
> Additionally, the related results e.g. Theorem 4.2. and 4.3 of the Amortila, Jiang, and Szepesvari paper are also under assumption of linearly independent features (invertible covariance matrix) which is their Assumption 2.3, which we believe applies to all results in the paper. Nevertheless, even if they did not intend for this assumption to apply to their theorem 4.2, it is implied by their assumption on $\sigma_{min}$ in the statement of theorems 4.2 and 4.3.
>
> Regarding the Amortila, Jiang, and Xie paper: We still need a little help understanding the connection. It seems that their work is concerned with questions of statistical efficiency under the assumption of linearly independent features (implied by part 2 of their coverage assumption, which requires a lower bound on the eigenvalues). Our work does not address statistical efficiency and also does not make a linear independence assumption. So, while that work seems quite interesting an important in terms of understanding the statistical efficiency challenges in applying our results in the batch setting, we do not see how it contradicts or subsumes our results. Please help us understand if we have overlooked something.
>
> In any case, we will endeavor to cite these papers properly and correctly acknowledge their true contribution in any final version.

---

### Comment · Area_Chair_3jZs · 2025-08-01
**Reviewer-author discussion start**

Thanks to everyone for writing the paper, evaluating it, and drafting reviews and rebuttals!

Could the referees please take a look at the authors' rebuttal and continue the discussion/amend the reviews as necessary?

Thank you again!

---

### Note · Authors · 2025-08-15

We appreciate all reviewers for their insightful feedback and constructive comments, and for providing suggestions that would improve our paper.
We are encouraged to find that the reviewers recognized the paper as “impactful for understanding these fundamental algorithms and providing definitive convergence conditions and relations between them” (by Reviewer iyii), “is mathematically rigorous and well written” (by reviewer Fdq8),
“a very solid paper which in my view should clearly be accepted to the conference”(by Reviewer iyii).

We would like to take this opportunity to clarify a couple of points raised in the reviews.

First, we thank reviewers for recognizing the contribution of rank invariance. Specifically, reviewer KHfd notes the value of rank invariance over previous analyses that focused on other criteria, such as linearly independent features. Reviewer 991y also notes the significance of rank invariance. While we appreciate this, we also noticed that these reviews might be interpreted as having gotten the impression that rank invariance plays a role in our convergence analysis. To avoid any misunderstandings, we'd like to emphasize that rank invariance does not appear in the necessary and sufficient convergence conditions in our theorems, which can be viewed as assumption-free. Rank invariance is still highly important though because of how it influences the target linear system and, therefore, the fixed points of these algorithms (as detailed in Section 4). Rank invariance can also lead to relaxed convergence conditions in some cases. For example, observe that the semi-convergence requirement is relaxed in Corollary 5.3 in comparison to Theorem 5.1.

Secondly, Our TD results extend to stochastic and batch settings (as we specified in line 129 and appendix E.14), not just for expected TD. Results about PFQI and FQI are easily adapted to the batch setting (line 1802, line 1815). The expected matrix form is simply for cleaner exposition.

In closing, we sincerely appreciate the reviewers’ recognition of the novelty, rigor, and significance of our work, and we thank them once again for their thoughtful evaluations and constructive feedback.

---

### Decision · Program_Chairs · 2025-09-17

**Decision:**

Accept (spotlight)

**Comment:**

Referees (and my own reading) agree that this paper is impactful in linear-approximation RL, and provides a novel understanding of the temporal-difference, fitted Q-iteration, and partially fitted Q-iteration methods. This is despite most of the analysis resulting from "basic" linear algebra -- which may in itself be seen as a strength of the paper.

Concerns on relation to related work already discussed, and to missing related work; as well as to the focus on the linear case whereas nonlinear is more widely used these days; are sufficiently addressed by the authors in the discussion with the referees. A key point here is the relation to [1] in Fdq8, which the authors resolve by stating they newly provide *necessary* and sufficient conditions.

Overall, my recommendation is positive given the paper, reviews, and discussion. Furthermore, since the paper elucidates important technical questions about widely used algorithms in a way that is relatively easy to follow, I further recommend classifying it as spotlight.